# Estimating Biases During Detection of Leads and Lags Between Climate Elements Across Dansgaard–Oeschger Events

John Slattery[1, 2], Louise C. Sime[1], Francesco Muschitiello[2], and Keno Riechers[3, 4]

[1]Ice Dynamics and Paleoclimate, British Antarctic Survey, High Cross, Madingley Road, Cambridge CB3 0ET, United Kingdom
[2]Department of Geography, University of Cambridge, 43 Saint Andrews Street, Cambridge CB2 1DB, United Kingdom
[3]Research Domain IV – Complexity Science, Potsdam Institute for Climate Impact Research, Telegrafenberg A31, 14473 Potsdam, Germany
[4]Earth System Modelling, School of Engineering & Design, Technical University of Munich, 80333 Munich, Germany

**Correspondence:** John Slattery (johatt11@bas.ac.uk)

**Abstract.** Dansgaard–Oeschger (DO) events occurred throughout the last glacial period. Greenland ice cores show a rapid warming during each stadial to interstadial transition, alongside abrupt loss of sea ice and major reorganisation of the atmospheric circulation. Other records also indicate simultaneous abrupt changes to the oceanic circulation. Recently, an advanced Bayesian ramp fitting method has been developed and used to investigate time lags between transitions in these different climate elements, with a view to determining the relative order of these changes. Here, we critically review this method in both its original implementation and a new, extended implementation. Using ice-core data, climate model output, and carefully synthesised data representing DO events, we demonstrate that both implementations of the method suffer from biases of up to 15 years. These biases mean that the method will tend to yield transition onsets that are too early. Further investigation of DO warming event records in climate models and ice-core data reveals that the biases are on the same order of magnitude as potential timing differences between the abrupt transitions of different climate elements. Additionally, we find that higher-resolution records would not reduce these biases. We conclude that decadal-scale leads and lags between climate elements across DO events cannot be reliably detected, as we cannot exclude the possibility that they result solely from the biases we present here.

## 1   Introduction

Proxy records from Greenland ice cores provide evidence for millennial-scale climate variability throughout the last glacial period (Dansgaard et al., 1982; Johnsen et al., 1992; Dansgaard et al., 1993; NGRIP members, 2004). The most striking feature of this variability is the repeated occurrence of Dansgaard–Oeschger (DO) warming events, during which Greenland temperatures increased by up to 15 degrees K in just a few decades (Steffensen et al., 2008; Kindler et al., 2014; Capron et al., 2021) as Greenland rapidly transitioned from a cold stadial to a warm interstadial state. Alongside this rapid warming in Greenland, there is evidence for abrupt retreat of sea ice in the North Atlantic and Nordic Seas (Li et al., 2005, 2010; Dokken et al., 2013; Hoff et al., 2016; Sime et al., 2019; Maffezzoli et al., 2019; Sadatzki et al., 2020), and a reinvigoration of the Atlantic Meridional Overturning Circulation (AMOC) from a rather weak stadial to strong interstadial state (Gottschalk et al.,

2015; Henry et al., 2016; Lynch-Stieglitz, 2017). Furthermore, beyond the North Atlantic, a number of paleoclimate archives provide evidence for global-scale reorganisations of the atmospheric circulation (Markle et al., 2017; Schüpbach et al., 2018; Buizert et al., 2018; Erhardt et al., 2019) including a northward displacement of the inter-tropical convergence zone (ITCZ)
(Schneider et al., 2014) and changes in global precipitation patterns (Fohlmeister et al., 2023). The latter are particularly well documented in the Asian (Wang et al., c, b; Li et al., 2017) and South American Monsoon regions (Wang et al., a; Kanner et al., 2012; Cheng et al., 2013), as well as the European–Mediterranean region (Drysdale et al., 2007; Fleitmann et al., 2009; Moseley et al., 2014; Corrick et al., 2020). However, as of yet it has not been possible to conclusively identify a trigger for these rapid Dansgaard–Oeschger warming events (Capron et al., 2021). These periods of abrupt warming in the North Atlantic can
be considered as but one of four parts in a larger DO cycle (Lohmann and Ditlevsen, 2019). Consistent with previous studies (Erhardt et al., 2019; Riechers and Boers, 2021), we do not consider the whole cycle but only the various abrupt changes that occurred during the stadial to interstadial transitions. Henceforth, we refer collectively to these abrupt changes simply as DO events.

One paradigm in which we can consider DO events is as cascades of sudden changes in different climate variables. From
35 this perspective, an analysis of the temporal order of events, in models or in the paleoclimate record, should unravel the mechanistic dynamics of DO events. It is however important to note that such an approach relies on the assumption that all DO events are realisations of the same underlying mechanism, which may not be the case. Previous research has attempted to identify temporal lags between the sudden changes in different proxies by analysing multi-tracer records from Greenland (Steffensen et al., 2008; Erhardt et al., 2019; Capron et al., 2021; Riechers and Boers, 2021). This has the advantage that jointly
measured proxy variables usually reflect the state of different components of the climate system free of any relative dating uncertainty. Nonetheless, such analysis is still difficult due to the challenge of inferring transition times from noisy data.

A Bayesian ramp-fitting method, designed to address this challenge, has been developed and presented by Erhardt et al. (2019). Stacking multiple DO events in a multi-proxy analysis of Greenland ice-core data, Erhardt et al. (2019) identified time lags between the different proxies. Considering DO events back to 60,000 years before present, the authors concluded
that atmospheric changes preceded the reduction in sea-ice extent by around a decade. Using the same method, Capron et al. (2021) followed up on this research by extending a similar multi-proxy analysis of Greenland ice-core data back to 120,000 years ago. Unlike the previous study, Capron et al. (2021) suggested that any leads and lags between climate elements might be impossible to detect due to both the tight coupling of the different climate elements and the substantial variability between different DO events.

Subsequently, Riechers and Boers (2021) re-examined the same ice-core data considered by Erhardt et al. (2019), using the same ramp-fitting method but adopting a different statistical framework that more rigorously propagates the uncertainties in the onset time of each DO event. Under this new framework, the authors found that the time lags between proxies are not statistically significant, in disagreement with Erhardt et al. (2019). Therefore, although the method developed by Erhardt et al. (2019) holds much promise, it has not yet led to a conclusive understanding of the temporal phasing of DO events. Nonetheless,
the previous research in this area has left open the possibility that such an understanding could be achieved in future through the application of this method to either improved ice-core records or data from model simulations.

Intriguingly, thanks to advances in both model development and computing power, spontaneous DO-like millennial variability is now captured by least six General Circulation Models (Brown and Galbraith, 2016; Vettoretti and Peltier, 2016; Klockmann et al., 2020; Zhang et al., 2021; Kuniyoshi et al., 2022; Malmierca-Vallet et al., 2023). The millennial variability is spontaneous in the sense that it is not externally forced by changes to atmospheric carbon dioxide concentrations or orbital parameters or by freshwater hosing. Whilst these model simulations are imperfect representations of real DO events, they nonetheless provide an invaluable means to help us investigate the question of whether it is possible to conclusively identify a trigger for rapid Dansgaard–Oeschger warming events. We can therefore use such model simulations to test the feasibility of determining the trigger and order of changes during DO-like warming events.

We build upon these recent advances, examining in detail the causes of uncertainty in the onset time of DO events in different paleoclimate proxies and model variables. This also enables us to comment on whether it may be possible to determine the order of changes within a DO cascade. Our manuscript firstly extends and critically reviews the Bayesian ramp fitting method provided by Erhardt et al. (2019), investigating whether the method is biased depending on the characteristics of a given transition as well as possible approaches to bias correction for application to real-world data. Secondly, we apply this method to data from the CCSM4 model (Vettoretti et al., 2022), chosen because the CCSM4 simulated DO events closely match real DO events in terms of their magnitude of Greenland warming, the duration of stadial and interstadial periods, and the bipolar seesaw relationship between Greenland and Antarctica. Thirdly, we revisit the original Greenland multi-proxy data from ice cores and comprehensively investigate relevant biases. Finally, we discuss the implications of our findings for whether the temporal phasing of DO events can be determined from paleo-data.

## 2    Data & Methodology

### 2.1    Model Data

The Spontaneous Dansgaard-Oeschger type Oscillations in climate models (SDOO) project, under the EU-TiPES program, gathers together available simulation output from models which show DO-like oscillations (Malmierca-Vallet et al., 2023, https://www.bas.ac.uk/project/sdoo/#data;). Included in this data-set are decadal mean output from six CCSM4 model runs which Vettoretti et al. (2022) have provided. Each of these six simulations is 8000 years long, and exhibits millennial-scale variability which strongly resembles DO events as observed in paleoclimate archives. The runs are forced using last glacial maximum boundary conditions (Vettoretti et al., 2022) alongside varying concentrations of atmospheric carbon dioxide, from 185 to 230 ppm. This range closely matches the atmospheric carbon dioxide concentrations during Marine Isotope Stage 3, when DO events were most frequent (Vettoretti et al., 2022). The number of events in each simulation depends on the $CO_2$ concentration, with the highest DO frequency at 200ppm (Vettoretti et al., 2022). In total, there are 19 abrupt warming events in these six simulations.

We select five variables from CCSM4 in order to compare the timing of their relative transitions. These are North Atlantic Surface Air Temperature, Precipitation, and sea-ice extent, as well as the North Atlantic Oscillation (NAO), and the Atlantic Meridional Overturning Circulation (AMOC). The first four of these are chosen as they have previously (Capron et al., 2021)

been used in a similar analysis of earlier simulations from the same model, though we consider a different region. We add the AMOC to this selection due to the integral part it is thought to play in DO events (*e.g.* Lynch-Stieglitz, 2017; Li and Born, 2019; Malmierca-Vallet et al., 2023). We calculate time series for Temperature, Precipitation, and Sea Ice by taking area-weighted means over a selected region of the North Atlantic (shown in Figure 4). AMOC is given by the spatial maximum of the stream-function in the North Atlantic between 20° and 60° North at any depth. The NAO index is calculated as the first principal component of sea level pressure across the region 30° - 80° North and 80° West to 40° East.

From the resulting time series, we visually identify the abrupt warming events. For the analysis we isolate the individual events in data windows of 800 years approximately centred on the transition. To ensure consistency, these same search windows are used for all five variables of interest.

Alongside the full data-set that is available at 10-year resolution, there are a small number of annually-resolved simulations. Although we do not have a sufficient number of annually-resolved abrupt warming events to assess potential systematic time lags, an application of the Bayesian ramp-fit allows us to gain a sense of the ranges of the ramp and noise parameters. This in turn allows us to gain insight into how increased resolution impacts the bias in the ramp fitting method.

## 2.2 Ice-Core Data

Alongside the model data, we also revisit data from the North Greenland Ice Core Project (NGRIP) (NGRIP members, 2004) that were previously analysed using this method (Erhardt et al., 2019). We make use of the data provided by Erhardt et al. (2019) for four proxies from this core over 16 stadial-interstadial transitions ranging in time from the Holocene onset to the onset of Greenland Interstadial 17.2 at around 60,000 years before present. Whilst Erhardt et al. considered a larger set of transitions, these 16 are the only ones for which they were successfully able to apply their method to all four proxies.

The four proxies in this data-set are $\delta^{18}$O, annual layer thickness, concentration of dust aerosol ($Ca^{2+}$ ions), and concentration of sea salt aerosol ($Na^+$ ions). For $\delta^{18}$O, the temporal resolution decreases from four years at the Holocene onset to seven years at 60,000 years before present, whilst for the other three proxies this resolution decreases from one to three years over the same period. These four proxies have previously been interpreted as representing the air temperature at the core site, precipitation at the site, large-scale Northern Hemisphere atmospheric circulation, and sea-ice extent in the oceans around Greenland (Erhardt et al., 2019). Our focus in this work is on the ramp fitting method itself rather than a detailed consideration of the ice-core proxies that we are applying it to and so we do not make any link between particular ice-core proxies and particular model output variables.

## 2.3 Ramp Fitting Method

The Bayesian method developed by Erhardt et al., and previously applied to DO events recorded in ice cores and speleothems (Adolphi et al., 2018; Erhardt et al., 2019; Capron et al., 2021) models each event as the sum of a deterministic linear transition (or ramp) between two fixed equilibria and an additive AR(1) noise process as follows:

$$x_i(t_i) = \hat{x}_i(t_i) + \epsilon_i, \tag{1}$$

$$\hat{x}_i(t_i) = \begin{cases} x_0 & \text{for} & t_i \le t_0 \\ x_0 + \frac{x_1 - x_0}{t_1 - t_0}(t_i - t_0) & \text{for} & t_0 < t_i < t_1 \\ x_1 & \text{for} & t_i \ge t_1. \end{cases} \tag{2}$$

The addition of the AR(1) noise process $\epsilon_i$ makes the model stochastic. For a given time series that comprises a DO event, the introduction of appropriate prior distributions directly yields Bayesian posterior distributions for the model parameters $\{t_0, t_1, x_0, x_1, \alpha, \sigma\}$, where $\alpha$ and $\sigma$ define the autocorrelation and the amplitude of the noise. The meanings of each of these parameters, their prior probability distributions, and a full description of the AR(1) noise model are given in Appendix A.

We observe that both the model and ice-core datasets contain some abrupt transitions with gradual trends before or after the transition itself (Figure A1). This behaviour is not captured by the original Erhardt formulation, and so we extend the method by adding possible slopes before and after the linear ramp. This leads to improved agreement of the transition model with the analysed data (Figure 1). This is particularly the case for AMOC in the CCSM4 model, which shows an exaggerated "saw-tooth" shape with a strong downward slope following the abrupt transition. In such cases where slopes are clearly present either before or after the transition, our extended method also leads to reduced uncertainty in the onset time $t_0$ and end time $t_1$ of the transition.

Additionally, for cases where slopes are present our extension of the method reduces the sensitivity of the transition timing to the search window, which is otherwise one of the drawbacks of this method (Capron et al., 2021). In order to apply the ramp-fit to individual transitions, one has to select a data window around that transition. There are no objective criteria for the starting point and the endpoint of these windows, other than that no other transition should be included. However, changing the boundaries of the data window influences the results of the estimation. This effect is reduced if slopes before and after the transition are allowed. For example, for an arbitrarily chosen transition of AMOC in the CCSM4 model, we applied both the original and extended ramp-fitting method to the same transition using 25 different search windows and found that the standard deviation in the posterior mean onset time decreased from 16.0 years when using the original method to just 4.3 years when using the extended method (Figure A3).

However, if no such slopes are present then our extended method results in increased uncertainty due to the addition of two unnecessary extra parameters. A full, quantitative, assessment of the performance of the two different implementations of the ramp-fitting method under different conditions can be found in Appendix A. Overall, when assessed in terms of timing uncertainty, neither the original nor the extended method are obviously superior and so we consider both throughout our analysis.

## 2.4 Hypothesis Testing

We require a means by which to test the statistical significance of any time lags that we may discover in either the model or ice-core data. Although the ramp fitting method is Bayesian, we adopt a frequentist perspective for our statistical analysis following Riechers and Boers (2021). For our analysis of the CCSM4 model, we define time lags for the transition onset of

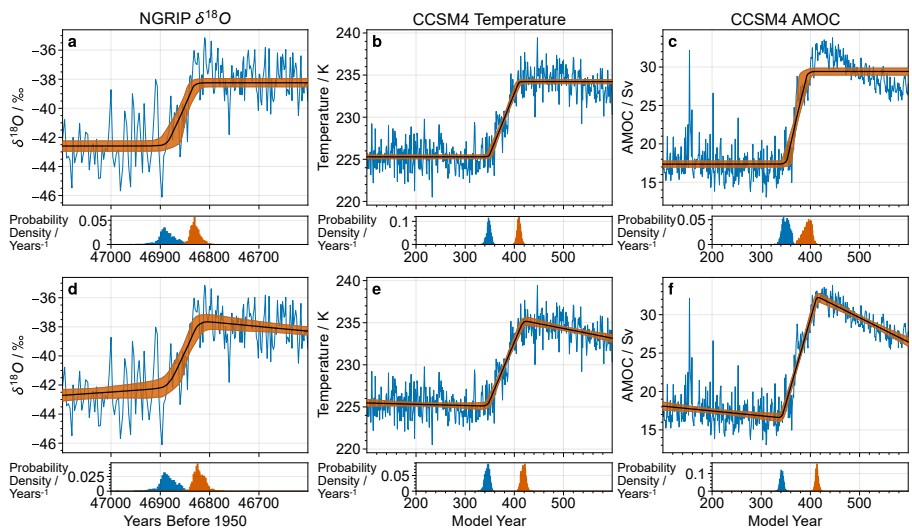

**Figure 1.** A comparison of the output when applying both the original (flat) implementation of the method (panels (a)–(c)) and the extended (slope) implementation (panels (d)–(f)) to three different transitions. The time series are shown in blue, with the median deterministic ramp as estimated by the Bayesian method in black and the 5-95% range shaded in orange. Below, the posterior distributions for the onset time and end time of the rapid transition are shown as blue and orange histograms, respectively. In panels (a) & (d), the two different implementations of the ramp-fitting method are applied to $\delta^{18}$O from the NGRIP core (Erhardt et al., 2019) covering the transition to GI-12c. Panels (b) & (e) show surface air temperature at the nearest grid cell to the NGRIP site across a transition from CCSM4 (Vettoretti et al., 2022), whilst (c) & (f) show AMOC for this same transition. The addition of slopes before and after the transition yields a clear improvement of the visual fit of the deterministic ramp to the CCSM4 data in panels (e-f), as compared to (b-c).

each of the other four variables relative to the transition onset in temperature $T$.

$$\Delta t^x = t_0^x - t_0^T. \tag{3}$$

A negative $\Delta t^x$ thus corresponds to an earlier transition onset in variable $x$ as compared to the temperature transition onset.

  We regard the $\Delta t^x$ from the different simulated DO events as realizations of a repeated (identical) random experiment. This view is only meaningful if one accepts the "one-mechanism" hypothesis, which states that all DO events follow the same mechanism and that differences in their expression are exclusively due to internal climate variability. While this hypothesis may not be generally accepted for the real-world's DO events, we believe that the regularity of stadial-interstadial cycle seen

in the DO simulations provides a strong argument that at least the simulated DO events are all governed by the same physics. We further emphasise that a statistical assessment in terms of hypothesis tests relies on the "one-mechanism" hypothesis and would be meaningless if individual DO events were caused by different physical drivers.

  We conduct pairwise hypothesis tests by comparing each of the other four variables to temperature using the following hypotheses:

– $H_0$: The population mean time lag of this variable relative to temperature is equal to zero.

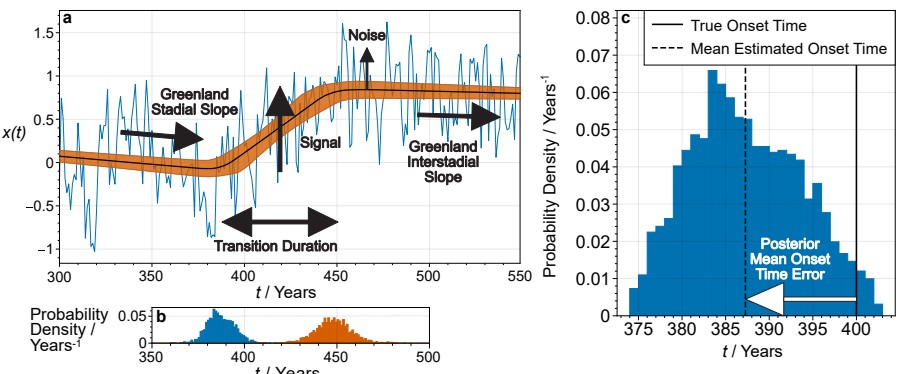

**Figure 2.** Panel (a): An example of the application of our extended method to a synthetic transition with annual resolution, where the parameters of interest for the assessment of bias are marked. Panel (b): The posterior distributions for the onset and end time of the transition. Panel (c): An enlarged reproduction of the posterior distribution for the transition onset time with the True Onset Time and Posterior-Mean Estimate Onset Time shown. The True Onset Time (thick line) is set to be the year 400. The Mean Estimated Onset Time is the mean of the posterior distribution for the onset time $t_0$ produced by the application of the Bayesian ramp fitting method to this noisy data. The Posterior-Mean Onset Time Error (PMOTE) is given by the difference between these two times, and so a negative PMOTE indicates that the transition onset has been estimated to occur earlier than it truly does.

- $H_1$: The population mean time lag of this variable relative to temperature is not equal to zero.

We perform two-tailed tests as we have no prior indication as to the sign or direction of any time lags that may be present.

We conduct our analysis of the NGRIP ice core in the same manner. In this instance, we calculate time lags for the other three proxies relative to $Ca^{2+}$ and conduct hypothesis tests as above.

$$\Delta t^x = t_0^x - t_0^{Ca^{2+}}. \tag{4}$$

Note that the choice of reference variable / proxy is arbitrary and separate between the model and ice-core datasets as we are not drawing any equivalence between particular model variables and ice-core proxies.

### 2.5 A Note on Nomenclature

In this work, we are dealing with two types of randomness. The first is the uncertainty in the determination of the transition time as reflected by Bayesian posterior distributions of the $t_0$ parameter. Secondly, as stated previously, we regard any set of abrupt transitions (be it in synthetic test data, GCM data or an ice-core record) as repeated realisations of the same random experiment. The transition times of the different climate components assume the role of (correlated) random variables in this perspective.

Correspondingly, there are two types of averages that we use throughout this work. For sake of clarity, we refer to averages over Bayesian posterior distributions of parameters as *posterior-averages* or *posterior-means*. Means over different realisations of the random experiment (i.e. over different events or transitions) are called *event-averages* or *event-means*.

## 3 Results

### 3.1 Synthetic Transitions

As noted in the introduction, we wish to systematically test the ramp fitting method and investigate whether there are any biases in its estimation of transition timing. Indeed, we identify the following transition parameters (Figure 2) as potential sources of bias: (i) Noise / Signal Ratio ($\xi$), (ii) Autocorrelation Time of the Noise, (iii) Greenland Stadial Slope, (iv) Greenland Interstadial Slope, (v) Transition Duration. Note that the Noise / Signal Ratio is calculated by dividing the amplitude of the AR(1) noise by the amplitude of the transition, and that we normalise the slopes relative to the amplitude of the transition to allow for comparison between different variables and proxies.

To quantify the strength of the bias, we construct synthetic transitions by the addition of randomly generated AR(1) noise to a piecewise linear ramp, exactly as in the extended transition model (Appendix A). Our intention is not to mimic true data as realistically as possible but instead to create data to which the ramp fitting method should be perfectly suited. We consider transitions with temporal resolutions of 10 years (decadal) and 1 year (annual) in order to investigate the impact of different resolutions on the bias. In all cases, the synthetic data series are 800 years long, with an abrupt transition that starts at the year 400. By comparing this true onset time to the posterior-mean onset time as estimated by the ramp fitting method, we calculate the Posterior-Mean Onset Time Error (PMOTE) for each synthetic transition (Figure 2). Although for an individual synthetic transition the PMOTE is sensitive to the particular realisation of the AR(1) process, the event-average of the PMOTE with respect to the AR(1) noise can be identified with the bias of the method. Accordingly, we take a further event-mean over 100 separate synthetic transitions for each unique combination of parameter values.

Our aim is ultimately to assess the degree of bias that is propagated to estimates of leads and lags. It is not clear whether the posterior-mean or posterior-median onset time more suitable to represent (or integrate) the bias, especially because two conflicting approaches have been proposed for the calculation of leads and lags from a set of events (Erhardt et al., 2019; Riechers and Boers, 2021). One would expect the posterior-mean onset time to generally be somewhat earlier than the posterior-median due to the asymmetry of the posterior distribution, which arises because the transition duration has a lower bound at zero but no upper bound. To ensure that this does not overly affect our results, we therefore also consider the posterior-median onset time in our systematic testing.

The ranges over which the parameters for the synthetic transitions are varied are chosen to reflect the ranges observed for these parameters when applying the ramp fitting method to different events and variables in the CCSM4 model simulations as well as different events and proxies in the NGRIP ice-core record. The appropriate ranges for the Noise / Signal Ratio ($\xi$) and the Autocorrelation Time of the AR(1) noise differ between the cases of decadal and annual resolution because higher-resolution data is generally noisier. Note that the Greenland Interstadial Slope is negative in all cases, reflecting the classic "saw-tooth" shape of DO events, however we plot the absolute value of this slope for ease of legibility.

We would ideally like to vary all five transition parameters simultaneously in order to assess possible inter-dependencies of the bias on different transition parameters. However, it would be challenging to visualise and interpret the resulting five-

215 dimensional parameter-space. We therefore vary only two transition parameters at any one time - Noise / Signal Ratio and one other - whilst keeping the rest fixed at standard values as follows:

- Autocorrelation Time = 10 years for the case of decadal resolution and 1 year for the case of annual resolution.

- Greenland Stadial Slope = 0 Kiloyears$^{-1}$.

- Greenland Interstadial Slope = -1 Kiloyears$^{-1}$. Note that the Greenland Interstadial Slope is always $\leq 0$ in our sensitivity
tests due to the saw-tooth shape of DO events. In Figure 3 we plot the absolute value or magnitude of this slope, which is positive. Note that the slopes are normalised relative to the amplitude of the transition. This standard value for the Greenland Interstadial Slope therefore means that the time series will return to its pre-transition level 1000 years after the transition ends.

- Transition Duration = 50 years.

Our systematic testing of the ramp fitting method using synthetic transitions finds that the onset time estimate can be biased due to the transition parameters (Figure 3). This is true for all four combinations of method implementation and temporal resolution considered here, but there are differences in the extent of the bias and the dependence on different parameters.

Firstly, for the original method we find that the Greenland Stadial Slope preceding the abrupt transition is the key driver of onset time bias, as shown by panels (b) & (f) of Figure 3. For both the NGRIP and CCSM4 events this slope is generally
positive, meaning that it is in the same direction as the abrupt transition itself. We find that this leads to a too-early bias that can exceed 10 years, although the extremities of the range considered here (and so the largest biases) occur only rarely in the NGRIP data. We also see bias arising from variations in the other parameters, but not to the same extent.

For the extended method there is generally a too-early bias of up to 10 years when the noise is high . We also find that the level of noise controls the strength of the impact of the other four parameters; when $\xi$ is small, variations in the other parameters have little impact on the bias, but when this ratio is high variations in the other parameters have a large effect ($\sim 10$ years).
Focusing on the case of decadal resolution (panel (h) of Figure 3), for $\xi \sim 0.2$ an increase in the transition duration from 10 years to 100 years may even reverse the bias from $\sim -10$ years to $\sim +8$ years. It is therefore clear that the noise / signal ratio $\xi$ is the key parameter in determining the bias of our extended ramp fitting method, but that the other parameters identified here also play an important role.

For both implementations of the ramp fitting method, the broad pattern of bias observed is similar between the cases of annual and decadal resolution. It is not possible to say whether the bias is generally greater for one case or the other, as the transition parameters at annual resolution map onto those at decadal resolution in a complex manner. This means that points in the same positions of their respective panels in Figure 3 are in fact not directly comparable between the two resolutions. We conduct further testing in Section 3.2 to explicitly address this point.

When compared to the original Erhardt et al. (2019) implementation of the ramp fitting method, our extended implementation that includes slopes leads to a large reduction in the bias for transitions that have strong slopes before or after the transition

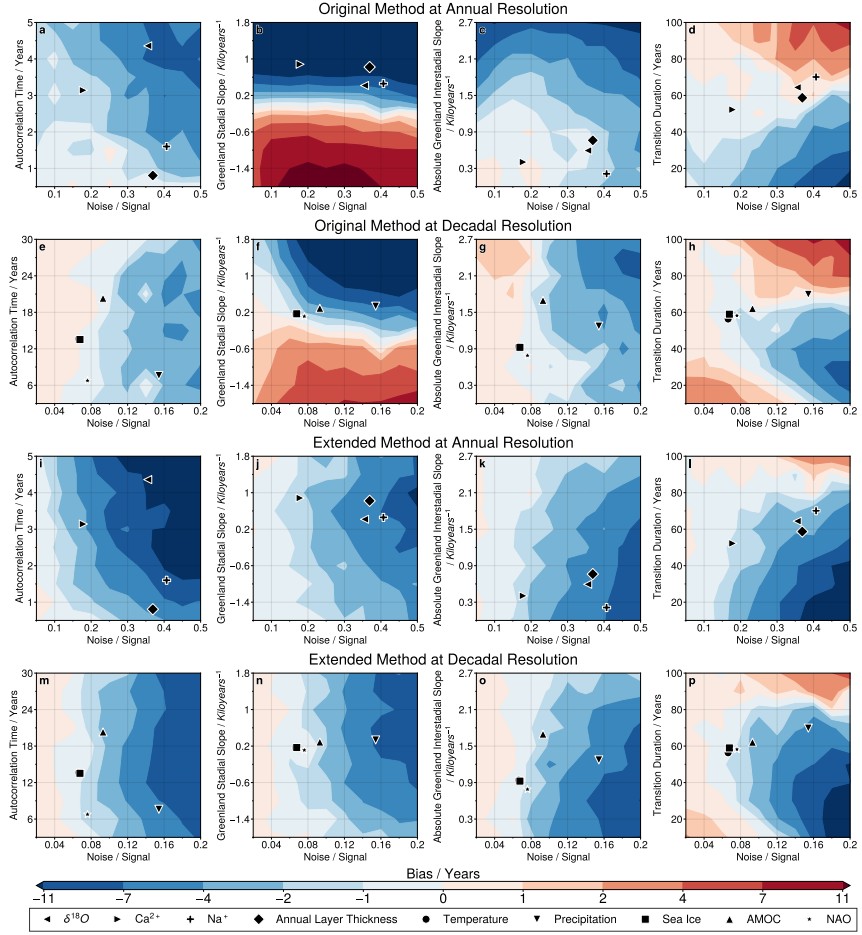

**Figure 3.** The bias in the posterior-mean transition onset time as a function of the five transition parameters. We consider both the original (a–h) and extended (i–p) implementations of the ramp fitting method at both annual (a–d & i–l) and decadal (e–h & m–p) resolution. Each panel contains 100 unique sets of parameter values, and for each set of parameters we take an event-average over 100 synthetic transitions. When not explicitly varied, parameters are fixed at standard values as given in Section 3.1. Blue colours indicate a bias towards transition onset times that are too early, with red indicating the opposite. The mean transition parameters for the four proxies from NGRIP are overlaid on panels (a–d & i–l) and for the five variables of interest in the CCSM4 model on panels (e–h & m–p). Note that the scales for Noise / Signal and Autocorrelation Time differ between the cases of annual and decadal resolution.

but relatively low values of $\xi$. In the opposite situation, where $\xi$ is large and there are no slopes present, our extended implementation instead leads to increased onset time bias. Neither is obviously more or less biased when considered across the whole range of possible parameter values.

The bias that we generally observe towards too-early onset times also leads to our estimates of the transition duration being too long. This is important because the transition duration is itself one of the determining factors of the bias in the transition onset time for all four combinations of method implementation and temporal resolution (Figure 3 (d), (h), (l), & (p)). Our lack of unbiased knowledge of the true transition duration therefore makes it extremely challenging to accurately estimate the bias affecting any particular model variable or ice-core proxy.

Further systematic testing reveals that such bias persists, though somewhat altered, when adopting an alternative set of prior probabilities (Figure C1) or even when taking the simplest possible approach of a least-squares fit to a linear ramp (Figure C2). This might suggest that such bias is a fundamental limitation of any attempt to identify the timing of abrupt transitions in noisy climate data. However, we equally cannot prove that there is no possible unbiased method - certainly in many other statistical settings there are estimators that remain unbiased even as their uncertainty increases. The important point for our analysis is

that we do not currently possess any unbiased method that we could use to calibrate or corroborate the transition onset time or duration.

Defining the bias as the expectation of the error in the posterior-median onset time, rather than the posterior-mean, results in slightly more positive values for the bias (see Figure C3), meaning that the generally negative biases we observe are somewhat decreased in magnitude. However, the differences are small and the overall pattern unchanged. This demonstrates that

the choice between posterior-mean and posterior-median when estimating bias is unimportant, and so we only consider the posterior-mean going forwards.

## 3.2    Impact of Changing Resolution

We are interested as to how the bias depends on the temporal resolution of the data. Although in Section 3.1 we have considered both annual and decadal resolution, it is far from trivial to make direct comparisons between the two. For climate time series,

the changes in the noise parameters $\xi$ and autocorrelation time when changing resolution depend on the full power spectrum of the noise. This is different for every variable or proxy and so there is no generally applicable relationship with which to predict the impact of changing resolution. Furthermore, as previously stated we do not have access to sufficient annually resolved data from the CCSM4 model to allow for a meaningful comparison.

Instead, we proceed by creating two further sets of synthetic transitions at annual resolution. These two sets of transitions

represent two contrasting cases of low-autocorrelation ("whiter") noise or high-autocorrelation ("redder") noise, with parameters chosen based on the extremes of the range observed in the CCSM4 model. For each of these sets of synthetic transitions, we then down-sample to decadal resolution by dividing the annual time series into sections of 10 years and taking the mean over each section as a single data point for our decadal time series. (Fig C4). Finally, we separately apply our extended ramp fitting method to the annually and decadally resolved transitions.

**Table 1.** The bias in the posterior-mean onset time for two different types of noise at both annual and decadal resolution in our extended ramp fitting method, each calculated from a sample of 1000 synthetic transitions. Uncertainties are given by the standard deviation of a bootstrapped distribution for the sample mean onset time error. This ensures that the uncertainties of the individual onset times are rigorously propagated.

| Type of Noise | Annual Bias / Years | Decadal Bias / Years |
|---------------|---------------------|----------------------|
| Whiter | $-3.1 \pm 0.3$ | $-3.4 \pm 0.3$ |
| Redder | $-7.2 \pm 0.5$ | $-7.0 \pm 0.5$ |

For both types of noise we find that the bias is unchanged, within uncertainty, when switching between the two resolutions. These results are summarised in Table 1. We also verify that when down-sampling to decadal resolution it does not matter whether the transition onset occurs at the edge of two averaging-sections or in the middle of one. We can therefore state that the temporal resolution has no impact on the bias, although higher resolution does at least reduce the uncertainty in the onset time estimates for individual transitions.

### 3.3 Bias in the CCSM4 Model and the NGRIP Ice-Core Record.

Following on from the above systematic testing, we investigate whether the ramp fitting method introduces any bias to the timing estimation of DO events in the CCSM4 model. To do this, we construct 1000 synthetic transitions that are "analogous" to each of the model variables of interest and assess the corresponding expected PMOTE as described above. The expected PMOTE serves as a first order approximation of the bias for each investigated climate variable and may thus be subtracted from the transition onset estimate obtained with the Bayesian ramp fitting method. To calculate representative parameter values we take both the posterior-mean and the event-mean over the corresponding marginal posterior distributions obtained by applying the extended ramp fitting method to the 19 transitions in the CCSM4 model simulations. For each variable, this gives a single value for each transition parameter (shown in Figure 3), which we use to create the "analogous" transitions.

We make an exception for transition duration, choosing instead to use a fixed duration of 50 years for all of our "analogous" transitions. This is because we suspect that much of the apparent difference in duration between different variables is in fact due to the bias we have identified. As we have been unable to identify any unbiased method by which to estimate the true transition durations, we simply fix them at a plausible value based on the durations of the transitions in two ice cores as estimated in a previous study (Capron et al., 2021).

The situation with precipitation is more difficult. Visual inspection of the precipitation time series reveals a much greater degree of noise during the stadials than the interstadials (Figure C5). To test the impact of this, we investigate a further set of synthetic transitions which have two different noise regimes - one during the stadial and the other during the interstadial. We find that the value of $\xi$ in the stadial is most important to the bias in the onset time (Fig C6). Because of this, creating synthetic transitions using a single noise regime is likely to underestimate the magnitude of the bias affecting precipitation, and

so it is important that we include these two distinct levels of noise for precipitation. To do so, we further adapt the Bayesian ramp fitting method, producing a new version that treats noise before and after the transition separately. We find that this has a negative impact on the success rate of the Markov Chain Monte Carlo (MCMC) sampler that forms part of the method, in that there are many more cases where the sampler fails to converge, and so we do not attempt to use this implementation to assess the timing. Instead, it simply provides a means of calculating appropriate parameter values to use for the synthetic transitions that are "analogous" to precipitation.

For completeness, we construct two sets of synthetic transitions that are "analogous" to precipitation: One where we do not account for the differing noise between stadials and interstadials, and one where we do. Our estimates of the magnitude of the bias are increased by around 2 years for the original method and around 3 years for the extended method when we include the two distinct noise regimes in our synthetic transitions (see Table 2). This confirms our expectation that failing to do so would lead to an under-estimate of the degree of bias.

Finally, we similarly investigate potential bias in the estimated transition timing of DO events in the NGRIP ice core. Here, we follow the same procedure as outlined above for CCSM4 model data. There is however an additional complexity due to the differing time resolutions. For simplicity, we choose round numbers that are representative of the resolution of each proxy. We therefore use a resolution of 2 years for $Ca^{2+}$, $Na^+$, and the annual layer thickness, and a resolution of 5 years for $\delta^{18}O$. Whereas for the CCSM4 model we used 800-year sections, here we create synthetic time series that are 500 years long to match the ice-core data. The true transition onset times are fixed to the year 250 and the transition durations are fixed at 50 years, for the same reason as discussed with regards to the model data.

The estimated biases for both the variables in the CCSM4 model and the proxies in the NGRIP ice core, and using both implementations of the ramp fitting method, are listed in Table 2. Reflecting the general pattern found in Section 3.1, all of the variables and proxies are negatively biased, that is towards onset times that are too early. Importantly, the biases are different for different variables and proxies, due to differences in both the ramp shapes and noise properties, and hence these biases limit our ability to assess leads and lags between associated transitions of different variables or proxies.

### 3.4 Time Lags in the CCSM4 Model in Light of the Bias.

In the following, we compare the transition onset times of the climate variables introduced in Section 2.1 at DO events simulated by the CCSM4 model. In doing so, we treat all simulated DO events equivalently, irrespective of the chosen $CO_2$ concentration. In particular, we take into account the bias which we quantified in the previous section. For each available DO event, we apply the extended ramp fitting method to our selection of climate variables on the predefined data window.

For sea ice, AMOC, and NAO, we observe that the transition onsets occur approximately simultaneously with those in temperature, irrespective of the bias correction (Figure 4), and so we do not discuss the hypothesis tests for these three variables. For precipitation the story is different. Without any bias correction the event-averaged time lag appears to be negative with certainty. Even under consideration of the bias, the uncertainty distribution of the precipitation-temperature lag is centred around -10 years with only small probabilities for a positive lag. This indicates that the transition onsets in precipitation are likely to occur before those in temperature. However, it is not clear whether this is statistically significant. Performing a hypothesis test

**Table 2.** Mean estimates and uncertainties for the biases affecting different variables in the CCSM4 model and different proxies in the NGRIP ice core, resulting from their differing noise and slope characteristics, using both the original and extended methods. Each bias is calculated as the mean of the Posterior-Mean Onset Time Error (PMOTE, see Figure 2) across a sample of 1000 synthetic transitions. The uncertainties are given by the standard deviation of a bootstrapped distribution for the sample mean onset time error. These reflect the uncertainty in the bias given a particular set of transition parameters, with the uncertainty in the timing of each individual transition rigorously propagated. There is additional uncertainty due to our imperfect knowledge of the true transition parameters that we have not attempted to quantify. The values in brackets for precipitation are the biases if we neglect the two separate noise regimes. As expected, these underestimate the true biases.

| CCSM4 Variable / NGRIP Proxy | Onset Time Bias Using Original Method / Years | Onset Time Bias Using Extended Method / Years |
|---|:---:|:---:|
| CCSM4 Temperature | -0.7 ± 0.2 | -1.0 ± 0.2 |
| CCSM4 Precipitation | -8.8 ± 0.7 (-7.1 ± 0.6) | -8.9 ± 0.6 (-6.0 ± 0.5) |
| CCSM4 Sea Ice | -0.9 ± 0.2 | -0.8 ± 0.2 |
| CCSM4 NAO | -1.5 ± 0.2 | -1.2 ± 0.2 |
| CCSM4 AMOC | -1.8 ± 0.4 | -2.1 ± 0.3 |
| NGRIP $\delta^{18}$O | -7.6 ± 0.8 | -10.2 ± 0.9 |
| NGRIP Annual Layer Thickness | -9.2 ± 0.5 | -6.0 ± 0.5 |
| NGRIP Na$^+$ | -6.8 ± 0.6 | -9.4 ± 0.6 |
| NGRIP Ca$^{2+}$ | -9.7 ± 0.5 | -4.8 ± 0.4 |

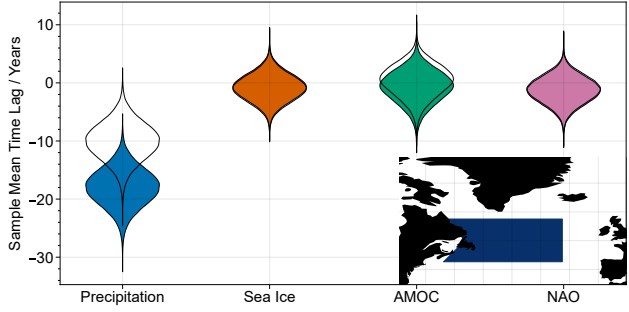

**Figure 4.** Uncertainty distributions for the uncorrected (filled) and bias-corrected (transparent) sample mean time lags of precipitation, sea ice, AMOC, and NAO as compared to temperature in the CCSM4 model, according to our extended ramp fitting method. The bias correction is applied both to the reference variable, temperature, and to the target variable. Inset is a map showing the region of the North Atlantic over which temperature, precipitation, and sea ice have been averaged. (Vettoretti et al., 2022).

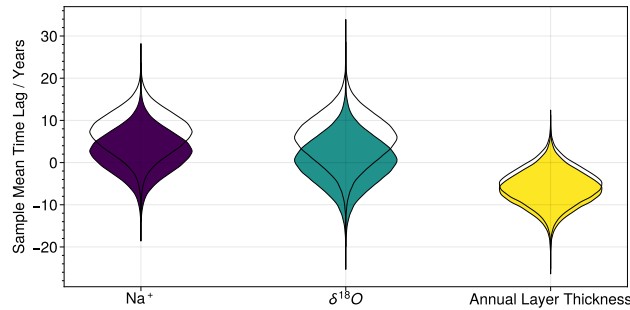

**Figure 5.** Probability distributions for the uncorrected (filled) and bias-corrected (transparent) sample mean time lags of $Na^+$, $\delta^{18}O$, and annual layer thickness as compared to $Ca^{2+}$ in the NGRIP ice core (NGRIP members, 2004; Erhardt et al., 2019), according to our extended ramp fitting method. The bias correction is applied both to the reference proxy, $Ca^{2+}$, and to the target proxy.

as described in Section 2.4, we calculate a p-value of 0.078. As this is slightly greater than the standard significance threshold of 0.05, we cannot ultimately rule out the null hypothesis that the transitions in temperature and precipitation occur simultane-

340 ously. This demonstrates how the bias we have identified can lead to false conclusions about the significance or otherwise of time lags between different variables or proxies.

### 3.5 Time Lags in the NGRIP Ice Core in Light of the Bias

Following the same procedure as for the model warming events, we calculate the sample mean time lags of $Na^+$, $\delta^{18}O$, and annual layer thickness, relative to $Ca^{2+}$, in the NGRIP ice core using our extended ramp fitting method. We again use the

345 "analogous" synthetic transitions to perform a bias correction. Note that in this case each random sample is comprised of 16 of these synthetic transitions, as this is the size of our sample of DO events in the ice core. For each of these proxies, Figure 5 clearly shows that the bias-corrected distributions are consistent with zero time lag, and so we do not discuss the formal hypothesis tests.

### 4 Discussion

We have demonstrated that the commonly used ramp fitting method is biased by up to 10 years when estimating the onset time of DO events. This is comparable to the magnitude of the time lags we might expect to find between different components of the climate (Erhardt et al., 2019), and so this bias severely limits the trust that we may have in any time lags identified in this manner. We have attempted, in Section 3.4, to correct for this bias in order to produce a more accurate estimate of the true time lags. In doing so, we found that even a large apparent time lag, such as the 18-year time lag we observed between precipitation

and temperature in the CCSM4 model, could simply be the result of methodological bias.

One important caveat is that the bias we have identified is generally fairly small relative to the timing uncertainty of individual proxies across single DO events, which is usually several decades (see e.g. Figure A2). Our bias of up to 10 years is therefore

unlikely to critically impact the findings of studies which only consider and compare individual transitions, for example Capron et al. (2021). However, the calculation of leads and lags may involve stacking many events so as to reduce the uncertainty - sometimes to as little as ± 5 years (Erhardt et al., 2019). In this context, the bias that we have identified is clearly extremely important to consider as it could easily lead to false conclusions, as we saw for the model data in Section 3.4.

Capron et al. (2021) also test for possible bias using synthetic transitions with autocorrelated noise and find no significant bias in either the transition midpoint or duration, implying that the transition onset time must also be unbiased. This stands in contrast to our findings in this study, and so merits further consideration. We suggest that the transitions tested by Capron et al. (2021) lie in a region of the parameter space for which the bias is small, even at high levels of noise. The most obvious reason for this is that the absence of any slopes before or after the ramp favours small biases. We observe in Figure 3 that, when using the original Erhardt et al. (2019) implementation, synthetic transitions without any slopes before or after the ramp show very little bias, even for relatively high levels of noise. However, the bias grows rapidly when even slight slopes are present. We therefore suggest that a key cause of the discrepancy is the assumption made by Capron et al. (2021) that there are no slopes in either the stadial that precedes the transition or the interstadial that follows. Similarly, Figure 3 shows that for certain "sweet spots" in the transition duration the bias remains small even as the level of noise increases. It could also be the case that the transition duration chosen by Capron et al. (2021) happens to lie in such a sweet spot, and that this is partly why they do not find significant bias.

In the frequentist framework that we have adopted, two key uncertainties enter into our hypothesis test for the significance of the difference in onset time between precipitation and temperature. These are the inherent uncertainty in the timing estimate of DO events and the uncertainty in the mean of a finite sample, which combine to form large uncertainties in both the observed and empirical null time lags. Considering this, the p-value of 0.078, though above the standard significance threshold of 0.05, may still appear rather small. This could suggest that the potential lead of precipitation merits further investigation. However, the standard significance threshold should really be adjusted downwards to correct for the fact that we have made multiple comparisons by comparing each of precipitation, sea ice, AMOC, and NAO to temperature. We do not feel the need to explicitly make this correction in this case because our finding is not significant even at the uncorrected threshold, and also because it is not clear precisely how this should be done. Nonetheless, this further strengthens the view that it is not possible to make any firm conclusion regarding the temporal phasing of DO events.

This is even more-so the case when we consider the major limitations of our bias-correction. The first of these is that we have followed the simplest possible approach of using single values for the parameters of our synthetic transitions. It would be more appropriate to instead sample from the range of parameter values exhibited by the same variable / proxy across different DO events. However, utilising the distribution of parameters in this way leads to estimates of the bias that are so uncertain as to lose meaning, and so we have used single parameter values (as given by combined posterior- and event- averages). Furthermore, the bias depends in a highly non-linear manner on interactions between the different transition parameters, and so our assumption that the bias resulting from the mean parameters is equal to the mean bias resulting from the whole range of parameters is a poor one.

Another issue limiting the possibility for bias correction is our inability to estimate the transition duration in an accurate manner. This has a major impact because the transition duration is a key determinant of the bias in the transition onset time, however the duration itself is also affected by this bias. We are thus stuck in an impossible situation whereby we cannot know the duration without first knowing the bias, but we simultaneously cannot know the bias without first knowing the duration. Because of this, the best we are able to do is to guess a plausible transition duration, for which we chose 50 years in all cases. The fact that these DO transition durations also cannot be determined is another important source of uncertainty that cannot be captured by a bias-correction procedure.

Given these limitations of the bias-correction procedure, there are clearly additional uncertainties affecting the time lags discussed in Sections 3.4 & 3.5 which we have not attempted to quantify. Even without incorporating these additional uncertainties, we reiterate that the identification of a statistically significant order to the DO events remains very challenging. Higher temporal resolution would allow us to somewhat reduce the uncertainty in the timing estimate of each DO event. But, as Section 3.2 shows, the temporal resolution of the data comprising a transition has no impact on the bias in the ramp fitting method. There is therefore limited prospect for improving this situation in the future through access to higher resolution model output data or indeed higher resolution paleo-measurements from ice cores.

## 5  Conclusions

Bayesian ramp fitting methods have shown great potential as a tool for identifying the temporal phasing of the changes in different climate components during climate transitions. For Do events, however, we demonstrate that even the advanced method of Erhardt et al. (2019) suffers from bias when estimating the onset time of these very fast transitions due to a combination of noise and the presence of gradual slopes before and after the transitions. Directly incorporating said slopes into the ramp fitting method reduces the extent of the bias when significant slopes are in fact present in the data, but doing so also worsens the problem when they are not, as is often the case. The bias ranges from approximately - 15 years to + 5 years, and so is comparable to or longer than the potential relative phasing of the climate elements that we seek to resolve. This severely limits the reliability of any time lags, whether in models or ice cores, that are found using this ramp fitting method. It is difficult to ascertain the magnitude of this bias in any individual case for any DO events, and so any attempt to correct for the bias will necessarily introduce major additional uncertainty.

This leads to the conclusion that it may never be possible to confidently determine the order of Dansgaard-Oeschger warming events from these types of methods, no matter how many new ice cores are drilled or higher resolution measurements taken, because neither of these would alleviate the fundamental problem of bias in the method. In this, our work helps underpin the conclusion of Capron et al. (2021), who previously suggested that "it may be elusive to search for a single sequence of events". The Bayesian ramp fitting method considered in this study remains a powerful tool for investigating individual abrupt transitions, as the bias that we find is small relative to the uncertainty of individual events. When calculating leads and lags from a sample of events, however, the bias becomes large relative to the uncertainty and so is likely to lead to false conclusions. It therefore appears impossible to reliably determine the temporal phasing when dealing with decadal-scale time lags such as

those that have been suggested for DO warming events. This does not necessarily exclude the careful application of this method to ascertain the phasing of different climate elements or proxies where the duration of the transition is longer, for example DO cooling events. However, further careful research would be required into any alternative applications to longer duration climate transitions.

*Code and data availability.* The CCSM4 model data are available on request from https://www.bas.ac.uk/project/sdoo/#data (last access: 22
August 2023).

The NGRIP ice-core data are available from https://doi.org/10.1594/PANGAEA.935838 (last access: 22 August 2023).

The code for the ramp fitting method as well as that used to produce the figures in this manuscript are available at https://github.com/johatt11/DO_Tempo (last access: 05 August 2024).

## Appendix A:  Ramp Fitting Method

To test whether ice-core data show significant slopes in the stadial periods preceding DO events, we consider the data segments used by Erhardt et al. (2019). We only consider data up to the fifth percentile of the posterior distribution for the transition onset time, as given by Erhardt et al. (2019), to ensure that the abrupt transition itself does not influence our results. We also only consider the sixteen DO events where Erhardt et al. (2019) provide onset times for all four proxies. We calculate linear slopes and test the significance of these using phase-randomised Fourier surrogates, which preserve the autocorrelation structure of
the data. We find that seven of the sixteen events show a significant slope ($p<0.05$) in $Ca^{2+}$ that is in the same direction as the following abrupt transition (Figure A1), as well as four for the Annual Layer Thickness and one for $Na^+$. This demonstrates the need to consider the impact of pre-transition slopes on the accuracy of the ramp fitting method, and to test whether this can be improved by directly incorporating said slopes into the transition model. This is in addition to the more obvious need to consider post-transition slopes due to the classic "saw-tooth" shape of DO events.

The deterministic component of the transition model used by Erhardt et al. (2019) to fit a series of observations $x$ taken at times $t$ is as follows:

$$\hat{x}_i(t_i) = \begin{cases} x_0 & \text{for} \quad t_i \leq t_0 \\ x_0 + \frac{x_1 - x_0}{t_1 - t_0}(t_i - t_0) & \text{for} \quad t_0 < t_i < t_1 \\ x_1 & \text{for} \quad t_i \geq t_1. \end{cases} \tag{A1}$$

The four parameters $x_0$, $\Delta x$, $t_0$, and $\Delta t = t_1 - t_0$ are the initial value, the magnitude of the transition, the time at which the transition starts, and the duration of the transition. There are two further parameters, $\sigma$ and $\tau$, that govern the AR(1) noise
which is added to the deterministic ramp in order to represent climate variability. This gives a total of six parameters. The AR1

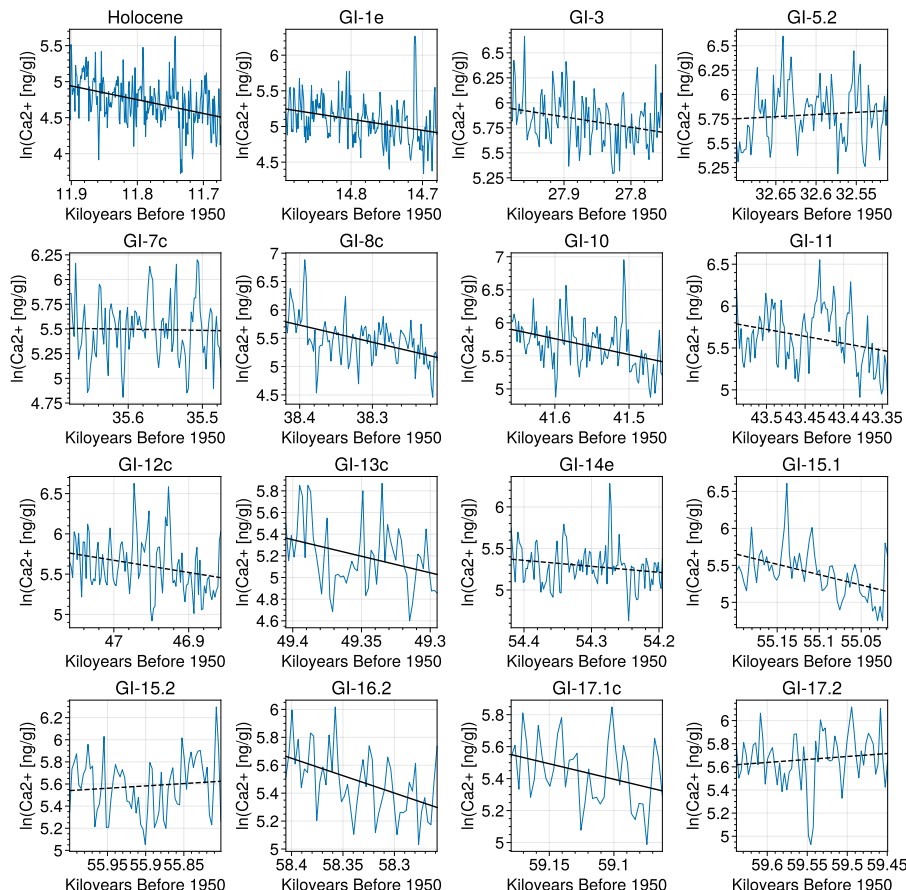

**Figure A1.** Slopes in the pre-transition stadial for NGRIP $Ca^{2+}$. Solid black lines indicate statistically significant downwards slopes (p<0.05 from a one-tailed hypothesis test using phase-randomised Fourier surrogates) whilst dashed black lines show non-significant slopes.

noise process is governed by the equations:

$$\epsilon_{i+1} = \alpha_i \epsilon_i + \sigma_{eff,i} \zeta_{i+1} \tag{A2}$$

$$\sigma_{eff,i} = \sigma \sqrt{1 - \alpha_i^2} \tag{A3}$$

$$\alpha_i = \exp\left(-\frac{t_{i+1} - t_i}{\tau}\right), \tag{A4}$$

where $\zeta_i$ is a normally distributed random variable with unit variance. The variance of this AR1 noise process is given by $\sigma^2$ and the autocorrelation time by $\tau$. If the time step $t_{i+1} - t_i$ is constant then $\alpha$ is also constant, but we allow for the possibility of unevenly sampled data where $t_{i+1} - t_i$ varies.

Given the transition model and the transition data, Bayes' theorem can be applied to infer posterior probabilities for the model parameters:

$$p(\theta|\mathcal{D}) = \frac{p(\mathcal{D}|\theta)p(\theta)}{p(\mathcal{D})}, \tag{A5}$$

where the probability, given the parameters $\theta$, that the model exactly reproduces the data $p(\mathcal{D}|\theta)$ is named the likelihood. The prior distribution $p(\theta)$ of the model parameters encodes any a-priori knowledge on the parameters, for example required positivity. The marginal probability to observe the data $p(\mathcal{D})$ is in this context nothing but a normalization constant which in practice is not relevant. We somewhat alter the priors used by Erhardt et al. (2019) in order to ensure that we can apply the ramp fitting method to different variables across a wide range of magnitudes and units. First, we calculate initial guesses for the six parameters based on simple heuristics and shift the time-series such that our initial guess for the transition onset time $t_0$ lies at time $t = 0$. The priors used are then as follows:

$$p(t_0) = \mathcal{N}(0, 50^2) \tag{A6}$$

$$p(\Delta t) = \text{Gamma}(2.0, 0.02) \tag{A7}$$

$$p(x_0) = 1 \tag{A8}$$

$$p(\Delta x) = 1 \tag{A9}$$

$$p(\sigma) = 1 \text{ for } 0 < \sigma \leq |\Delta y| \tag{A10}$$

$$p(\tau) = \text{Gamma}(1.5, 0.05), \tag{A11}$$

where the Normal distribution $\mathcal{N}$ and Gamma distribution used here are defined as:

$$\mathcal{N}(\mu, \sigma^2) = \frac{1}{\sqrt{2\pi}\sigma} \exp\left(-\frac{1}{2\sigma^2}(x - \mu)^2\right) \tag{A12}$$

$$\text{Gamma}(\alpha, \beta) = \frac{\beta^\alpha}{\Gamma(\alpha)} x^{\alpha-1} e^{-\beta x}. \tag{A13}$$

The prior probability for the transition onset time is a normal distribution centred around our initial guess. The use of the Gamma distribution as a prior for $\Delta t$ and $\tau$ ensures that these parameters are always positive, as we require. The prior for $\sigma$ also achieves this, whilst giving a uniform probability for values up to our initial guess of $|\Delta x|$. This limit is set to ensure that the noise does not dominate over the underlying transition. We use uniform distributions for $x_0$ and $\Delta x$ to ensure that the method can be applied to any possible transition, no matter the units or magnitude.

Having a model for the underlying transition, a noise process to represent short-term climate variability, and a set of prior probabilities, the final quantity we need to produce our posterior distributions is a likelihood function. This measures the likelihood of observing the data given a particular choice of model parameters. To do so, we decompose the observed data $x_i$ into a deterministic component $\hat{x}_i$ and a noise component $\epsilon_i$, $x_i = \hat{x}_i + \epsilon_i$. The likelihood function is then:

$$p(\epsilon_{i+1}|\epsilon_i, t_{i+1}, t_i, \sigma, \tau) = \mathcal{N}\left(\epsilon_i \exp\left(-\frac{t_{i+1} - t_i}{\tau}\right), \sigma_{eff,i}^2\right) \tag{A14}$$

Together with the prior distributions, the likelihood function defines the posterior distribution $p(\theta|\mathcal{D})$ up to a constant. Since this distribution is six dimensional, it is computationally very expensive to perform any subsequent analysis. To circumvent this, an MCMC-algorithm is used to sample a representative set from $p(\theta|\mathcal{D})$. All results presented in this paper are based on the application of the computations to corresponding representative sets.

This initial formulation of the ramp fitting method is very successful when applied to transitions that do not show a large gradient either before or after the transition, as is the case for most of the ice-core data for which this method was originally developed. However, some model transitions show a much more exaggerated "saw-tooth" shape, with a strongly negative gradient following the abrupt warming event. In these cases the original formulation of the method performs poorly, or sometimes fails entirely.

We need to ensure that we can successfully characterise the full range of transitions within model simulations. To this end, we extend the original Erhardt method to include gradients before and after the transition with the addition of two new parameters, taking the total to eight. These are $slope_{GS}$, the gradient in the stadial prior to the DO event, and $slope_{GIS}$, the gradient in the following interstadial. This allows us to more accurately capture the shape of the transitions. Rather than being the initial value of the observations, $y_0$ is now the observed value at the start of the transition. The other parameters retain their original meanings, and the noise process is unchanged. With this extension, the deterministic component of the model is now described by the following equation:

$$\hat{x}_i(t_i) = \begin{cases} x_0 - slope_{GS}(t_0 - t_i), & \text{if } t_i \leq t_0 \\ x_0 + \Delta x \frac{t_i - t_0}{\Delta t}, & \text{if } t_0 < t_i < t_1 \\ x_0 + \Delta x + slope_{GIS}(t_i - t_1), & \text{if } t_i \geq t_1 \end{cases} \tag{A15}$$

Our initial guesses for these two new parameters are zero, and the prior probabilities, which are identical, are chosen to avoid gradients in the stadial or interstadial which are greater than those during the transition itself.

$$p(slope_{GS}) = p(slope_{GIS}) = \mathcal{N}\left(0.0, \frac{|\Delta x|}{10\Delta t}\right) \tag{A16}$$

The prior probabilities for the other six parameters remain unchanged, as does the likelihood function.

We assess the performance of both the original and extended implementations of the ramp fitting method using decadal-resolution synthetic transitions generated as described in Section 3.1 for different values of Noise / Signal, Greenland Stadial Slope, and Greenland Interstadial Slope. Our performance metric is the uncertainty of the transition onset time, which we quantify as the width of the 5–95% credible interval. We take an event-mean across 100 synthetic transitions for each set of parameters to reduce the impact of random variability.

For cases of small slopes and large Noise / Signal, for example comparing the lower right corners of panels (b) and (d) in Figure A2, we find that the original method outperforms our extended method in that it has lower uncertainty in the onset time. This is unsurprising, as in these cases we have essentially introduced two unnecessary parameters. However, for cases where meaningful slopes are present our extended method outperforms the original method.

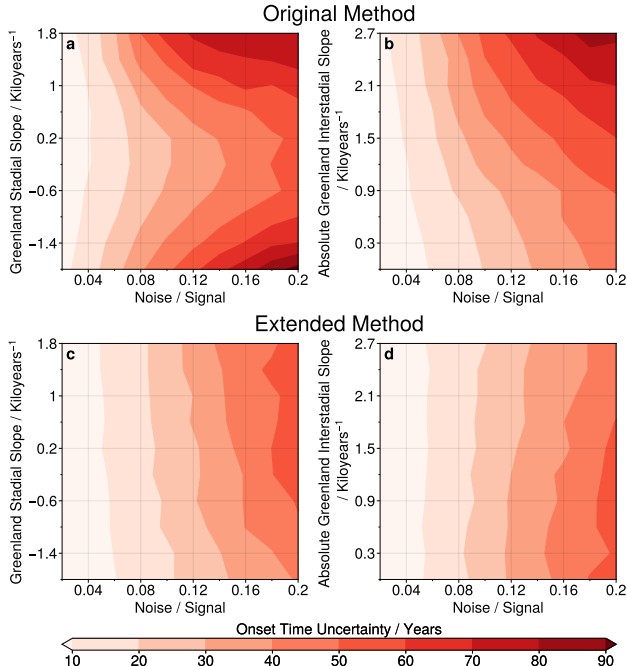

**Figure A2.** The onset time uncertainty, as measured by the width of the 5–95% credible interval, for different combinations of noise and slope parameters. We consider both the original and extended implementations of the ramp fitting method as applied to synthetic transitions with decadal resolution. Each panel contains 100 unique sets of parameter values, and for each set we take the event-mean over 100 synthetic transitions. When not explicitly varied, parameters are fixed at standard values as given in Section 3.1.

We also test the sensitivity to the choice of search window for the two implementations of the ramp fitting method. Choosing an AMOC transition from the CCSM4 model, we calculate the posterior-mean onset time using 25 different search windows for both the original and extended implementations. Figure A3 shows that the spread of these posterior-mean onset times is much lower in our extended method than in the original method, with the standard deviation reduced from 24.7 to 6.4 years. This indicates that, at least for a case where noticeable slopes are present before or after the transition, our extended method is less sensitive to the choice of search window.

Finally, we test the accuracy of the posterior-mean estimates from our extended method of the Autocorrelation Time, Greenland Stadial Slope, Greenland Interstadial Slope, and Transition Duration under very low levels of noise. Varying only one parameter at a time, we find largely accurate estimates of these parameters (Figure A4). The exceptions are the autocorrelation time, which is consistently slightly overestimated, and the stadial slope, where our extended method leads to an overestimate for cases in which the true stadial slope is strongly negative.

In addition to the original and extended implementations which are the focus of this work, we also consider an implementation of the ramp fitting method using an alternative set of prior-probabilities similar to those employed by Capron et al. (2021). The key difference here is that the prior probabilities for the transition onset time $t_0$ and the transition duration $dt$ are uniform.

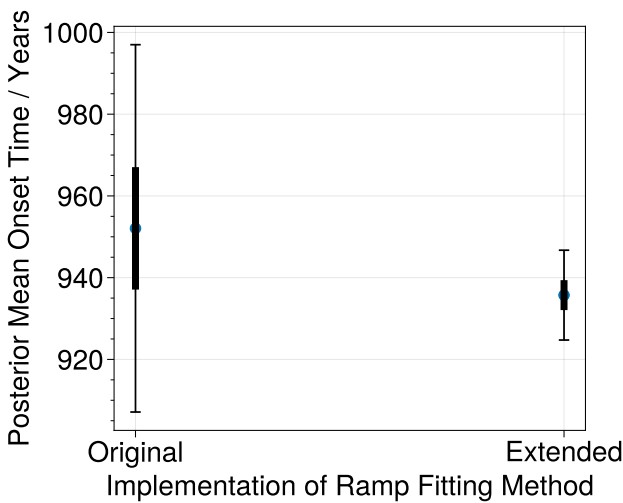

**Figure A3.** The posterior mean onset times for an AMOC transition in the CCSM4 model in both the original and extended implementations of the ramp fitting method when using 25 different search windows. The extended implementation shows a reduced spread in the posterior-mean onset times, indicating reduced sensitivity to the choice of search window.

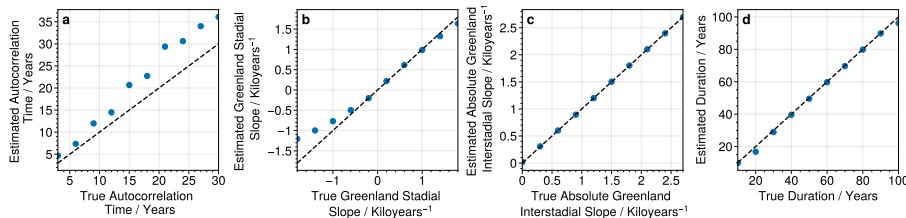

**Figure A4.** Low-noise benchmarks of our extended ramp-fitting method. Varying only one parameter at a time, we plot true values against estimated values for (a) Autocorrelation Time, (b) Greenland Stadial Slope, (c) Greenland Interstadial Slope, and (d) Transition Duration. The Noise / Signal ratio is set to 0.01, and when not explicitly varied the other parameters are fixed at their standard values as given in Section 3.1. Each data point is given by the event-mean across 100 synthetic transitions of the posterior-mean for that parameter.

This implementation includes the slopes $slope_{GS}$ and $slope_{GIS}$ discussed above, and aside from the uniform priors there are no further differences.

## Appendix B: Hypothesis Test

After the bias correction, the event-averaged lead of the precipitation's transition is approximately symmetrically uncertainty-distributed around $-10.0$ years, with the 95% confidence interval reaching from $-3.9$ to $-16.5$ years.

In order to assess significance of this result, we test the following competing hypotheses:

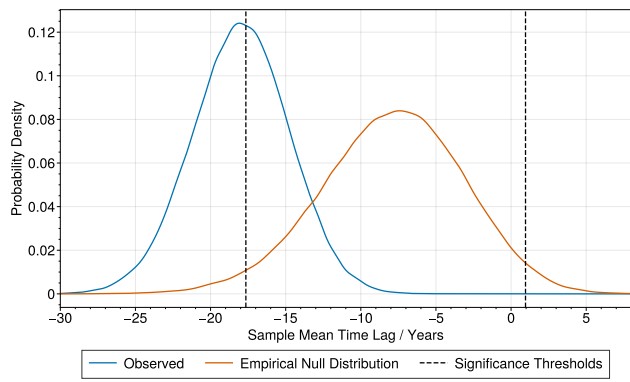

**Figure B1.** The observed and empirical null distributions for the sample mean time lag between precipitation and temperature in the CCSM4 model. The observed distribution here is identical to the uncorrected distribution in Figure 4. 5% of the empirical null distribution lies outside of the significance thresholds shown as dashed lines.

- $H_0$: The population mean time lag for precipitation relative to temperature is equal to zero. The observed (uncorrected) precipitation lead is a spurious result induced by the method.

- $H_1$: The population mean time lag for precipitation relative to temperature is not equal to zero.

To test these hypothesis against each other, we design an empirical null distribution. Our null distribution will reflect the plausibility that the ramp fit assigns to a certain event-averaged (over 19 events) temporal lag $\langle\Delta t\rangle_{19}$ given that the null hypothesis is true. (The randomness associated with the individual time series involved in the computation of $\langle\Delta t\rangle_{19}$ is integrated out in our null hypothesis). We then compare the uncertainty distribution obtained from the assessment of the CCSM4 data with this null distribution. In a slight abuse of terminology, we compute a corresponding p-value as the probability that any $\langle\Delta t\rangle_{19}^{\mathrm{obs}}$ sampled from the observed uncertainty distribution is closer to the mean of the null distribution $\mu_0$ than a second one $\langle\Delta t\rangle_{19}^{\mathrm{null}}$ sampled from the null distribution.

$$p = P\Big(\big|\langle\Delta t\rangle_{19}^{\mathrm{obs}} - \mu_{\mathrm{null}}\big| < \big|\langle\Delta t\rangle_{19}^{\mathrm{null}} - \mu_{\mathrm{null}}\big|\Big) = 0.078, \tag{B1}$$

To construct this distribution, we randomly sample 19 pairs of time series from the "analogous" synthetic transitions for precipitation and temperature. Application of the ramp fit to the individual pairs yields 19 uncertain time lags. From each of these 19 uncertainty distributions we sample a single value and subsequently perform the event average. 100,000-fold repetition of this procedure yields a set of time lags averaged over 19 events that reflects the randomness in the 19-fold observation of the random experiment *DO event* and the uncertainty in the timing of each transition onset.

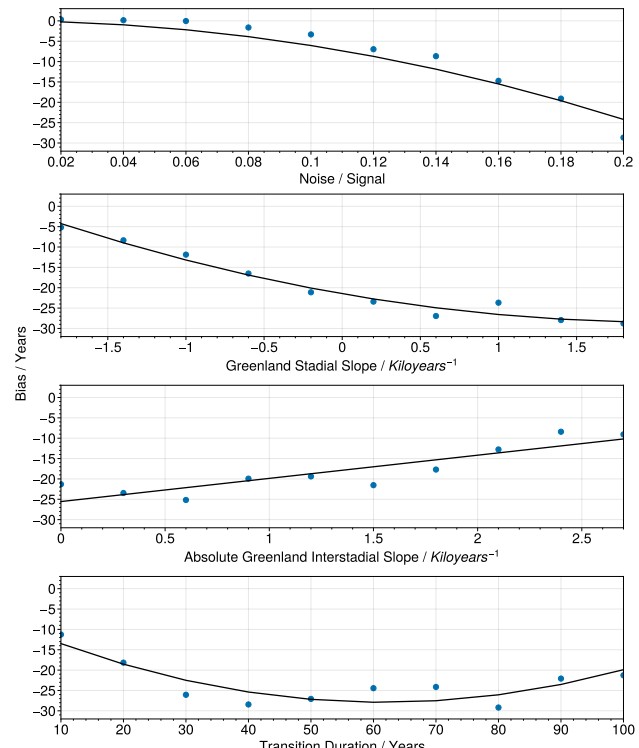

**Figure C1.** The dependence of the transition onset time bias on four transition parameters when using our extended ramp fitting method with uniform priors similar to those adopted by Capron et al. (2021), for transitions with decadal resolution. For further details see Appendix A. When not explicitly varied, the Noise / Signal ratio is fixed at 0.2, and the other parameters are fixed at standard values as given in Section 3.1.The bias is greater than when using our standard priors (Figure 3).

## Appendix C:  Supplementary Figures

## Appendix D:  Transition Parameters for Bias Estimation

We present here the minimum, event-mean, and maximum values, across the sample of DO events, of the posterior-mean transition parameters for all of the CCSM4 variables (Table D1) and NGRIP proxies (Table D2) considered in this study, as given by the extended ramp fitting method. These values are used to generate the "analogous" synthetic transitions which we use to estimate the bias affecting each variable or proxy, with one exception for NGRIP as follows: We convert each posterior mean autocorrelation time into an autocorrelation parameter $\alpha$, and then convert the event-mean of $\alpha$ back into an

autocorrelation time using the appropriate time resolution. We do this because we find that the autocorrelation time is correlated with the temporal resolution whereas $\alpha$ is not, and so it is more appropriate to take our mean over $\alpha$.

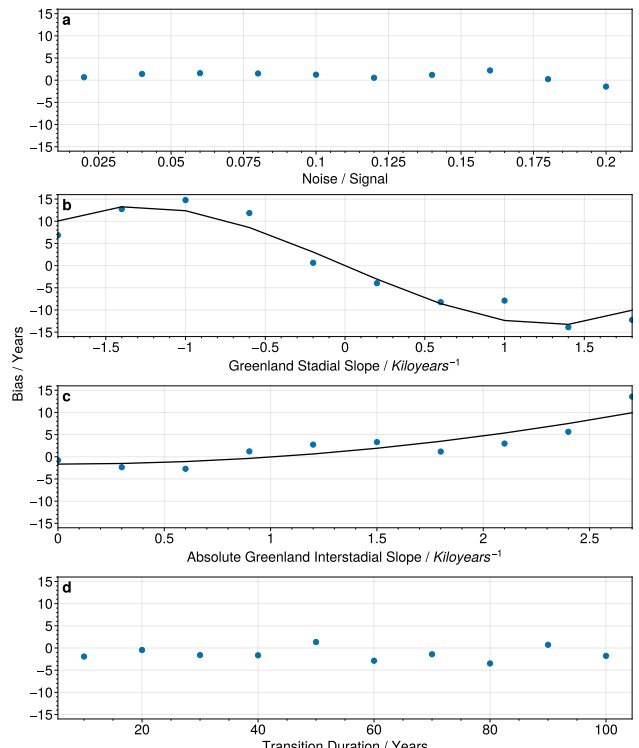

**Figure C2.** The dependence of the bias on four transition parameters when using a simple least-squares method instead of the Bayesian fitting method, for transitions with decadal resolution. When not explicitly varied, the Noise / Signal ratio is fixed at 0.2, and the other parameters are fixed at standard values as given in Section 3.1. No trend is plotted for the noise / signal ratio or for the duration, as the bias does not significantly depend on these parameters. However, there is a strong, non-linear dependence of the bias on the slopes.

For the CCSM4 model data, all of the slopes during the interstadials following the transition are negative, and so we give the absolute values as in Figure 3. However, for the NGRIP data there are a small number of cases where this slope is in fact positive, and so for NGRIP we do not give the absolute values.

The minima and maxima of the different transitions parameters are used to select plausible parameter ranges over which to perform the systematic testing in Section 3.1.

*Author contributions.* JS and LCS designed the study in consultation with KR. JS conducted the analysis under the guidance of KR. JS wrote the first draft of the manuscript. All authors contributed to the interpretation of the results and to the final manuscript draft.

*Competing interests.* The authors declare that they have no conflict of interest.

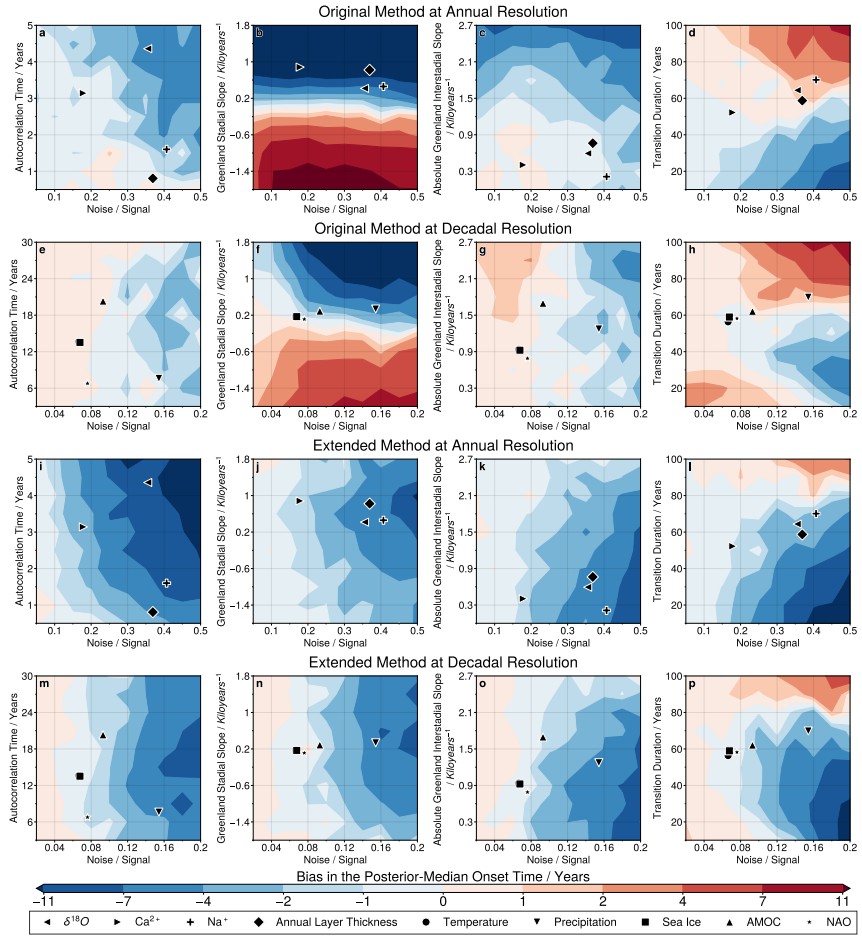

**Figure C3.** The bias in the posterior-median transition onset time, calculated following the same approach as for the posterior-mean in Figure 3. Considering the posterior-median onset time instead of the posterior-mean leads to slightly more positive values of the bias, but the difference is small.

*Disclaimer.* TEXT

*Acknowledgements.* JS is funded by a C-CLEAR NERC DTP studentship. LCS and KR acknowledge funding and much inspiration from the EU-H2020: Tipping Points in the Earths System: TiPES program, under grant number 820970. This is TiPES output #261. FM acknowledges funding from a Natural Environment Research Council (NERC) Discovery Science Grant (NE/W006243/1).

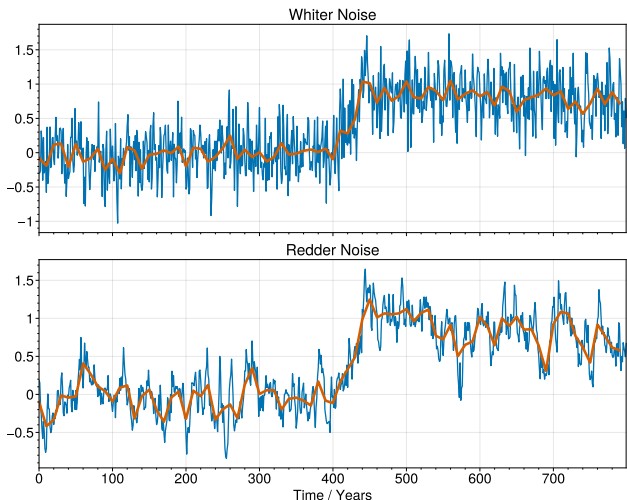

**Figure C4.** Examples of how we down-sample annual resolution synthetic data to decadal resolution, for the cases of both "whiter" and "redder" noise.

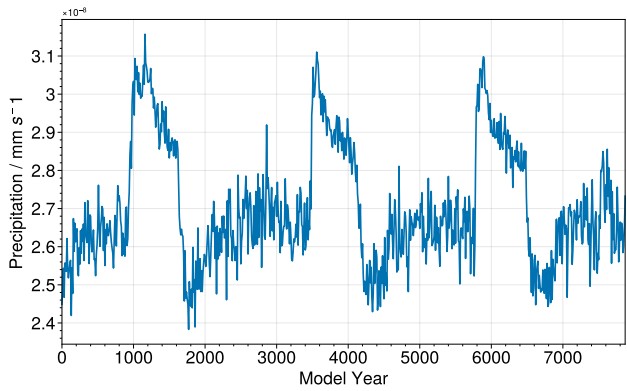

**Figure C5.** The time series for precipitation in the CCSM4 simulation with 200 ppm of atmospheric carbon dioxide, demonstrating the relatively higher level of noise during the cooler stadial periods, which are visible here as periods of lower precipitation.

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

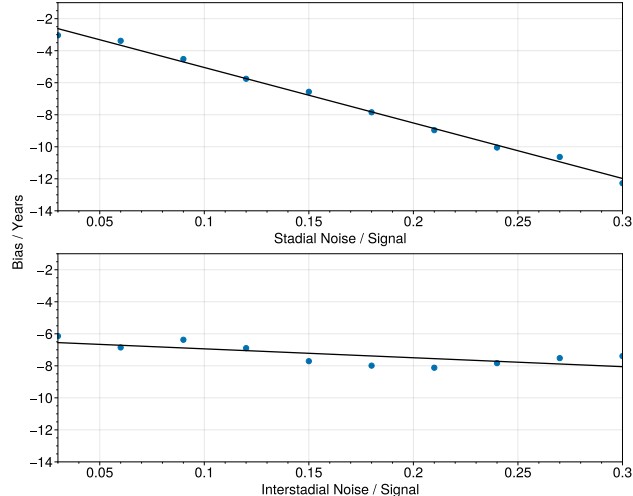

**Figure C6.** The dependence of the bias on noise / signal in both the preceding stadial and following interstadial, for the case of two independent noise regimes. The bias depends much more on the stadial noise / signal.

Buizert, C., Sigl, M., Severi, M., Markle, B. R., Wettstein, J. J., McConnell, J. R., Pedro, J. B., Sodemann, H., Goto-Azuma, K., Kawamura, K., Fujita, S., Motoyama, H., Hirabayashi, M., Uemura, R., Stenni, B., Parrenin, F., He, F., Fudge, T. J., and Steig, E. J.: Abrupt ice-age shifts in southern westerly winds and Antarctic climate forced from the north, Nature, 563, 681–685, https://doi.org/10.1038/s41586-018-0727-5, number: 7733 Publisher: Nature Publishing Group, 2018.

Capron, E., Rasmussen, S. O., Popp, T. J., Erhardt, T., Fischer, H., Landais, A., Pedro, J. B., Vettoretti, G., Grinsted, A., Gkinis, V., Vaughn, B., Svensson, A., Vinther, B. M., and White, J. W. C.: The anatomy of past abrupt warmings recorded in Greenland ice, Nature Communications, 12, 2106, https://doi.org/10.1038/s41467-021-22241-w, number: 1 Publisher: Nature Publishing Group, 2021.

Cheng, H., Sinha, A., Cruz, F. W., Wang, X., Edwards, R. L., d'Horta, F. M., Ribas, C. C., Vuille, M., Stott, L. D., and Auler, A. S.: Climate change patterns in Amazonia and biodiversity, Nature Communications, 4, 1411, https://doi.org/10.1038/ncomms2415, number: 1 Publisher: Nature Publishing Group, 2013.

Corrick, E. C., Drysdale, R. N., Hellstrom, J. C., Capron, E., Rasmussen, S. O., Zhang, X., Fleitmann, D., Couchoud, I., Wolff, E., and Monsoon, S. A.: Synchronous timing of abrupt climate changes during the last glacial period, Science, 369, 963–969, https://doi.org/10.1126/science.aay5538, 2020.

Dansgaard, W., Clausen, H. B., Gundestrup, N., Hammer, C. U., Johnsen, S. F., Kristinsdottir, P. M., and Reeh, N.: A New Greenland Deep Ice Core, Science, 218, 1273–1277, https://doi.org/10.1126/science.218.4579.1273, 1982.

Dansgaard, W., Johnsen, S. J., Clausen, H. B., Dahl-Jensen, D., Gundestrup, N. S., Hammer, C. U., Hvidberg, C. S., Steffensen, J. P., Sveinbjörnsdottir, A. E., Jouzel, J., and Bond, G.: Evidence for general instability of past climate from a 250-kyr ice-core record, Nature, 364, 218–220, https://doi.org/10.1038/364218a0, number: 6434 Publisher: Nature Publishing Group, 1993.

Dokken, T. M., Nisancioglu, K. H., Li, C., Battisti, D. S., and Kissel, C.: Dansgaard-Oeschger cycles: Interactions between ocean and sea ice intrinsic to the Nordic seas, Paleoceanography, 28, 491–502, https://doi.org/10.1002/palo.20042, 2013.

Drysdale, R. N., Zanchetta, G., Hellstrom, J. C., Fallick, A. E., McDonald, J., and Cartwright, I.: Stalagmite evidence for the precise timing of North Atlantic cold events during the early last glacial, Geology, 35, 77–80, https://doi.org/10.1130/G23161A.1, 2007.

Erhardt, T., Capron, E., Rasmussen, S. O., Schüpbach, S., Bigler, M., Adolphi, F., and Fischer, H.: Decadal-scale progression of the onset of Dansgaard–Oeschger warming events, Climate of the Past, 15, 811–825, https://doi.org/10.5194/cp-15-811-2019, publisher: Copernicus GmbH, 2019.

Fleitmann, D., Cheng, H., Badertscher, S., Edwards, R. L., Mudelsee, M., Göktürk, O. M., Fankhauser, A., Pickering, R., Raible, C. C., Matter, A., Kramers, J., and Tüysüz, O.: Timing and climatic impact of Greenland interstadials recorded in stalagmites from northern Turkey, Geophysical Research Letters, 36, https://doi.org/10.1029/2009GL040050, _eprint: https://onlinelibrary.wiley.com/doi/pdf/10.1029/2009GL040050, 2009.

Fohlmeister, J., Sekhon, N., Columbu, A., Vettoretti, G., Weitzel, N., Rehfeld, K., Veiga-Pires, C., Ben-Yami, M., Marwan, N., and Boers, N.: Global reorganization of atmospheric circulation during Dansgaard–Oeschger cycles, Proceedings of the National Academy of Sciences, 120, e2302283 120, https://doi.org/10.1073/pnas.2302283120, publisher: Proceedings of the National Academy of Sciences, 2023.

Gottschalk, J., Skinner, L. C., Misra, S., Waelbroeck, C., Menviel, L., and Timmermann, A.: Abrupt changes in the southern extent of North Atlantic Deep Water during Dansgaard–Oeschger events, Nature Geoscience, 8, 950–954, https://doi.org/10.1038/ngeo2558, number: 12 Publisher: Nature Publishing Group, 2015.

Henry, L. G., McManus, J. F., Curry, W. B., Roberts, N. L., Piotrowski, A. M., and Keigwin, L. D.: North Atlantic ocean circulation and abrupt climate change during the last glaciation, Science, 353, 470–474, https://doi.org/10.1126/science.aaf5529, publisher: American Association for the Advancement of Science, 2016.

Hoff, U., Rasmussen, T. L., Stein, R., Ezat, M. M., and Fahl, K.: Sea ice and millennial-scale climate variability in the Nordic seas 90 kyr ago to present, Nature Communications, 7, 12 247, https://doi.org/10.1038/ncomms12247, number: 1 Publisher: Nature Publishing Group, 2016.

Johnsen, S. J., Clausen, H. B., Dansgaard, W., Fuhrer, K., Gundestrup, N., Hammer, C. U., Iversen, P., Jouzel, J., Stauffer, B., and Steffensen, J. P.: Irregular glacial interstadials recorded in a new Greenland ice core, Nature, 359, 311–313, https://doi.org/10.1038/359311a0, number: 6393 Publisher: Nature Publishing Group, 1992.

Kanner, L. C., Burns, S. J., Cheng, H., and Edwards, R. L.: High-Latitude Forcing of the South American Summer Monsoon During the Last Glacial, Science, 335, 570–573, https://doi.org/10.1126/science.1213397, publisher: American Association for the Advancement of Science, 2012.

Kindler, P., Guillevic, M., Baumgartner, M., Schwander, J., Landais, A., and Leuenberger, M.: Temperature reconstruction from 10 to 120 kyr b2k from the NGRIP ice core, Climate of the Past, 10, 887–902, https://doi.org/10.5194/cp-10-887-2014, publisher: Copernicus GmbH, 2014.

Klockmann, M., Mikolajewicz, U., Kleppin, H., and Marotzke, J.: Coupling of the Subpolar Gyre and the Overturning Circulation During Abrupt Glacial Climate Transitions, Geophysical Research Letters, 47, e2020GL090 361, https://doi.org/10.1029/2020GL090361, 2020.

Kuniyoshi, Y., Abe-Ouchi, A., Sherriff-Tadano, S., Chan, W.-L., and Saito, F.: Effect of Climatic Precession on Dansgaard-Oeschger-Like Oscillations, Geophysical Research Letters, 49, e2021GL095 695, https://doi.org/10.1029/2021GL095695, 2022.

Li, C. and Born, A.: Coupled atmosphere-ice-ocean dynamics in Dansgaard-Oeschger events, Quaternary Science Reviews, 203, 1–20, https://doi.org/10.1016/j.quascirev.2018.10.031, 2019.

Li, C., Battisti, D. S., Schrag, D. P., and Tziperman, E.: Abrupt climate shifts in Greenland due to displacements of the sea ice edge, Geophysical Research Letters, 32, https://doi.org/10.1029/2005GL023492, _eprint: https://onlinelibrary.wiley.com/doi/pdf/10.1029/2005GL023492, 2005.

Li, C., Battisti, D. S., and Bitz, C. M.: Can North Atlantic Sea Ice Anomalies Account for Dansgaard–Oeschger Climate Signals?, Journal of Climate, 23, 5457–5475, https://doi.org/10.1175/2010JCLI3409.1, publisher: American Meteorological Society Section: Journal of Climate, 2010.

Li, T.-Y., Han, L.-Y., Cheng, H., Edwards, R. L., Shen, C.-C., Li, H.-C., Li, J.-Y., Huang, C.-X., Zhang, T.-T., and Zhao, X.: Evolution of the Asian summer monsoon during Dansgaard/Oeschger events 13–17 recorded in a stalagmite constrained by high-precision chronology from southwest China, Quaternary Research, 88, 121–128, https://doi.org/10.1017/qua.2017.22, publisher: Cambridge University Press, 2017.

Lohmann, J. and Ditlevsen, P. D.: Objective extraction and analysis of statistical features of Dansgaard–Oeschger events, Climate of the Past, 15, 1771–1792, https://doi.org/10.5194/cp-15-1771-2019, publisher: Copernicus GmbH, 2019.

Lynch-Stieglitz, J.: The Atlantic Meridional Overturning Circulation and Abrupt Climate Change, Annual Review of Marine Science, 9, 83–104, https://doi.org/10.1146/annurev-marine-010816-060415, 2017.

Maffezzoli, N., Vallelonga, P., Edwards, R., Saiz-Lopez, A., Turetta, C., Kjær, H. A., Barbante, C., Vinther, B., and Spolaor, A.: A 120 000-year record of sea ice in the North Atlantic?, Climate of the Past, 15, 2031–2051, https://doi.org/10.5194/cp-15-2031-2019, publisher: Copernicus GmbH, 2019.

Malmierca-Vallet, I., Sime, L. C., and the D–O community members: Dansgaard–Oeschger events in climate models: review and baseline Marine Isotope Stage 3 (MIS3) protocol, Climate of the Past, 19, 915–942, https://doi.org/10.5194/cp-19-915-2023, publisher: Copernicus GmbH, 2023.

Markle, B. R., Steig, E. J., Buizert, C., Schoenemann, S. W., Bitz, C. M., Fudge, T. J., Pedro, J. B., Ding, Q., Jones, T. R., White, J. W. C., and Sowers, T.: Global atmospheric teleconnections during Dansgaard–Oeschger events, Nature Geoscience, 10, 36–40, https://doi.org/10.1038/ngeo2848, number: 1 Publisher: Nature Publishing Group, 2017.

Moseley, G. E., Spötl, C., Svensson, A., Cheng, H., Brandstätter, S., and Edwards, R. L.: Multi-speleothem record reveals tightly coupled climate between central Europe and Greenland during Marine Isotope Stage 3, Geology, 42, 1043–1046, https://doi.org/10.1130/G36063.1, 2014.

NGRIP members: High-resolution record of Northern Hemisphere climate extending into the last interglacial period, Nature, 431, 147–151, https://doi.org/10.1038/nature02805, number: 7005 Publisher: Nature Publishing Group, 2004.

Riechers, K. and Boers, N.: Significance of uncertain phasing between the onsets of stadial–interstadial transitions in different Greenland ice core proxies, Climate of the Past, 17, 1751–1775, https://doi.org/10.5194/cp-17-1751-2021, publisher: Copernicus GmbH, 2021.

Sadatzki, H., Maffezzoli, N., Dokken, T. M., Simon, M. H., Berben, S. M. P., Fahl, K., Kjær, H. A., Spolaor, A., Stein, R., Vallelonga, P., Vinther, B. M., and Jansen, E.: Rapid reductions and millennial-scale variability in Nordic Seas sea ice cover during abrupt glacial climate changes, Proceedings of the National Academy of Sciences, 117, 29 478–29 486, https://doi.org/10.1073/pnas.2005849117, publisher: Proceedings of the National Academy of Sciences, 2020.

Schneider, T., Bischoff, T., and Haug, G. H.: Migrations and dynamics of the intertropical convergence zone, Nature, 513, 45–53, https://doi.org/10.1038/nature13636, number: 7516 Publisher: Nature Publishing Group, 2014.

Schüpbach, S., Fischer, H., Bigler, M., Erhardt, T., Gfeller, G., Leuenberger, D., Mini, O., Mulvaney, R., Abram, N. J., Fleet, L., Frey, M. M., Thomas, E., Svensson, A., Dahl-Jensen, D., Kettner, E., Kjaer, H., Seierstad, I., Steffensen, J. P., Rasmussen, S. O., Vallelonga,

P., Winstrup, M., Wegner, A., Twarloh, B., Wolff, K., Schmidt, K., Goto-Azuma, K., Kuramoto, T., Hirabayashi, M., Uetake, J., Zheng, J., Bourgeois, J., Fisher, D., Zhiheng, D., Xiao, C., Legrand, M., Spolaor, A., Gabrieli, J., Barbante, C., Kang, J.-H., Hur, S. D., Hong, S. B., Hwang, H. J., Hong, S., Hansson, M., Iizuka, Y., Oyabu, I., Muscheler, R., Adolphi, F., Maselli, O., McConnell, J., and Wolff, E. W.: Greenland records of aerosol source and atmospheric lifetime changes from the Eemian to the Holocene, Nature Communications, 9, 1476, https://doi.org/10.1038/s41467-018-03924-3, number: 1 Publisher: Nature Publishing Group, 2018.

Sime, L. C., Hopcroft, P. O., and Rhodes, R. H.: Impact of abrupt sea ice loss on Greenland water isotopes during the last glacial period, Proceedings of the National Academy of Sciences, 116, 4099–4104, https://doi.org/10.1073/pnas.1807261116, publisher: Proceedings of the National Academy of Sciences, 2019.

Steffensen, J. P., Andersen, K. K., Bigler, M., Clausen, H. B., Dahl-Jensen, D., Fischer, H., Goto-Azuma, K., Hansson, M., Johnsen, S. J., Jouzel, J., Masson-Delmotte, V., Popp, T., Rasmussen, S. O., Röthlisberger, R., Ruth, U., Stauffer, B., Siggaard-Andersen, M.-L., Svein-björnsdóttir, A. E., Svensson, A., and White, J. W. C.: High-Resolution Greenland Ice Core Data Show Abrupt Climate Change Happens in Few Years, Science, 321, 680–684, https://doi.org/10.1126/science.1157707, publisher: American Association for the Advancement of Science, 2008.

Vettoretti, G. and Peltier, W. R.: Thermohaline instability and the formation of glacial North Atlantic super polynyas at the onset of Dansgaard-Oeschger warming events, Geophysical Research Letters, 43, 5336–5344, https://doi.org/10.1002/2016GL068891, 2016.

Vettoretti, G., Ditlevsen, P., Jochum, M., and Rasmussen, S. O.: Atmospheric CO2 control of spontaneous millennial-scale ice age climate oscillations, Nature Geoscience, 15, 300–306, https://doi.org/10.1038/s41561-022-00920-7, number: 4 Publisher: Nature Publishing Group, 2022.

Wang, X., Auler, A. S., Edwards, R. L., Cheng, H., Cristalli, P. S., Smart, P. L., Richards, D. A., and Shen, C.-C.: Wet periods in northeastern Brazil over the past 210 kyr linked to distant climate anomalies, Nature, 432, 740–743, https://doi.org/10.1038/nature03067, number: 7018 Publisher: Nature Publishing Group, a.

Wang, Y., Cheng, H., Edwards, R. L., Kong, X., Shao, X., Chen, S., Wu, J., Jiang, X., Wang, X., and An, Z.: Millennial- and orbital-scale changes in the East Asian monsoon over the past 224,000 years, Nature, 451, 1090–1093, https://doi.org/10.1038/nature06692, number: 7182 Publisher: Nature Publishing Group, b.

Wang, Y. J., Cheng, H., Edwards, R. L., An, Z. S., Wu, J. Y., Shen, C.-C., and Dorale, J. A.: A High-Resolution Absolute-Dated Late Pleistocene Monsoon Record from Hulu Cave, China, Science, 294, 2345–2348, https://doi.org/10.1126/science.1064618, publisher: American Association for the Advancement of Science, c.

Zhang, X., Barker, S., Knorr, G., Lohmann, G., Drysdale, R., Sun, Y., Hodell, D., and Chen, F.: Direct astronomical influence on abrupt climate variability, Nature Geoscience, 14, 819–826, https://doi.org/10.1038/s41561-021-00846-6, number: 11 Publisher: Nature Publishing Group, 2021.

**Table D1.** The minimum, event-mean, and maximum of the posterior-mean parameter values for each variable across the sample of 19 abrupt warming events in the CCSM4 model simulations. The mean values are used to construct the synthetic transitions with which we estimate bias, as described in Section 3.3.

| CCSM4 Variable | Autocorrelation Time / Years | Noise / Signal | Duration / Years | Stadial Slope / Kiloyears$^{-1}$ | Absolute Interstadial Slope / Kiloyears$^{-1}$ |
|---|---|---|---|---|---|
| Temperature | 2.54, 13.65, 34.02 | 0.045, 0.066, 0.110 | 13.6, 56.3, 91.7 | -0.316, 0.164, 0.694 | 0.211, 0.940, 1.512 |
| Precipitation | 3.42, 7.61, 13.42 | 0.124, 0.154, 0.190 | 43.1, 69.9, 125.1 | -0.552, 0.335, 1.090 | 0.026, 1.273, 1.973 |
| Sea Ice | 2.23, 13.52, 36.55 | 0.047, 0.068, 0.094 | 24.4, 59.0, 93.7 | -0.350, 0.168, 0.580 | 0.068, 0.924, 1.564 |
| AMOC | 4.08, 20.32, 55.94 | 0.065, 0.093, 0.156 | 23.6, 62.1, 84.5 | -0.205, 0.289, 0.780 | 0.456, 1.697, 2.688 |
| NAO | 2.28, 6.77, 11.98 | 0.063, 0.076, 0.103 | 34.6, 58.2, 83.6 | -0.355, 0.112, 0.379 | 0.447, 0.789, 1.362 |

**Table D2.** The minimum, event-mean, and maximum of the posterior-mean parameter values for each proxy across the sample of 16 DO events in the NGRIP ice-core record. The mean values are used to construct synthetic transitions with which we estimate bias, as described in Section 3.3.

| NGRIP Proxy | Autocorrelation Time / Years | Noise / Signal | Duration / Years | Stadial Slope / Kiloyears$^{-1}$ | Interstadial Slope / Kiloyears$^{-1}$ |
|---|---|---|---|---|---|
| Na$^+$ | 0.77, 1.78, 3.60 | 0.275, 0.407, 0.640 | 28.0, 70.1, 125.6 | -1.299, 0.459, 1.977 | -1.622, -0.212, 1.578 |
| Annual Layer Thickness | 0.40, 0.85, 1.98 | 0.271, 0.369, 0.543 | 32.8, 58.7, 115.1 | -0.552, 0.822, 3.152 | -1.992, -0.762, 0.822 |
| Ca$^{2+}$ | 1.48, 3.81, 8.96 | 0.125, 0.179, 0.229 | 31.3, 52.2, 109.1 | 0.025, 0.883, 1.988 | -1.858, -0.405, 0.805 |
| $\delta^{18}$O | 2.19, 5.66, 13.13 | 0.267, 0.355, 0.452 | 35.3, 64.4, 91.3 | -0.674, 0.416, 2.973 | -2.194, -0.598, 0.545 |