# Peer review of "Estimating Biases During Detection of Leads and Lags Between Climate Elements Across Dansgaard–Oeschger Events"

_EGUsphere, 2023_

## Referee Comment (RC1)

**Review of The Temporal Phasing of Rapid Dansgaard–Oeschger Warming Events Cannot Be Reliably Determined**

The manuscript studies the time lags between climate parameters during transition in rapid warming events using Bayesian ramp fitting, and the associated uncertainties that the methods imply. They create analogue simulations of such transitions as represented in General Circulation Models (GCM) and proxy records (namely the North Grip ice core), and evaluate biases that the ramp fitting can induce in estimating the lags between the climatic parameters.

The study relies on sound statistical approaches that provide the authors with estimate of such biases of the same order of magnitude than the expected lags, suggesting that at the present stage, it might be difficult to evaluate what parameters describing the past climatic conditions changed first during a given DO event.

The manuscript is relatively well written and though it is quite technical on a possibly niche topic, I believe it fits really well within the scope of Climate of the Past. Some improvement on the structure could be beneficial to highlight the strength of the manuscript, but beyond that, I recommend the manuscript be published after minor corrections.

**General comments:**

The approaches described here, i.e. studying the impact of the Bayesian ramp fitting on the uncertainty of the estimation of lags between climatic parameters simulated by GCM or reconstructed from ice core records, is usually a discussion topic for paleoclimate studies. I think it is a perfectly valid study, and do not question the relevance of the publication, but highlight this point because I would say that the *results* of your study correspond to what would be the *discussion* of a paleo reconstruction of a DO event lag study: indeed, classical paleo papers would estimate the lags and in the discussion attempt to provide uncertainties or biases of their estimates. Here, I found the structure of the result section a bit confusing and would suggest the authors to reorganise it a bit keeping in mind what the results are. Specifically, to me, the actual results come in section 3.3., while sections 3.1 and 3.2 detail sensitivity tests on the parameters that were used for the synthetic transitions and the resolution.

Overall, there are implicit equivalences that are made between the study of the outputs of CCSM4 and the parameters reconstructed from the NGRIP ice core which might be not accurate. In Section 2.2, a list of what climatic parameters are usually associated with the proxies retrieved from the ice core record is provided, but it is mentioned that the one to one equivalence is not totally correct, for instance, $\delta^{18}O$ is influenced by both temperature and sea ice extent changes (Sime et al., 2019). Then, the way that Table 3 and Figures 4 and 5 are represented suggests that some parameters are equivalent with the way that they are represented either on the same line or with the same colour.

Finally, I'm a bit confused with the choice of the reference parameter for the lag estimate for CCSM4 and the NGRIP ice core. Indeed, I don't understand why in CCSM4 the lag is calculated against temperature, while NGRIP ice core, it is calculated against the calcium, which is referenced to be a proxy of the NAO. This might be something classically done here, but since amongst the four parameters used from NGRIP ice core, $\delta^{18}O$ would be the one the most closely related to temperature, it appears that the lag as presented here would have different meanings for models and paleoclimate reconstructions.

**Specific comments:**

Introduction: the introduction overall feels too detailed, to the point that I struggled to see what was the topic of the manuscript when I first read it. Line 58 says "Subsequent to this work…", suggesting that the overall timeline of the study of lags of climatic parameters for DO events are detailed, while an introduction probably just needs the most up-to-date information that is needed to understand the interest of the manuscript.

Line 44: "On the first line of evidence, Adolphi et al…"

It is not clear what line of evidence this is. The similar formulation line 53 also puzzled me.

Line 89: "which Vettoretti et al. (2022) have kindly provided."

"Kindly" is not an appropriate word to use, especially considering that most journals now require open access to the data of a published paper.

Lines 124 – 125: "For in-depth discussion of the physical interpretation of these proxies, see Section 2 of Erhardt et al. (2019) and references therein."

This could be personal style, but I believe that a manuscript should be stand-alone. The detailed discussion of the physical interpretation of the proxies is beyond the scope of this manuscript, but overall idea should be provided in the introduction or the methods.

Lines 128 – 129: "The aerosol concentrations were measured using Continuous Flow Analysis (CFA) at temporal resolution ranging from 2 years for the most recent period to 3 years for the oldest period"

Was anything measured for this manuscript? Is the fact that the data are measured via CFA relevant here? This sentence was very confusing, making me think that you measured aerosol concentrations within this manuscript. Overall, only the effective resolution of the data is important here. Considering that all this information is already provided in the table, I would remove this paragraph.

Line 138: "The addition of an AR(1) noise process makes the model probabilistic"

Where is the AR(1) noise added? Why is it not explicitly given in an equation which shows exactly what is added where?

Line 143: "This leads to improved agreement of the transition model with the analysed data (Figure 1)."

This is not clear. Is the improved agreement shown in the histogram below each figure? What do they mean? they have no label, no caption, and no explanation.

Lines 143 – 144: "Additionally, our extension of the method reduces the sensitivity of the transition timing to the search window, which is otherwise one of the drawbacks of this method (Capron et al., 2021)."

Also unclear what is meant here.

Line 157: The detailed description of each hypothesis is one line of text each. They should be included here, and not in the appendix.

Lines 160 – 161: "We conduct our analysis of the NGRIP ice core in the same manner. In this instance, we calculate time lags for the other three proxies relative to Ca"

Why calcium which you say represent the NAO (line 120), and not d18O which is closer to temperature? This would be more coherent compared to what is done with models?

Lines 227 – 229: "However, we have established that there is no unbiased means by which to estimate these transition durations, and so we could not guarantee that this would be any more accurate."

Where was this established? And shouldn't be this line in discussion rather than in Results?

Figure 5: The matching colours between Figures 4 and 5 suggest equivalency between precipitation and Sodium, sea ice and d18O, and AMOC and thickness. The colours should be different in both figures if they are not supposed to represent equivalent parameters.

---

## Referee Comment (RC2)

**Review of "The Temporal Phasing of Rapid Dansgaard–Oeschger Warming Events Cannot Be Reliably Determined" by John Slattery et al., https://doi.org/10.5194/egusphere-2023-2496**

The manuscript presents an analysis of some limitations of ramp fitting as a tool to determine the characteristics of abrupt events seen in palaeoclimate data and model runs. The study is relevant, generally well presented although (some more details could be wished for in places), and suitable for publication in Climate of the Past.

First, the ramp fitting method is extended to take pre- and post-transition trends in the data into account. This is a good move when the method is applied to model data which exhibit more pronounced trends in especially the interstadials, but see further comments below on the need for modelling a trend in the stadials. The main finding is that the ramp fitting method detects ramp start points that are generally too early when the method is tested on synthetic data, especially with high noise levels. This effect is denoted "bias", and the bias is then quantified for a range of scenarios. The effect can be anything from one year upwards, depending mainly on the noise-to-signal levels.

I have some methodological concerns and some regarding the presentation of the results, that I think should be addressed before publication:

1. The manuscript does not really discuss the magnitude of the suggested bias compared to the credible intervals produced by the ramp fitting method as published in earlier studies. If, for example, the bias is a few years in cases where the credible intervals span decades, the bias will unlikely lead to wrong conclusions. In this case, the existence of a systematic bias would still be a valid result (and to people interested in the methodology also an important one), but the analysis should consistently address whether the bias is sufficiently contained within the existing uncertainty estimates.

2. There is often a downward slope during interstadials, especially in the model time series. However, it is not obvious whether the model should be changed to allow/detect a slope during the stadial pre-ramp period. Do the authors find significant (negative or positive) slopes of pre-ramp levels in real data (both proxy and model data)? It is simply not clear in the manuscript whether the addition of a slope in the stadial fixes a real problem. It is not surprising that allowing a negative slope in the stadial before a steep positive ramp will tend to lead to earlier ramp onsets, especially with noisy data. As I understand it, this would likely lead the authors to conclude that the model without stadial slopes has a bias. But if there is no significant slope in the stadial (and preferably a physical reason to why such a slope would be expected), it is not clear that allowing the model to have a sloping stadial is an improvement in terms of representing the underlying physics. If there is no significant stadial slope, an observed difference in the timing of the ramp onset between the two models (with/without stadial slope) is simply a result of different assumptions about the signal structure and not necessarily a bias as such. The authors should analyze this question carefully to justify whether/when/why addition of a pre-ramp slope is an improvement. And they should test (for synthetic data created without any pre-ramp slope in the noise-free data) how adding this extra degree of freedom changes the detected ramp characteristics and report the results explicitly in the manuscript.

3. In Capron, the results of a single test of possible bias are given in the Methods section (and Fig. S2). Here, no significant bias was detected (the 50-year duration of the synthetic data was reconstructed as approximately 51 years by the method (and as an add-on to the previous point: the results in the S2 figure illustrate that the uncertainties are considerable and easily would include a bias of a few years). Because the results differ, it would be interesting if the authors could discuss their analysis in the context of the published test. Maybe the way the test data were created could make a difference (also see next point)?

4. The authors do not describe in detail how they generate their synthetic test data, and when they estimate the bias, how many independent realizations are run (my experience is that especially for high noise levels, averaging over at least several tens of realizations is necessary). This information should be provided somewhere. Also, and especially because the conclusions differ from those of earlier tests (e.g. those mentioned in Capron), the synthetic data series used here should be made available to allow others to make future tests against the exact same test data, thus reducing the risk that differences could be caused by implementation issues.

5. Related to the former point, one could suspect that especially when adding relatively high levels of autocorrelated noise, the transitions (i.e., ramps) of the synthetic data actually become more gradual (with longer durations and/or earlier onsets as a consequence). This is what was tested in Capron Fig. S2 for one scenario, but the results could be different for other parameters and other noise-to-signal levels (see PS below). If this indeed is the case, I would say that it changes the way we should look at this: it's not "just" the ramp fitting procedure that has an issue, but an inherent limitation related to the nature of climate data with autocorrelated noise (or data affected by autocorrelated short-term variability, if you like). I think this issue is related to what the authors are hinting at around line 226-228, but it could be expanded and made clearer.

6. Somewhat along the same lines: It is not surprising in itself that high noise levels lead to longer transitions: As the noise level increases, (probably all) ramp detection methods will have a harder time reconstructing the (no-noise) ramp duration, and there will therefore be a larger spread in the detected ramp durations. If the noise is large enough, the range of ramp durations becomes asymmetric because the ramp durations by design are positive. An example with a synthetic ramp duration of 50 years: For high noise levels, the distribution of accepted solutions produced by the method will include values that are larger than 100 years, but by design none which are smaller than zero. The mean of the distribution will be biased towards larger durations and lead to earlier onset points (and/or later ramp end points). In the test described in Capron, this effect of asymmetric distributions is observed although it does not lead to a bias in transition duration similar to the one reported in the current manuscript (see PS below). However, by using the median rather than the mean, the effect on the result can be reduced. This highlights two points. Firstly, using the median may be more appropriate than the mean. This should be discussed and tested explicitly. Secondly, this is not really an issue with the ramp fitting method, but a consequence of noisy ramped data.
The observation is consistent with Fig. 3h which shows that there is largest bias towards early onsets for the most noisy and fastest transitions.

I think it would strengthen the manuscript a lot if the authors could go a bit deeper into discussing the issues raised in point 5 and 6. If they agree with these points, I encourage them to think about how to phrase their results in a way that better acknowledges that it's not "just" a methodological problem with ramp fitting, but an inherent limitation that both is related to the method and to the nature of autocorrelated ramped data and/or that (some of) their bias comes from the way the synthetic data are created. The conclusion of Capron already discusses some of these issues (and tries to separate those that originate from the complexities of the climate system and those that are more related to the noise and method). It would be good if the authors spend a bit more space on discussing how their results confirm, refine (or even contradict) the ideas presented there, and separate between the different types of difficulties.

[PS on the test from Capron Fig. S2:
While writing the review, and with inspiration from the manuscript and some tests performed by a student in Copenhagen last year (not published), I first ran some more realizations of the same test as in Capron: The mean reconstructed ramp duration was 51.9 years, and the median 51.5 years.

This would probably correspond to an early ramp onset bias of ~1 year, considerably less than reported in the manuscript at hand, and well within any reasonable uncertainty estimate based on the width of the distributions. Reducing the noise level to half led to no significant bias (median 49.6 years, mean 50.2 years). Then I tried with double noise level (stadial noise level is the default in this test, which is already a relatively high noise level) and observed a more pronounced bias in transition length, which was also larger when using the mean rather than median. I did not check this through well enough to share the results, so I provide only this indication. I think this supports the idea mentioned in point 5-6. It also indicates that in Capron (at least for Calcium and in the absence of significant slopes of the pre- and post-transition values), bias does not play any important role for the results.]

On a more technical note, I assume that the authors have tested their implementation (including generation of synthetic data). It would be nice with a summary of the tests: Does the method faithfully identify the properties of the synthetic data (including the autocorrelation time) for low noise levels? Figure 3 goes some way in demonstrating this, but I think a bit more details and a dedicated section would be appropriate (possibly in a supplement).

Finally, I think the title is not fully aligned with the content and conclusions. The following line from the conclusion "Despite the bias that we have uncovered here, we nevertheless suggest that this Bayesian ramp fitting method remains the best approach to understanding the temporal phasing of DO events" does not really work well together with the title "The Temporal Phasing of Rapid Dansgaard–Oeschger Warming Events Cannot Be Reliably Determined".

In summary, I think the manuscript is appropriate for publication in Climate of the Past, but that additions and revisions are needed to fully document the analysis and make it clearer to which degree and under which circumstances the extended model with stadial slopes represents an improvement in the context of the data at hand, discuss in more depth whether the bias is a methodological problem or arise from the inherent difficulties of identifying ramps in noisy data, and assess whether the proposed bias will influence earlier conclusions based on the ramp fitting methodology.

Review by Sune O. Rasmussen, Jan 10th, 2024.

Line-by-line comments

29: "However, as of yet it has not been possible to conclusively identify a cause for these rapid Dansgaard–Oeschger warming events" could be rephrased. The exact mechanisms leading to the described changes are not understood but the AMOC state change is generally accepted as the main cause of most of the observed changes in the climate system.

34-39: The papers of Adolphi and Corrick align records from ice cores and speleothems and have to deal with non-climatic dating offsets between different dating approaches. The WAIS and Svensson papers use synchronized ice-core records. Because the uncertainties (in terms of both magnitude and nature) are quite different, it would be better to describe them independently. Bias in event detection of ~20 years are only marginally relevant in this context, which is not the case for the methods described in line 39-41.

39: (Buizert et al., a) should be cited as "WAIS Divide Project Members".

47-48 and 51-52: Elaborate or remove. The current sentence is vague. How do they contradict each other (given the error bars)? Consider if it would make the discussion clearer to include a discussion of the difference of absolute vs. relative dating errors. An ice-core timescale can build up significant bias over long stretches, and the published maximum error of GICC05 is therefore large in MIS 3.

58-63: I believe the papers came out in the reverse order: Capron et al. was accepted while the manuscript of Riechers and Boers was in the discussion phase. Please adjust wording accordingly, and in particular, be careful to cite the conclusions of Capron et al. correctly. I do not find a statement in Capron et al. to support the claim "they also concluded that applying a method of the type developed by Erhardt et al. (2019) to ice core data may not allow the identification of a unique order". In contrast, Capron et al. acknowledges that the Erhardt et al. approach of stacking may be appropriate under certain assumptions. The lack of a well-established order of events which is similar between events could both be due to noise / internal variability making detection difficult, but also that such a "standard sequence of events" may not exist.

72: Maybe reconsider use of "cause" (as above).

90: Proofreading needed.

93-94: Vague. Refer to Vettoretti et al. 2022 Fig. 2a or describe how your analysis is different.

110: It is not clear what "parameter ranges" refer to.

117: Describe somewhere (here, in Methods, or supplementary material) what "for which the method was successful" means.

119-120: Add charge to ion or spell out full names.

120: Note that Capron et al. did not make a one-to-one "identification" of proxies and model data extracts: "Over each D-O-like transition, we extract the time series of four climatic measures from the model on the assumption that they reflect some of the same elements of the climate system as our ice-core proxy data". Revise accordingly.

122: Revise syntax.

137: It would be prettier if x(t) was defined for t_0 and t_1 as well (as in appendix).

154: This is a critical assumption which is also employed by Erhardt and discussed in Capron. I encourage the authors to consider and discuss the implications of the assumption (e.g. here or in section 2.5).

176: It is more common to use the SNR and not the "Noise / Signal ratio". I recommend using the more standard SNR unless there are good reasons for why not to do this.

178: Give more details on how the synthetic data were generated in order to realistically mimic true data.

192: (also see main point raised above). It is hardly surprising that adding a lot of noise to a steep transition tends to smoothen out the transition found by the ramp fitting method. In order to judge how serious this problem is, I suggest adding the following pieces of information:

- How does the calculated UMOTE compare to the distributions of onset times? I.e., are the confidence intervals wide enough to include the bias?

- It would be informative to add some estimates of noise/signal for the actual data used by Erhardt at the start of this section, so the reader gets a feeling for how the observed biases compare to what would be expected in real data.
- Same for the stadial and interstadial slopes: What are the realistic values for the Erhardt data sets?

2010-201: (Optional). Maybe comment on how an autocorrelation time on decadal scale could occur in the climate system?

202: Figure 3: Mention that the scales are not the same for a-d end e-h, respectively. I guess that noise/signal does not mean the same for annual and decadal data. Is there a way to compare the two more directly? For example for a certain annually resolved data set with a given noise/signal value, what is the corresponding noise/signal value for the same data set averaged to decadal resolution? That would elucidate the dependence of resolution more clearly.

206: For the tests in C1-C4: Are these tests run on the same synthetic data? How exactly were the synthetic data generated, and what value of the autocorrelation length was applied? Synthetic data with characteristics spanning a range of parameter values are tested, but without describing how the ranges compare to those of the real data, the results are hard to interpret in a useful way.
For the test on Figure C2: Is this relevant given that Capron did not use data with decadal resolution? Or am I misunderstanding the caption: "transitions with decadal resolution"?

214-216: Please provide the details of the autocorrelation range observed in the model data. What noise-to-signal characteristics were chosen? It's fine to have the results illustrated in Fig. 3, but they should also be tabulated.

225: It would be clearer to integrate the section starting at 230 with this section.

241: Please explain what you mean by "reliability".

245: Please provide table with the characteristics of the synthetic data that resemble the ice-core data sets.

265: (Figure 5). It is not clear whether the Calcium data have also been corrected for bias and how this factors into Fig. 5: Is the zero defined from corrected or uncorrected Calcium data ramps? Please provide the details.

303: I don't see why this is "another issue". It is quite clear that observing a too long duration is equivalent to the combination of a too early onset and a late end point, and as stated in the introduction, I would claim that both effects are likely to occur when you turn up the noise.

312-15: Section 3.2 is based on CCSM4-like data, and it is not clear if the same result holds for paleoclimate data. Please clarify/rephrase.

Throughout: Please add hyphens consistently to "sea-ice extent", "ice-core records" and other similar compound adjectives. It's somewhat a matter of style, but generally increases readability.

---

## Referee Comment (RC3)

In their manuscript Slattery et al. present an analysis of the performance of a modified ramp-fitting method for the determination of the onset of stadial to interstadial transitions and apply this modified method to decadal resolution model data.

Overall, the topic of the manuscript is of interest to the wider paleo-climate community and especially the detailed analysis of interstadial onsets in model output is interesting and a good fit for CP However, there are some larger issues with the manuscript concerning the conceptual background as well as to the presentation of the results that need to be worked out before the manuscript should be considered for publication.

**General remarks**

The authors extent the ramp-fitting method of Erhardt et al. (2019) (E19) by including two extra parameters for the pre- and post-transition slope of the data. This is necessary to better describe the model-output that the authors want to apply the method to, especially the AMOC strength. The authors do not show, if the inclusion of the slopes into the model is necessary for all their data and also do not provide any evidence that the inclusion of the slopes decreases the uncertainty of the estimates for the relevant parameters. One would also hope, that the addition of two extra parameters to the model would make the model consistently outperform the original E19 version. Judging from the figures in the appendix this does not necessarily be the case. However, due to presentation of the results that is difficult to judge.

Before coming continuing, a note here on terminology: The authors chose to coin a new term for the average value of the posterior samples. This is unfortunate, as there already is a term that is used for this: posterior mean or more specifically the marginal posterior mean. This is the term that I will use and that I strongly urge the authors to use, too.

In an extensive sensitivity study, the authors test if the posterior mean of the transition onset is on average an unbiased estimator of the onset of the transition. Most, if not all, of the content of the paper hinges on the results of this sensitivity study. Despite that, the authors never explicitly state the exact focus of the sensitivity study and the implications that come with this choice. However, that exact focus of the tests is a big potential issue: In the sensitivity tests, the marginal posterior mean does not need to equal to the known parameter value and same is true for any other point estimate from the full marginal posterior. If the posterior is skewed or multi-modal then any point estimate will grossly misrepresent its true location and disregard its shape. Especially if skewness of the marginal posterior is systematically induced due to the noise or due to trends around the transition, it can introduce a systematic bias into the marginal posterior

mean. This can for example be seen in Figure 2 of E19, where the two marginal posterior distributions for the start and the end of the transition show a clear indication of skewness and in the case of the onset maybe even bi-modality. The same argument is of course true for the length of the transition and also extends to all other estimated parameters. Sune Olander Rasmussen (SOR) raised a not dissimilar issue in his review: By using the marginal posterior mean the authors disregard the sometimes large uncertainty in the estimated parameters that is carried in the complete marginal posterior distribution. As pointed out by SOR, the width of the marginal posterior distributions can easily be larger than the reported bias of the posterior mean. From my point of view, this sole, narrow focus on the marginal posterior means is in stark contrast to the strong general statements on applicability of the method throughout the paper, especially because this tight focus is not explicitly communicated to the reader.

In addition to the conceptual shortcomings of the sensitivity study, the presentation of its results in Figure 3 and in the appendices is also not ideal. If I guess correctly (there is nothing written in the manuscript), the authors average over the variables that they do now show to present the data in a reduced number of dimensions. The main issue with this approach is, that it hides any possible inter-dependencies between variables and does not allow for clear separation of the effect of the single variables. I think this is a missed opportunity and might be potentially misleading, depending on the parameter interdependencies.

The authors apply their modified ramp model to climate model output of DO-type transitions that is available in decadal averaged output. For this the authors chose to stay with an AR(1) noise term for the application of the ramp model. I am, like SOR, a bit puzzled by the long autocorrelation times in Fig 3e. Of all the parameters only the ocean circulation would likely have long-term memory, of all the other parameters, I would assume that decadal averages would mostly eliminate the autocorrelation. I would ask the authors to provide a more detailed explanation of this phenomenon and a discussion of the influence of the long autocorrelation times on the timing estimates. Generally the autocorrelation of the residuals in the model is in competition with the transition ramp itself, for lack of a better term. If the model fails to describe the noise of the data accurately then the model will give unreliable results. Long autocorrelation times can smear out the transition for sure leading to larger uncertainties and maybe even "biases" towards earlier or later transitions. This can make the choice of the noise model and the choice of priors for its parameters very important. In the case of the presented study, the authors explicitly excludes the case for non-autocorrelated data and put, in comparison to E19, larger prior probability on long autocorrelation times. Especially the former might be a serious issue in case of the decadal averaged data from the model and merits a thorough discussion.

To overcome the bias that the authors discovered, they use "analogous" data to estimate the bias and later correct for that. Generally the authors do not provide much detail about the exact parameters that they use to generate the "analogous" datasets that they use to make the bias correction for the different parameters. It would be great if the authors could provide more detailed information about the parameters that they chose such that the results are later reproducible. Their statements about the fact that they cannot reliably estimate the onset or the length of the transition given their methods makes me also wonder how they estimated all the parameters that they need for the "analogue" data. Did they use a different method or can they use the ramp-fitting method to estimate all other parameters needed for the surrogate data reliably? Especially in light of the statement by the authors that the relationship of the bias to the ramp parameters is complicated it would seem to be necessary to estimate this parameters

very precisely or carry out the bias calculations for a range of possible parameters. I strongly urge the authors to investigate this in more detail before making definitive claims about a bias. The proposed bias correction also carries additional uncertainty with it, how do the authors propagate that to the estimates later?

In the analysis of the modelled precipitation data, the authors state, that the noise characteristics of the data change over the transition. The authors somewhat disregard this observation with the notion that they failed to include this in the model. However, the results of the analysis are dependent on the noise model adequately describing the data. This is especially worrisome as the precipitation time series exhibits the biggest lead over the temperature in the model data. The fact, that the authors state that inclusion of changing noise characteristic over transition negatively impacts the reliability of the method might hint at a deeper problem. This makes me question the results that they present and the applicability of the estimates for the generation of the "analogous" data and thus the bias correction that the authors apply

**Specific remarks**

**58–63**  Order of publications is wrong.

**Table 1**  This suggests a connection between Ca and the NAO, please justify or remove

**125ff**  Focus of the work: If that is the case, please rephrase the introduction and the abstract accordingly. In this case you can also remove all the model data. Or spend some time discussing the data and its interpretation. There is no need for statistics without a clear question to answer.

**138ff**  This is a wrong statement. If that were true then OLS-regression would be probabilistic, too.

**143ff**  Please show proof of this.

**149**  Remove reference to Erhardt et al., as they do not use any frequentist methods.

**169ff**  Nomenclature: There are some already established terms that the authors should use as much as possible instead of inventing new ones: the average of the posterior distribution is the posterior average or posterior mean.

**Fig 3**  The presentation of the results is unclear. The authors do not state what they do with the other parameters that were varied. I assume that they just averaged over all the other parameters that are not shown in the figure. However this approach will hide any co-dependencies of the bias between different parameters.

**213ff** : Please specify how you set these two sets of data up. I would imaging that the exact difference between the two resolutions depends a lot on the parameters as well as the timing of the onset, depending if it happens at a boundary of an averaging period. The results here are also contradicted a little by Fig C1, which shows clear differences between annual and decadal resolution data

**Table 2** The bias of what? Specify in the table header.

**Sect 3.3** Please specify the exact parameters used to generate the analogous transitions. You say that you cannot reliably estimate the transition length, but you do not show that you can actually estimated the other parameters without issues. I think this is minor but a few sentences in this regard would be good.

**230ff** It is unclear how the posterior mean and event-means are set up here. Are you generating 19 events or are you subsampling 19 events out of your 1000 realisations to calculate the event-averaged timing?

**239ff** Please elaborate on your the heteroscedacity issue: If the method is unreliable with the inclusion of different noise levels over the transition, than how is it reliable enough to estimate analogous parameters. It is also unclear if you generated the analogues with or without heteroscedacity which makes the last sentence of the paragraph quite confusing. Please rework. It also needs to be clearly shown and discussed whether the bias correction for the modelled precipitation is valid and if the model can be applied at all to the heteroscedastic data. The issue with these models is that their results are conditional on the model being a good description of the data, if that is not the case the results are to be taken only with a grain of salt and proper justification.

**Tab 3** : Please do not format the table in such a way that Ca seems to be associated with the NAO. This is potentially misleading especially as you shy away from discussing the proxy interpretation at all. Also, Even though temperature and precipitation over the North Atlantic that you use is probably not unrelated to the values at NGRIP, they are not the same. Please redo the analysis with the relevant variables or rework the manuscript in such a way that it is clear that you are not looking at model output for Greenland at all. This included the Table where these are stated next to each other as if they are the same.

**255ff** Please discuss why the lags are so different between the different variables. Is that because of the shape of the transitions or the noise levels or both?

**Sect. 3.5** Again there is no clear description on how the analogous transitions were generated. Without that information the results cannot be reproduced or verified by anyone.

**281ff** I think the leading paragraph is a bit misleading without the actual information under which circumstances the bias arises.

**285ff**  This paragraph probably should state somewhere that you look at this from a frequentist point of view. Erhardt et al. never made any claims about statistical significance, they provide credible intervals which are the parameter ranges that are consistent with the data under the assumptions of the model. This is not the same as the assumptions that go into the significance test, namely that under repeated observations and calculation of the significance interval of 95% the true value of the parameter will fall into the significance interval 95% of the time. These two views are fundamentally different and are not necessarily compatible with each other.

**318ff**  The paper mostly focuses on the adjusted method that you are proposing, and not the original method of E19. Please rephrase as all results with regard to the original method are only in the appendix of the paper.

**327f**  The conclusions of Capron et al. (2021) are based on fundamentally different reasons than the once here. Your's are purely related to the method and the interpretation of the results whereas those of Capron et al. (2021) are based on considerations with respect to climate variability and the tight coupling between different parts of the climate system. Please remove or rephrase.

**329ff**  The conclusion is inconsistent. The method is either good or not for the investigation of DO events. Please decide and rewrite. Please also state why you think that the method would be suited if the lags are larger than 20yrs. This is not entirely clear given the discussion beforehand.

**Supplement**

**Choice of prior distributions**

- Either Equation 10 or your code is not correct, please check.
- The prior for tau in your analysis puts a lot more emphasis on much longer and much shorter autocorrelation times than the one in the original method. This is a bit problematic as we do not expect to see decades long autocorrelation in climate data as far as I know. Please elaborate.

**Sampler setup**  Please clearly describe the setup of the ensemble sampler. I have noted that you chose to keep the setup of E19 despite increasing the number of parameters. This should also go hand in hand with a change in the sampling strategy i.e. burnin and thinning and maybe even the number of ensemble walkers. These need to be carefully chosen to not get nonsensical results in the end.

**Figures C1-C3**  Judging from a closer comparison of the results from your method to the results of the original E19 method in Figures C1 and C2 it would seem that the addition of the slopes generally decreases the performance of the method. This seems to be the case for the sensitivity of the bias towards noise, the transition length and very, very worryingly towards the inclusion of an interstadial slope. Please elaborate and discuss this especially in light of the analysis that you base upon the result from the fit including the slopes. Also it would seem, that at zero slopes prior and after the transition the original method overall outperformed yours, which would mean that part of the bias is because of the inclusion of the slopes into the model not despite.

---

## Author Comment (AC2)

**The Temporal Phasing of Rapid Dansgaard–Oeschger Warming Events Cannot Be Reliably Determined**

**Authors' response to the referees' comments.**

**Referee #1**

**The authors would like to thank the referee for their insightful comments and suggestions. The referee's comments are normal text and our responses follow in bold text.**
* * *
**General comments:**

The approaches described here, i.e. studying the impact of the Bayesian ramp fitting on the uncertainty of the estimation of lags between climatic parameters simulated by GCM or reconstructed from ice core records, is usually a discussion topic for paleoclimate studies. I think it is a perfectly valid study, and do not question the relevance of the publication, but highlight this point because I would say that the results of your study correspond to what would be the discussion of a paleo reconstruction of a DO event lag study: indeed, classical paleo papers would estimate the lags and in the discussion attempt to provide uncertainties or biases of their estimates. Here, I found the structure of the result section a bit confusing and would suggest the authors to reorganise it a bit keeping in mind what the results are. Specifically, to me, the actual results come in section 3.3., while sections 3.1 and 3.2 detail sensitivity tests on the parameters that were used for the synthetic transitions and the resolution.

**We recognise that the structure of our study is somewhat unusual for the field. However, the main purpose of this work is to test and critique the method being employed. From this perspective, section 3.3 is simply a further example that demonstrates the importance of the bias we find. We considered employing a more "standard" paleo reconstruction structure in which we would present a results section focused on the ice core data and then demonstrate the bias in the discussion section. However, if our study were structured in this way then the key finding of bias would not be introduced until late on in the manuscript, at which point it would largely invalidate the "results". For these reasons, we think that it would be more sensible to retain the current structure.**

Overall, there are implicit equivalences that are made between the study of the outputs of CCSM4 and the parameters reconstructed from the NGRIP ice core which might be not accurate.

In Section 2.2, a list of what climatic parameters are usually associated with the proxies retrieved from the ice core record is provided, but it is mentioned that the one to one equivalence is not totally correct, for instance, δ 18 O is influenced by both temperature and sea ice extent changes (Sime et al., 2019). Then, the way that Table 3 and Figures 4 and 5 are represented suggests that some parameters are equivalent with the way that they are represented either on the same line or with the same colour.

**As multiple referees have made similar comments, it is clear that the equivalences we suggest between ice core proxies and model output variables are a source of confusion and that this distracts from the key point of our study. We will therefore remove any such suggestions of equivalences from the revised manuscript. In particular, we will separate out Table 3 in such a way that it is clear that we are dealing with two different sets of data between which we do not make a connection. Furthermore, we will ensure that Figures 4 and 5 use distinct colours to avoid any suggestion of equivalence.**

Finally, I'm a bit confused with the choice of the reference parameter for the lag estimate for CCSM4 and the NGRIP ice core. Indeed, I don't understand why in CCSM4 the lag is calculated against temperature, while NGRIP ice core, it is calculated against the calcium, which is referenced to be a proxy of the NAO. This might be something classically done here, but since amongst the four parameters used from NGRIP ice core, δ 18 O would be the one the most closely related to temperature, it appears that the lag as presented here would have different meanings for models and paleoclimate reconstructions.

**We agree that this is potentially confusing, and thank the referee for pointing this out. The choice of the reference for the lag estimates is largely arbitrary, in the sense that we do not consider any single proxy / variable to be more important than any others. As we will remove any equivalences between proxies and model variables going forwards (see comments above), we hope that this confusion should resolve itself. The analyses of the model simulations and the ice core proxies should be considered as separate demonstrations of the impact of the bias we find.**

**Specific comments:**

Introduction: the introduction overall feels too detailed, to the point that I struggled to see what was the topic of the manuscript when I first read it.

**We will reduce the length of the introduction and make sure that the topic is made more obvious. In particular, we will remove any reference to the multi-archive studies (Buizert et al., 2018; Adolphi et al., 2018; Corrick et al., 2020; Svensson et al., 2020) as these are only tangentially relevant to this work.**

Line 58 says "Subsequent to this work…", suggesting that the overall timeline of the study of lags of climatic parameters for DO events are detailed, while an introduction probably just needs the most up-to-date information that is needed to understand the interest of the manuscript.

**We agree that the phrasing here suggests a more detailed review of the literature than is necessary or is indeed the case. There are three key papers regarding this topic that have been published in the last five years (Erhardt et al., 2019; Capron et al., 2021; Riechers and Boers, 2021) and we think that it is important to discuss the key findings of each. However, we shall rewrite this section to make it clear that these three papers come to somewhat different conclusions, and that there is not yet full consensus, rather than suggesting (as our initial manuscript perhaps does) that there is a timeline whereby later research has superseded earlier work.**

Line 44: "On the first line of evidence, Adolphi et al…" It is not clear what line of evidence this is. The similar formulation line 53 also puzzled me.

**In our original manuscript, by "the first line of evidence" we refer to multi-archive studies, and by "the second line of evidence" we refer to analysis of multi-tracer records from Greenland ice cores. We will remove any reference to multi-archive studies, which will allow us to simplify the introduction and ensure that it is clearer.**

Line 89: "which Vettoretti et al. (2022) have kindly provided." "Kindly" is not an appropriate word to use, especially considering that most journals now require open access to the data of a published paper.

**We will remove the word "kindly" from this line.**

Lines 124 – 125: "For in-depth discussion of the physical interpretation of these proxies, see Section 2 of Erhardt et al. (2019) and references therein." This could be personal style, but I believe that a manuscript should be stand-alone. The detailed discussion of the physical interpretation of the proxies is beyond the scope of this manuscript, but overall idea should be provided in the introduction or the methods.

**We would reiterate that this study is primarily concerned with testing the Bayesian ramp-fitting method, and so we feel that the current level of**

**discussion of the proxy interpretations is sufficient. As discussed above, we will also remove any suggestion of equivalence between proxies and model variables, which we feel further reduces the need for detailed discussion of proxy interpretations in this work.**

Lines 128 – 129: "The aerosol concentrations were measured using Continuous Flow Analysis (CFA) at temporal resolution ranging from 2 years for the most recent period to 3 years for the oldest period" Was anything measured for this manuscript? Is the fact that the data are measured via CFA relevant here? This sentence was very confusing, making me think that you measured aerosol concentrations within this manuscript. Overall, only the effective resolution of the data is important here. Considering that all this information is already provided in the table, I would remove this paragraph.

**We agree with this suggestion and will remove this paragraph.**

Line 138: "The addition of an AR(1) noise process makes the model probabilistic" Where is the AR(1) noise added? Why is it not explicitly given in an equation which shows exactly what is added where?

**Thank you for raising this. We will explicitly write out the AR(1) noise in equation (1).**

Line 143: "This leads to improved agreement of the transition model with the analysed data (Figure 1)." This is not clear. Is the improved agreement shown in the histogram below each figure? What do they mean? they have no label, no caption, and no explanation.

**Thank you for this remark. Indeed, the histograms are not conveniently described in the figure caption, nor are the y-axes of the histograms labelled. We apologise for the oversight and will correct this issue in a revised manuscript. The blue and orange histograms below each panel respectively show the posterior distributions for the onset and end times of the transition. The y-axes of the histograms should be labelled as Probability Density / Years$^{-1}$, as is the case in Figure 2.**

**We have not quantified the overall improvement of the fits. However, we believe that the improvement is clear from a visual comparison of the top row with the bottom row, particularly panels c and f. We will add the sentence: 'The additions of slopes before and after the transition yields a clear improvement of the fit of the deterministic model core to the data.' at the end of the model caption.**

Lines 143 – 144: "Additionally, our extension of the method reduces the sensitivity of the transition timing to the search window, which is otherwise one of the drawbacks of this method (Capron et al., 2021)." Also unclear what is meant here.

**We agree that the sentence is not directly clear for people that are not already familiar with this method. We therefore will add the following explanation in a revised manuscript: 'In order to apply the ramp-fit to individual transitions, one has to select a data window around that transition. There are no objective criteria for the starting point and the endpoint of these windows, other than that no other transition should be included. However, changing the boundaries of the data window influences the results of the estimation. This effect is reduced if slopes before and after the transition are allowed.' Also, as requested by another referee, we shall add here a quantification of this reduction in sensitivity to the search window for a particular case.**

Line 157: The detailed description of each hypothesis is one line of text each. They should be included here, and not in the appendix.

**Thank you for this remark. We will move the formulation of the hypothesis to this point.**

Lines 160 – 161: "We conduct our analysis of the NGRIP ice core in the same manner. In this instance, we calculate time lags for the other three proxies relative to Ca"

Why calcium which you say represent the NAO (line 120), and not d18O which is closer to temperature? This would be more coherent compared to what is done with models?

**As discussed above, we will remove any equivalence between proxies and model variables. The choice of reference from which to calculate leads and lags is then arbitrary.**

Lines 227 – 229: "However, we have established that there is no unbiased means by which to estimate these transition durations, and so we could not guarantee that this would be any more accurate."

Where was this established? And shouldn't be this line in discussion rather than in Results?

**Indeed, this statement is a bit misleading. We were trying to say the following: For constructing the analogous synthetic transitions, we could in principle estimate typical transition durations from the data using the ramp-fit. However, we have established that the transition duration is itself prone to bias, as bias**

**in the onset time directly propagates to bias in the transition duration. This is mentioned in lines 202-205.**

**We cannot definitively say that there does not exist any method to accurately estimate the transition duration, as this would require us to test every conceivable approach. What we were instead intending to communicate is that we have not been able to identify any such unbiased method for estimating the duration. Because of this, we simply used a plausible fixed value to estimate the bias in the different model variables and ice core proxies.**

**We agree that this line would more suitably fall in the discussion, and will move it there in our revised manuscript.**

Figure 5: The matching colours between Figures 4 and 5 suggest equivalency between precipitation and Sodium, sea ice and d18O, and AMOC and thickness. The colours should be different in both figures if they are not supposed to represent equivalent parameters.

**We thank the referee for noting this. We will change the colours to avoid any suggestion of equivalence.**

**Referee #2**

**The authors would like to thank Referee #2 for their extremely thorough and useful comments. The referee's comments are normal text, and the authors' responses are in bold.**
* * *
The manuscript presents an analysis of some limitations of ramp fitting as a tool to determine the characteristics of abrupt events seen in palaeoclimate data and model runs. The study is relevant, generally well presented although (some more details could be wished for in places), and suitable for publication in Climate of the Past. First, the ramp fitting method is extended to take pre- and post-transition trends in the data into account. This is a good move when the method is applied to model data which exhibit more pronounced trends in especially the interstadials, but see further comments below on the need for modelling a trend in the stadials. The main finding is that the ramp fitting method detects ramp start points that are generally too early when the method is tested on synthetic data, especially with high noise levels. This effect is denoted "bias", and the bias is then quantified for a range of scenarios.

The effect can be anything from one year upwards, depending mainly on the noise-to-signal levels. I have some methodological concerns and some regarding the presentation of the results, that I think should be addressed before publication:

1. The manuscript does not really discuss the magnitude of the suggested bias compared to the credible intervals produced by the ramp fitting method as published in earlier studies. If, for example, the bias is a few years in cases where the credible intervals span decades, the bias will unlikely lead to wrong conclusions. In this case, the existence of a systematic bias would still be a valid result (and to people interested in the methodology also an important one), but the analysis should consistently address whether the bias is sufficiently contained within the existing uncertainty estimates.

**This is an important point, and we thank the referee for drawing attention to it. As can be seen from the blue histograms in Figure 1, showing the posterior distribution for the transition onset time, the width of the credible interval is generally on the order of a few decades. It is therefore absolutely correct to state that, for individual events, the bias is contained within the existing uncertainty estimates. So, if this method is used only to compare individual events, as for example is done by Capron et al. (2021), then the bias we have uncovered is relatively unimportant.**

**However, this changes if one attempts to calculate characteristic leads and lags from a set of DO events. Although employing different methods, both Erhardt et al. (2019) and Riechers & Boers (2021) essentially combine many events in order to reduce the uncertainty on the leads / lags between proxies. Because of this, the range of the credible intervals shrinks to around a decade - for example, Erhardt et al. quote a lag of 7 ± 6 years between calcium and sodium in the NGRIP core. The bias we find is therefore no longer contained within the existing uncertainty estimates, and so when trying to assess leads or lags this bias could easily lead to wrong conclusions.**

**This is the point we are trying to make with our hypothesis test in Section 3.4: Without any consideration of bias, we have an apparently statistically significant lag between precipitation and temperature in the CCSM4 simulations. However, if we correct for a plausible estimate of the bias, this is no longer the case. This is a clear example of a situation where the bias leads one to the wrong conclusion.**

**We agree that our original manuscript does not make this distinction explicitly enough. We will make clear in the discussion of our revised manuscript that the bias only becomes particularly important when considering leads and lags that have been calculated by combining many events, reducing the uncertainty in the process.**

2. There is often a downward slope during interstadials, especially in the model time series. However, it is not obvious whether the model should be changed to allow/detect a slope during the stadial pre-ramp period. Do the authors find significant (negative or positive) slopes of pre- ramp levels in real data (both proxy and model data)? It is simply not clear in the manuscript whether the addition of a slope in the stadial fixes a real problem.

**Perhaps the simplest way to persuade the reviewer that there are meaningful slopes during the pre-ramp stadial is to show some examples where this is obviously the case - for example AMOC in the CCSM4 simulation with 225 ppm of $CO_2$.**

[Figure]

*Additional Figure 1: The extended ramp fitting method as applied to AMOC for three abrupt warming events in the CCSM4 simulation 225 ppm of $CO_2$. There is an obvious positive pre-ramp (stadial) slope in all three cases. This will not be included in our revised manuscript, but it instead simply included here to address the referee's comment.*

**Furthermore, for the NGRIP ice core proxies, our extended method finds slopes of comparable magnitude in the pre-ramp stadial and post-ramp interstadial. This will be made clear in our revised manuscript as all of the relevant parameters used to create the "analogous" synthetic parameters for the purpose of bias estimation will be listed in a table in the appendix. We most often find that the slope during the pre-ramp stadial is positive - that is to say in the same direction as the ramp itself. See also the mean parameters for each proxy (including slopes) shown on Additional Figure 2 (page 10). The proxies all show significant slopes, in the sense that these slopes lead to a significant bias, however we have not tested the statistical significance of these slopes in isolation.**

It is not surprising that allowing a negative slope in the stadial before a steep positive ramp will tend to lead to earlier ramp onsets, especially with noisy data. As I understand it, this would likely lead the authors to conclude that the model without

stadial slopes has a bias. But if there is no significant slope in the stadial (and preferably a physical reason to why such a slope would be expected), it is not clear that allowing the model to have a sloping stadial is an improvement in terms of representing the underlying physics. If there is no significant stadial slope, an observed difference in the timing of the ramp onset between the two models (with/without stadial slope) is simply a result of different assumptions about the signal structure and not necessarily a bias as such. The authors should analyze this question carefully to justify whether/when/why addition of a pre-ramp slope is an improvement. And they should test (for synthetic data created without any pre-ramp slope in the noise-free data) how adding this extra degree of freedom changes the detected ramp characteristics and report the results explicitly in the manuscript.

**We would like to make it very clear that we always calculate the bias against the true, known value of the onset time for synthetic transitions. We do not ever conclude that bias is present based on the difference between the original and extended implementations of the ramp-fitting method. Furthermore, panel f in Figure 3 shows that we actually see later ramp onsets (compared to the case of zero slope) when we have a negative slope in the stadial before a steep positive ramp - contrary to what the referee suggests they would expect.**

**However, we agree that we have not sufficiently compared the efficacy of the original (without slopes) and extended (with slopes) implementations. We will therefore include a further figure which is equivalent to Figure 3 but based on the original Erhardt et al. implementation (Additional Figure 2, page 10). From visual comparison, we can see that the original implementation (with no slopes) shows lower bias for the case of no or negligible slopes in either stadial or interstadial (the middle band of panels b and f, or the bottom region of panels c and g). However, this original implementation suffers strongly from bias when there is even a slight slope in the pre-ramp stadial.**

[Figure]

*Additional Figure 2: The bias in the transition onset time, based on sensitivity testing of the original Erhardt et al. implementation of the ramp fitting method, for annual (a-d) and decadal (e-h) resolution. Overlaid are points showing the mean transition parameters for the ice-core proxies and model output variables that are considered in this study. When not explicitly varied, parameters are fixed at standard values. These panels will be incorporated into Figure 3 in our revised manuscript.*

**When faced with transitions that include significant pre- and post-ramp slopes, we see better results (less bias) from representing these slopes in the ramp-fitting method. However, when dealing with transitions that contain no such slopes, we see worse results (more bias) as we have introduced two unnecessary new degrees of freedom. For more realistic intermediate scenarios it is less immediately obvious which of the two implementations would lead to more bias. To address this point, we will add an extra column to Table 3 that includes the bias based on the original Erhardt implementation that does not include slopes. We include these results here to demonstrate that neither of the two implementations is obviously superior overall.**

| Variable / Proxy | Onset Time Bias Under Original Implementation / Years | Onset Time Bias Under Extended Implementation / Years |
|---|---|---|
| **Temperature** | -0.7 ± 0.2 | -1.0 ± 0.2 |
| **Precipitation** | -8.8 ± 0.7 (-7.1 ± 0.6) | -8.9 ± 0.6 (-6.0 ± 0.5) |
| **Sea Ice** | -0.9 ± 0.2 | -0.8 ± 0.2 |
| **NAO** | -1.5 ± 0.2 | -1.2 ± 0.2 |
| **AMOC** | -1.8 ± 0.4 | -2.1 ± 0.3 |

| δ¹⁸O | -7.6 ± 0.8 | -10.2 ± 0.9 |
|---|---|---|
| **Annual Layer Thickness** | -9.2 ± 0.5 | -6.0 ± 0.5 |
| **Na** | -6.8 ± 0.6 | -9.4 ± 0.6 |
| **Ca** | -9.7 ± 0.5 | -4.8 ± 0.4 |

*Additional Table 1: Onset Time Bias for the different model variables and ice-core proxies in both the original and extended implementations of the ramp-fitting method. The uncertainties here are the standard errors on the mean across 1000 synthetic transitions. Unlike in Table 3 of our original manuscript, the uncertainty in each individual event is now fully propagated, resulting in slightly larger uncertainties. These uncertainties reflect only the uncertainty in the bias given a particular set of transition parameters. They do not account for uncertainty in our estimates of those transition parameters. This will replace Table 3 in our revised manuscript. Note that we no longer equate certain variables and proxies.*

3. In Capron, the results of a single test of possible bias are given in the Methods section (and Fig. S2). Here, no significant bias was detected (the 50-year duration of the synthetic data was reconstructed as approximately 51 years by the method (and as an add-on to the previous point: the results in the S2 figure illustrate that the uncertainties are considerable and easily would include a bias of a few years).

Because the results differ, it would be interesting if the authors could discuss their analysis in the context of the published test. Maybe the way the test data were created could make a difference (also see next point)?

**This is an important point and we thank the referee for raising it. We will make sure to address this, as outlined below, in the discussion section of our revised manuscript.**

**Capron et al. test 20 realisations of a synthetic ramp, and find no significant evidence of a bias in either the transition midpoint or duration. Neither of these are directly comparable to the transition onset time, which we focus on as we feel it is more physically meaningful when trying to understand the progression of DO events. Nonetheless, if there were a bias in the onset time then Capron et al. would surely have seen this reflected in either the midpoint or duration, and they do not in fact see this. As the referee notes, the uncertainties shown in Supplementary Figure 2 of Capron et al. are large, and so could include a bias of a few years. Even so, the test conducted by Capron et al. would seem to rule out decadal-scale bias of the kind that we find in our study, at least for this particular combination of transition shape and noise.**

**We would therefore suggest that the transitions tested by Capron et al. happen to lie in a region of parameter space for which the bias is small, even at high levels of noise. When using the original Erhardt et al. implementation we find very little bias for synthetic transitions with no pre- or post-ramp slopes (see Additional Figure 2 on page 10 of this response). Although the shape of the deterministic ramps used by Capron et al. is not made clear, it is likely that they are flat before and after the ramp. If Capron et al. had instead included even very slight pre- and post- ramp slopes, as we believe are present in both the model and ice-core data, then we suggest that they may have found the same bias that we do. We thus feel that our results do not in any way contradict the lack of bias found by Capron et al. Instead, we demonstrate that their finding depends strongly on the unstated assumption that there are no pre- or post-ramp slopes.**

4. The authors do not describe in detail how they generate their synthetic test data, and when they estimate the bias, how many independent realizations are run (my experience is that especially for high noise levels, averaging over at least several tens of realizations is necessary). This information should be provided somewhere.

Also, and especially because the conclusions differ from those of earlier tests (e.g. those mentioned in Capron), the synthetic data series used here should be made available to allow others to make future tests against the exact same test data, thus reducing the risk that differences could be caused by implementation issues.

**This study makes use of around 200,000 synthetic transitions, and so we do not feel that it is practical to provide these. We will however provide the code which was used to create these synthetic transitions so that interested readers can gain a full understanding of how this was done. We will also add the following in Section 3.1:**

**"When not explicitly being varied, transitions parameters are fixed at the following standard values:**

- **Autocorrelation Time = 10 years for the case of decadal resolution and 1 year for the case of annual resolution.**
- **Greenland Stadial Slope = 0 Kiloyears$^{-1}$**
- **Greenland Interstadial Slope = -1 Kiloyears$^{-1}$. Note that the Greenland Interstadial Slope is always <= 0 in our sensitivity tests due to the sawtooth shape of DO events. In Figure 3 we plot the absolute value or magnitude of this slope, which is positive.**
- **Transition Duration = 50 years.**

**Each panel (a-h) in Figure 3 contains 100 unique sets of parameter values. For each unique set of parameter values, we average over 100 synthetic transitions in order to estimate the bias."**

**We will also ensure that it is clear from the caption of Figure 3 that parameters are kept fixed when not explicitly varied, and that we average over 100 synthetic transitions for each unique set of parameter values.**

**To more clearly explain the bias corrections in Section 3.3, we will include in the appendices all of the parameter values used for the "analogous" synthetic transitions to each model variable and ice core proxy. We already state in the caption of Table 3 that we take a mean across 1000 synthetic transitions to calculate each bias correction, but we will include this information in the main text as well.**

5. Related to the former point, one could suspect that especially when adding relatively high levels of autocorrelated noise, the transitions (i.e., ramps) of the synthetic data actually become more gradual (with longer durations and/or earlier onsets as a consequence). This is what was tested in Capron Fig. S2 for one scenario, but the results could be different for other parameters and other noise-to-signal levels (see PS below). If this indeed is the case, I would say that it changes the way we should look at this: it's not "just" the ramp fitting procedure that has an issue, but an inherent limitation related to the nature of climate data with autocorrelated noise (or data affected by autocorrelated short-term variability, if you like). I think this issue is related to what the authors are hinting at around line 226-228, but it could be expanded and made clearer.

**This is an interesting interpretation of our findings, but we do not fully agree. The synthetic transitions that we use for testing have a true duration that is known and unrelated to the degree of noise. Clearly, higher noise-to-signal ratios will lead to a higher degree of uncertainty in both the onset time and the duration. However, it is generally possible in statistics to have an estimator that remains unbiased even as the level of uncertainty increases. We therefore feel that one cannot conclude from our work that bias of this sort is an inherent limitation of climate data with autocorrelated noise - it remains possible that there exists an unbiased method to address this problem, and that we simply haven't been able to identify it.**

**In light of this, we will rephrase lines 226-228 to state that "we have been unable to identify an unbiased method", instead of the current statement that "there is no unbiased [method]". We will also include further consideration of this interpretation in the discussion.**

**It also might actually be that there are no conflicting views here, and that the referee simply chose different words than we would choose to express very similar thoughts. We agree that given the current methods, the autocorrelation of the noise in climate data ultimately limits this sort of analysis. However, we would still ascribe the bias to the method and not to the data.**

6. Somewhat along the same lines: It is not surprising in itself that high noise levels lead to longer transitions: As the noise level increases, (probably all) ramp detection methods will have a harder time reconstructing the (no-noise) ramp duration, and there will therefore be a larger spread in the detected ramp durations. If the noise is large enough, the range of ramp durations becomes asymmetric because the ramp durations by design are positive. An example with a synthetic ramp duration of 50 years: For high noise levels, the distribution of accepted solutions produced by the method will include values that are larger than 100 years, but by design none which are smaller than zero. The mean of the distribution will be biased towards larger durations and lead to earlier onset points (and/or later ramp end points).

In the test described in Capron, this effect of asymmetric distributions is observed although it does not lead to a bias in transition duration similar to the one reported in the current manuscript (see PS below). However, by using the median rather than the mean, the effect on the result can be reduced. This highlights two points. Firstly, using the median may be more appropriate than the mean. This should be discussed and tested explicitly. Secondly, this is not really an issue with the ramp fitting method, but a consequence of noisy ramped data. The observation is consistent with Fig. 3h which shows that there is largest bias towards early onsets for the most noisy and fastest transitions. I think it would strengthen the manuscript a lot if the authors could go a bit deeper into discussing the issues raised in point 5 and 6.

If they agree with these points, I encourage them to think about how to phrase their results in a way that better acknowledges that it's not "just" a methodological problem with ramp fitting, but an inherent limitation that both is related to the method and to the nature of autocorrelated ramped data and/or that (some of) their bias comes from the way the synthetic data are created.

The conclusion of Capron already discusses some of these issues (and tries to separate those that originate from the complexities of the climate system and those that are more related to the noise and method). It would be good if the authors spend a bit more space on discussing how their results confirm, refine (or even contradict) the ideas presented there, and separate between the different types of difficulties.

**The referee is correct to state that for high enough levels of noise, the distributions will become asymmetric because the duration must be greater than 0. This may partially explain the bias towards early transition onsets and longer durations. However, this does not explain why the other transition**

parameters beside the noise / signal ratio also impact the bias, or why we sometimes see a bias towards later transitions (and so shorter durations), e.g. Figure 3h for true transition duration > 80 years. We therefore feel that this is certainly not the whole picture.

Our key aim is to assess the bias affecting calculations of leads and lags. It is not obvious to us whether the posterior mean or posterior median is more appropriate for this purpose, but we agree that using the median may reduce the apparent bias. We have tested this, and we show below (Additional Figure 3) a figure that is equivalent to Figure 3 in our original manuscript but using the median as suggested. We do, as the referee suggests, observe a slight reduction in the magnitude of the bias, but this is a relatively small change that we feel does not impact any of our conclusions. Nonetheless, we will include a supplementary figure showing this and be sure to discuss it in our revised manuscript.

[Figure]

*Additional Figure 3: The bias in the median onset time, using the extended ramp fitting method. We see very similar results to the bias in the mean onset time (Figure 3 of original manuscript), with only a slight reduction in the size of the bias. This will be included as a figure in the appendices of our revised manuscript.*

Finally, we feel that the distinction between an issue with the ramp fitting method and an issue with noisy data is somewhat unclear, being that the ramp fitting method is inherently designed to work with noisy data. Furthermore, the synthetic data that we create assumes the same underlying transition model as the ramp fitting method itself, and so there shouldn't be any issues arising because of the synthetic data generation. In our view, it remains unclear whether this is purely a methodological problem or in fact an inherent limitation of dealing with noisy climate data, as the referee suggests. This is

**ultimately a question of whether or not there exists a method that can produce an unbiased estimate of the timing and duration of a transition in noisy, autocorrelated data. As stated previously, we do not know of any such method, but equally we do not feel confident in stating that such a method does not exist.**

**These suggestions and interpretations put forward by the referee are intriguing, and we will make sure to consider them in the discussion section of our revised manuscript.**

[PS on the test from Capron Fig. S2: While writing the review, and with inspiration from the manuscript and some tests performed by a student in Copenhagen last year (not published), I first ran some more realizations of the same test as in Capron: The mean reconstructed ramp duration was 51.9 years, and the median 51.5 years.This would probably correspond to an early ramp onset bias of ~1 year, considerably less than reported in the manuscript at hand, and well within any reasonable uncertainty estimate based on the width of the distributions. Reducing the noise level to half led to no significant bias (median 49.6 years, mean 50.2 years). Then I tried with double noise level (stadial noise level is the default in this test, which is already a relatively high noise level) and observed a more pronounced bias in transition length, which was also larger when using the mean rather than median. I did not check this through well enough to share the results, so I provide only this indication. I think this supports the idea mentioned in point 5-6. It also indicates that in Capron (at least for Calcium and in the absence of significant slopes of the pre- and post-transition values), bias does not play any important role for the results.]

**We thank the referee for making the effort to perform tests of their own in order to help refine our study. As mentioned in our response to point 3 above, when using both synthetic transitions and a ramp fitting method that do not include slopes, we find a weak bias, even at relatively high levels of noise. However, when we account for such slopes, we find that both implementations (with and without slopes) of the ramp fitting method that we considered do in fact have meaningful bias for all of the ice core proxies considered.**

On a more technical note, I assume that the authors have tested their implementation (including generation of synthetic data). It would be nice with a summary of the tests: Does the method faithfully identify the properties of the synthetic data (including the autocorrelation time) for low noise levels? Figure 3 goes some way in demonstrating this, but I think a bit more details and a dedicated section would be appropriate (possibly in a supplement).

**We present here the results of some low noise tests. The noise / signal ratio is fixed at 0.01, and unless explicitly varied the other parameters are fixed at their standard values as given on page 12 of this response. We see that**

**autocorrelation time is slightly overestimated, and that large negative slopes in the pre-ramp stadial are underestimated, although we rarely see such large negative pre-ramp slopes in either the model or ice-core data. These are limitations that need to be discussed, and we thank the referee for raising this issue. We will include Additional Figure 4 (below) in an appendix and make sure to mention this issue in the discussion section of our revised manuscript.**

[Figure]

*Additional Figure 4: Tests of the extended ramp fitting method at low noise / signal. This will be included in the appendices of our revised manuscript.*

Finally, I think the title is not fully aligned with the content and conclusions. The following line from the conclusion "Despite the bias that we have uncovered here, we nevertheless suggest that this Bayesian ramp fitting method remains the best approach to understanding the temporal phasing of DO events" does not really work well together with the title "The Temporal Phasing of Rapid Dansgaard–Oeschger Warming Events Cannot Be Reliably Determined".

**We agree that the title and conclusions do not fully align as currently stated. In line with our response to the referee's point 1, we will change the highlighted line of the conclusion to something along the following lines:**

**"The Bayesian ramp fitting method considered in this study remains a powerful tool for investigating individual abrupt transitions, as the bias that we find is small relative to the uncertainty of individual events. When calculating leads and lags, however, the bias is large relative to the uncertainty and so is likely to lead to false conclusions."**

In summary, I think the manuscript is appropriate for publication in Climate of the Past, but that additions and revisions are needed to fully document the analysis and make it clearer to which degree and under which circumstances the extended model with stadial slopes represents an improvement in the context of the data at hand, discuss in more depth whether the bias is a methodological problem or arise from the inherent difficulties of identifying ramps in noisy data, and assess whether the proposed bias will influence earlier conclusions based on the ramp fitting methodology.

Review by Sune O. Rasmussen, Jan 10 th , 2024. Line-by-line comments

29: "However, as of yet it has not been possible to conclusively identify a cause for these rapid Dansgaard–Oeschger warming events" could be rephrased. The exact mechanisms leading to the described changes are not understood but the AMOC state change is generally accepted as the main cause of most of the observed changes in the climate system.

**We will change the word "cause" to "trigger" and cite Capron et al., as they state in their opening paragraph that "no consensus exists yet to explain what triggers the abrupt warmings".**

34-39: The papers of Adolphi and Corrick align records from ice cores and speleothems and have to deal with non-climatic dating offsets between different dating approaches. The WAIS and Svensson papers use synchronized ice-core records. Because the uncertainties (in terms of both magnitude and nature) are quite different, it would be better to describe them independently. Bias in event detection of ~20 years are only marginally relevant in this context, which is not the case for the methods described in line 39-41.

**We agree with the referee that our work is largely irrelevant to the papers cited in these lines, and vice versa. We will remove all discussion of inter-archive studies and focus solely on the studies that compare different proxies within ice cores.**

39: (Buizert et al., a) should be cited as "WAIS Divide Project Members".

**We will do so.**

47-48 and 51-52: Elaborate or remove. The current sentence is vague. How do they contradict each other (given the error bars)? Consider if it would make the discussion clearer to include a discussion of the difference of absolute vs. relative dating errors. An ice-core timescale can build up significant bias over long stretches, and the published maximum error of GICC05 is therefore large in MIS 3.

**As stated above, we will remove all discussion of these inter-archive studies.**

58-63: I believe the papers came out in the reverse order: Capron et al. was accepted while the manuscript of Riechers and Boers was in the discussion phase. Please adjust wording accordingly, and in particular, be careful to cite the conclusions of Capron et al. correctly. I do not find a statement in Capron et al. to support the claim "they also concluded that applying a method of the type developed by Erhardt et al. (2019) to ice core data may not allow the identification of a unique order". In contrast, Capron et al. acknowledges that the Erhardt et al. approach of stacking may be appropriate under certain assumptions. The lack of a

well-established order of events which is similar between events could both be due to noise / internal variability making detection difficult, but also that such a "standard sequence of events" may not exist.

**We thank the referee for pointing out that we had the papers in the wrong order. We also agree that our original formulation in this section was imprecise, and we will change it in the revised manuscript. Our statement unintentionally suggests that it is a failure of the method that leads to no systematic pattern of leads and lags being found by Capron et al. However, as the referee helpfully notes, Capron et al. actually attribute this finding to either i) a large degree of climate variability obscuring a common mechanism or ii) different mechanisms for different DO events. We will revise this paragraph so that the order of the papers is correct and so that the conclusions of Capron et al. are more accurately summarised.**

72: Maybe reconsider use of "cause" (as above).

**We will change "cause" to "trigger" as above.**

90: Proofreading needed.

**We thank the referee for bringing to our attention the poor grammar in this line, and we will correct it.**
93-94: Vague. Refer to Vettoretti et al. 2022 Fig. 2a or describe how your analysis is different.

**We will remove these lines as they are not necessary.**

110: It is not clear what "parameter ranges" refer to.

**Thank you for raising this lack of clarity. We meant the ranges of the ramp parameters that one obtains when applying the Bayesian ramp-fit to the annual simulation data. We will clarify this statements by writing**

> **"Although we do not have a sufficient number of annually-resolved abrupt warming events from which to assess potential systematic time lags, an application of the Bayesian ramp-fit allows us to gain a sense of the ranges of the ramp parameters. This in turn allows us to gain insight into how increased resolution impacts the bias in the ramp fitting method."**

**instead of**

>**"Although not sufficient to assess potential systematic time lags, these roughly indicate the possible parameter ranges at this higher resolution, and so allow us to gain an insight into how increased resolution impacts the bias in the ramp fitting method."**

117: Describe somewhere (here, in Methods, or supplementary material) what "for which the method was successful" means.

**We agree that this should indeed be clarified. We mean by this that Erhardt et al. were successful in applying their method. We will change the sentence that starts on line 116 to the following:**

**"Whilst Erhardt et al. considered a larger set of transitions, these 16 are the only ones for which they were successfully able to apply their method to all four proxies. Although not made explicit, this was likely judged based on the rejection fraction of the MCMC sampler."**

119-120: Add charge to ion or spell out full names.

**We will add charges to the ions.**

120: Note that Capron et al. did not make a one-to-one "identification" of proxies and model data extracts: "Over each D-O-like transition, we extract the time series of four climatic measures from the model on the assumption that they reflect some of the same elements of the climate system as our ice-core proxy data". Revise accordingly.

**Upon reflection, we feel that the identification of proxies with model variables is unimportant to the results of our study, and so for the sake of simplicity we will remove any mention of such a connection, including by reformatting Table 3.**

122: Revise syntax.

**As we will no longer equate certain proxies and model variables, we will remove this discussion.**

137: It would be prettier if x(t) was defined for t_0 and t_1 as well (as in appendix).

**Agreed. We will make Eq.(1) and Eq.(A1) consistent in the revised manuscript.**

154: This is a critical assumption which is also employed by Erhardt and discussed in Capron. I encourage the authors to consider and discuss the implications of the assumption (e.g. here or in section 2.5).

**Indeed, this assumption has far-reaching consequences for the physical interpretation of DO events. We agree with the referee that these should be made explicit. We will therefore add the following sentences:**

> **"This view is only meaningful if one accepts the 'one-mechanism hypothesis', which states that all DO events follow the same mechanism and that differences in their expression are exclusively due to internal climate variability. While this hypothesis may not be generally accepted for the real-world's DO events, we believe that the regularity of stadial-interstadial cycle seen in the DO simulations provides a strong argument that at least the simulated DO events were all governed by the same physics. We further emphasise that a statistical assessment in terms of hypothesis tests relies on the 'one-mechanism' hypothesis and would be meaningless if individual DO events were caused by different physical drivers."**

176: It is more common to use the SNR and not the "Noise / Signal ratio". I recommend using the more standard SNR unless there are good reasons for why not to do this.

**We recognise it is more standard to use SNR. In our case, the signal is fixed whilst we are varying the level of noise linearly. If we were to plot the bias in terms of SNR, we would therefore have non-equidistant data points. Overall, we feel that the Noise / Signal ratio both more directly reflects our approach (with signal fixed and noise varied) and provides greater ease of understanding plots such as Figure 3. On that basis we intend to stick with our current approach.**

178: Give more details on how the synthetic data were generated in order to realistically mimic true data.

**We will alter the line 178 to the following:**

**"To quantify the strength of the bias, we construct synthetic transitions by the addition of randomly generated AR(1) noise to a piecewise linear ramp, exactly as in the extended probabilistic transition model (Appendix A). Our intention is not to mimic true data as realistically as possible but instead to create data to which the ramp fitting method should be perfectly suited."**

192: (also see main point raised above). It is hardly surprising that adding a lot of noise to a steep transition tends to smoothen out the transition found by the ramp fitting method. In order to judge how serious this problem is, I suggest adding the following pieces of information: - How does the calculated UMOTE compare to the

distributions of onset times? I.e., are the confidence intervals wide enough to include the bias?

**As in our response to the main point, we feel that the important comparison is not to the credible intervals of individual transition onsets, but instead to the uncertainty in calculated leads and lags derived from stacking many events.**

It would be informative to add some estimates of noise/signal for the actual data used by Erhardt at the start of this section, so the reader gets a feeling for how the observed biases compare to what would be expected in real data.

Same for the stadial and interstadial slopes: What are the realistic values for the Erhardt data sets?

**We will add shapes representing the mean values of the parameters in the Erhardt et al. NGRIP data to Figure 3, as we already have for the CCSM4 model variables.**

200-201: (Optional). Maybe comment on how an autocorrelation time on decadal scale could occur in the climate system?

202: Figure 3: Mention that the scales are not the same for a-d end e-h, respectively.

**Thank you. We will add this remark to the figure caption.**

I guess that noise/signal does not mean the same for annual and decadal data. Is there a way to compare the two more directly? For example for a certain annually resolved data set with a given noise/signal value, what is the corresponding noise/signal value for the same data set averaged to decadal resolution? That would elucidate the dependence of resolution more clearly.

**Unfortunately, it is not generally possible to directly compare noise / signal at different resolutions. Two annually resolved time series with the same noise / signal but different power spectra would have very different noise / signal ratios when down-sampled to decadal resolution.**

**As an example of this, we include below Additional Figure 5 showing the mean estimated noise parameters at both annual and decadal resolution for the "redder noise" and "whiter noise" synthetic transitions discussed in section 3.2. At annual resolution, the "redder noise" case has a lower noise / signal than the "whiter noise" case. However, due to the differing autocorrelation times, when we down-sample to decadal resolution we find the opposite. We will address this point in Section 3.2 when demonstrating that changing resolution does not change the bias.**

[Figure]

***Additional Figure 5: The noise parameters of "whiter" and "redder" noise cases at (a) annual resolution and (b) after being down-sampled to decadal resolution. Also included are the dependencies of the bias on the two noise parameters, with other transition parameters fixed, showing that the bias is unchanged by down-sampling to a lower resolution. This is shown here solely to address the referee's comment, and will not feature in our revised manuscript.***

206: For the tests in C1-C4: Are these tests run on the same synthetic data? How exactly were the synthetic data generated, and what value of the autocorrelation length was applied? Synthetic data with characteristics spanning a range of parameter values are tested, but without describing how the ranges compare to those of the real data, the results are hard to interpret in a useful way. For the test on Figure C2: Is this relevant given that Capron did not use data with decadal resolution? Or am I misunderstanding the caption: "transitions with decadal resolution"?

**We apologise for not sufficiently explaining how this data was generated. As with all the synthetic data in this study, we add AR(1) noise to a deterministic linear ramp. When not explicitly being varied, all parameters are fixed at standard values as given on page 12 of this response. For the autocorrelation time, the standard values are 1 year for annual resolution and 10 years for decadal resolution. The tests in figures C1-C4 are not run on the exact same synthetic data, but each is run on a large set of synthetic data that is generated in exactly the same manner, and so any slight variations due to the random noise generation should be averaged out. Specifically, each data point represents the average over 100 synthetic transitions, as in Figure 3.**

**In Figure C2 we briefly assess the impact of using an alternative set of priors as suggested by Capron et al. Although annual resolution would be a closer reflection of the data used by Capron et al., for our purposes of simply checking that the bias is still present we do not feel that this is overly important. This is especially as we demonstrate elsewhere that changing resolution does not change the bias.**

214-216: Please provide the details of the autocorrelation range observed in the model data. What noise-to-signal characteristics were chosen? It's fine to have the results illustrated in Fig. 3, but they should also be tabulated.

**We will include this information in a table in the appendix.**

225: It would be clearer to integrate the section starting at 230 with this section.

**We agree. We will move the statement:**

> **"The "analogous" transitions have noise and slope parameters that are representative of the appropriate variable, whilst retaining a fixed transition onset time and duration (50 years). We could instead use different, representative transition durations for each model variable. However, we have established that there is no unbiased means by which to estimate these transition durations, and so we could not guarantee that this would be any more accurate."**

**behind the statement**

> **"To calculate representative parameter values we take both the uncertainty-mean and the event-mean over the corresponding marginal posterior distributions obtained by applying the ramp fitting method to the 19 available transitions. For each variable or proxy this gives a single value for each transition parameter (shown in Fig. 3), which we use to create the "analogous" transitions."**

241: Please explain what you mean by "reliability".

**We agree that this is indeed not clear. We will reformulate the statement as follows:**

> **"We find that this has a negative impact on the success rate of the MCMC sampler, in that there are many more cases where the sampler fails to converge, and so we do not attempt to use this implementation to assess the timing. Instead, it simply provides a**

**means of calculating appropriate parameter values to use for the
synthetic transitions that are "analogous" to precipitation."**

245: Please provide table with the characteristics of the synthetic data that resemble
the ice-core data sets.

**We will provide a table with this information in an appendix.**

265: (Figure 5). It is not clear whether the Calcium data have also been corrected for
bias and how this factors into Fig. 5: Is the zero defined from corrected or
uncorrected Calcium data ramps? Please provide the details.

**Yes, indeed. That is an important bit of information missing here, and we will
include it in the caption of Figure 5 and likewise in Figure 4. The transparent
violin plots display lag distributions after bias-correcting both the reference
variable and the target variable.**

303: I don't see why this is "another issue". It is quite clear that observing a too long
duration is equivalent to the combination of a too early onset and a late end point,
and as stated in the introduction, I would claim that both effects are likely to occur
when you turn up the noise.

**We believe that there is a misunderstanding here. The term 'another issue'
expresses that the bias correction suffers from different issues. We are at this
point not talking about 'issues' in the estimation procedure. We have shown
that the bias depends on the transition duration. So if we wanted to correct the
bias in the application of the method, we would need an accurate estimate of
the duration of the transition under study. However, the biases in the method
themselves make such an estimate impossible, thus limiting the ability to use
bias correction.**

**We hope that slightly rephrasing the sentence as follows will make things
clearer:**

> **"Another issue limiting the possibility for bias correction is our
> inability to estimate the transition duration in an accurate manner."**

312-15: Section 3.2 is based on CCSM4-like data, and it is not clear if the same
result holds for paleoclimate data.

**The statement is indeed misleading. However, Section 3.2 is based on
synthetic data and the conclusion that increasing resolution does not remove
the bias is thus also equally valid for paleoclimate data. We will slightly modify
the statement:**

> **"But, as Section 3.2 shows, the temporal resolution of DO event paleo-data has no impact on the bias in the ramp fitting method."**

**and write in a revised version**

> **"But, as Section 3.2 shows, the temporal resolution of the data comprising a transition has no impact on the bias in the ramp fitting method."**

Please clarify/rephrase. Throughout: Please add hyphens consistently to "sea-ice extent", "ice-core records" and other similar compound adjectives. It's somewhat a matter of style, but generally increases readability.

**Thank you for this suggestion to help increase readability. We will adopt this style throughout.**

**Referee #3**

**The authors would like to thank the referee for their extremely thorough comments and suggestions. The referee's comments are normal text, and our responses follow in bold.**

Review of Slattery et al.
In their manuscript Slattery et al. present an analysis of the performance of a modified ramp-fitting method for the determination of the onset of stadial to interstadial transitions and apply this modified method to decadal resolution model data.
Overall, the topic of the manuscript is of interest to the wider paleo-climate community and especially the detailed analysis of interstadial onsets in model output is interesting and a good fit for CP However, there are some larger issues with the manuscript concerning the conceptual background as well as to the presentation of the results that need to be worked out before the manuscript should be considered for publication.

General remarks
The authors extent the ramp-fitting method of Erhardt et al. (2019) (E19) by including two extra parameters for the pre- and post-transition slope of the data. This is necessary to better describe the model-output that the authors want to apply the method to, especially the AMOC strength. The authors do not show, if the inclusion of the slopes into the model is necessary for all their data and also do not provide any evidence that the inclusion of the slopes decreases the uncertainty of the estimates for the relevant parameters.

**We agree that we have not made it sufficiently clear that the inclusion of slopes is either necessary or beneficial. We will include an additional appendix in our revised manuscript that contains detailed comparison of the original and extended implementations of the ramp fitting method. As part of this, we will give the average and ranges of the different parameters (including slopes) for all of the model variables and ice core proxies. We hope that this will demonstrate why including slopes is necessary. We will also include in this appendix Additional Figure 6, shown below, which demonstrates that the inclusion of slopes results in a reduction in the uncertainty for cases where slopes are in fact present, but an increase in uncertainty when no slopes are present. We thank the referee for prompting us to consider this in more detail.**

One would also hope, that the addition of two extra parameters to the model would make the model consistently outperform the original E19 version. Judging from the figures in the appendix this does not necessarily be the case. However, due to presentation of the results that is difficult to judge.

**We agree that this is an important point to discuss, however we disagree that the addition of two parameters should consistently improve performance. For cases where slopes are present then clearly this is the case. However, when no slopes are present, the addition of two unnecessary parameters may plausibly lead to worse performance. As seen in Additional Figure 6 below, this is exactly what we observe.**

**For regions of parameter space with zero or only slight slopes during the preceding stadial and following interstadial (e.g. the middle band of panels a & b, and the bottom region of panels c & d), the original method outperforms the extended method, in the sense of having lower uncertainty in the onset time. However, for regions of parameter space with more substantial slopes the extended method performs better than the original.**

**We will include this discussion and Additional Figure 6 in our revised manuscript.**

[Figure]

***Additional Figure 6: The uncertainty in the onset time, depending on the noise noise / signal and slopes, for both the original (a,b) and extended (c,d) ramp fitting methods. The uncertainty is defined as the width of the 90% credible interval for the onset time. This will be included in the appendices of our revised manuscript.***

Before continuing, a note here on terminology: The authors chose to coin a new term for the average value of the posterior samples. This is unfortunate, as there already is a term that is used for this: posterior mean or more specifically the marginal posterior mean. This is the term that I will use and that I strongly urge the authors to use, too.

**We will adopt this suggestion.**

In an extensive sensitivity study, the authors test if the posterior mean of the transition onset is on average an unbiased estimator of the onset of the transition. Most, if not all, of the content of the paper hinges on the results of this sensitivity study. Despite that, the authors

never explicitly state the exact focus of the sensitivity study and the implications that come with this choice. However, that exact focus of the tests is a big potential issue: In the sensitivity tests, the marginal posterior mean does not need to equal to the known parameter value and same is true for any other point estimate from the full marginal posterior. If the posterior is skewed or multi-modal then any point estimate will grossly misrepresent its true location and disregard its shape. Especially if skewness of the marginal posterior is systematically induced due to the noise or due to trends around the transition, it can introduce a systematic bias into the marginal posterior mean.

This can for example be seen in Figure 2 of E19, where the two marginal posterior distributions for the start and the end of the transition show a clear indication of skewness and in the case of the onset maybe even bi-modality. The same argument is of course true for the length of the transition and also extends to all other estimated parameters.

Sune Olander Rasmussen (SOR) raised a not dissimilar issue in his review: By using the marginal posterior mean the authors disregard the sometimes large uncertainty in the estimated parameters that is carried in the complete marginal posterior distribution. As pointed out by SOR, the width of the marginal posterior distributions can easily be larger than the reported bias of the posterior mean. From my point of view, this sole, narrow focus on the marginal posterior means is in stark contrast to the strong general statements on applicability of the method throughout the paper, especially because this tight focus is not explicitly communicated to the reader.

**The referee is correct in that we put strong emphasis on the posterior mean. The point they make, that this is not necessarily a good point estimate of the true value, is also valid and important. However, we remain convinced that the bias of posterior mean with respect to the true value is a good indication of how lead and lag estimates derived from a set of events are corrupted by the inaccuracies of the method.**

**The key quantity we would like to probe is the degree of bias that will be propagated to calculations of leads and lags. As there are different proposed methods of calculating such leads and lags (e.g. Erhardt et al. 2019, Riechers & Boers 2021), it is not immediately obvious whether the posterior mean or posterior median is a better indicator of this bias. We will include Additional Figure 3 (page 15 of this response) in the appendices of our revised manuscript to demonstrate that we find very similar results when considering the median as we do when considering the mean. Given this, we feel comfortable in stating that the centre of the probability distribution is shifted away from the true value, which is what we term a bias, and that this will necessarily be propagated to the calculation of leads and lags.**

**We apologise for failing to address the relative size of the bias and the width of the marginal posterior distribution. As also mentioned in our response to Sune Olander Rasmussen (Referee #2), we agree that our bias is relatively unimportant when considering individual events due to the large uncertainty in the timing. However, when calculating leads and lags this uncertainty is reduced by stacking many events, and so our bias then becomes extremely important. We will ensure that this distinction is clearly communicated throughout our revised manuscript.**

In addition to the conceptual shortcomings of the sensitivity study, the presentation of its results in Figure 3 and in the appendices is also not ideal. If I guess correctly (there is nothing written in the manuscript), the authors average over the variables that they do now show to present the data in a reduced number of dimensions. The main issue with this approach is, that it hides any possible inter-dependencies between variables and does not allow for clear separation of the effect of the single variables. I think this is a missed opportunity and might be potentially misleading, depending on the parameter interdependencies.

**We thank the referee for pointing out that we have failed to sufficiently describe how the synthetic data in Figure 3 is generated. We will add the following in our revised manuscript:**

**"When not explicitly being varied, transitions parameters are fixed at the following standard values:**

- **Autocorrelation Time = 10 years for the case of decadal resolution and 1 year for the case of annual resolution.**
- **Greenland Stadial Slope = 0 Kiloyears$^{-1}$**
- **Greenland Interstadial Slope = -1 Kiloyears$^{-1}$. Note that the Greenland Interstadial Slope is always <= 0 in our sensitivity tests due to the sawtooth shape of DO events. In Figure 3 we plot the absolute value or magnitude of this slope, which is positive.**
- **Transition Duration = 50 years.**

**Each panel (a-h) in Figure 3 contains 100 unique sets of parameter values. For each unique set of parameter values, we average over 100 synthetic transitions in order to estimate the bias."**

**As this hopefully makes clear, we have not averaged over any other parameters. However, there are certainly still possible interdependencies that are not displayed in Figure 3. Unfortunately, we can not think of any sensible means by which to present the full interdependencies of the bias on the five different transition parameters. We feel that the current presentation sufficiently captures the key dependencies, which are on the noise / signal ratio and its interaction with the other four parameters.**

The authors apply their modified ramp model to climate model output of DO-type transitions that is available in decadal averaged output. For this the authors chose to stay with an AR(1) noise term for the application of the ramp model. I am, like SOR, a bit puzzled by the long autocorrelation times in Fig 3e. Of all the parameters only the ocean circulation would likely have long-term memory, of all the other parameters, I would assume that decadal averages would mostly eliminate the autocorrelation.

I would ask the authors to provide a more detailed explanation of this phenomenon and a discussion of the influence of the long autocorrelation times on the timing estimates. Generally the autocorrelation of the residuals in the model is in competition with the transition ramp itself, for lack of a better term. If the model fails to describe the noise of the data accurately then the model will give unreliable results. Long autocorrelation times can smear out the transition for sure leading to larger uncertainties and maybe even "biases"

towards earlier or later transitions. This can make the choice of the noise model and the choice of priors for its parameters very important. In the case of the presented study, the authors explicitly excludes the case for non-autocorrelated data and put, in comparison to E19, larger prior probability on long autocorrelation times.

Especially the former might be a serious issue in case of the decadal averaged data from the model and merits a thorough discussion.

**This is an interesting point, and we thank the referee for prompting us to consider it. We agree that the long autocorrelation times of the decadal resolution model output data are rather confounding. After some investigation, we believe that this is an inherent result of approximating climate variability as an AR(1) noise process, rather than being because of our choices of priors.**

[Figure]

*Additional Figure 7: The posterior mean values of the autocorrelation time tau and the equivalent autocorrelation coefficient alpha plotted against temporal resolution for the 16 studied DO events in the NGRIP δ¹⁸O record. This is included to address the referee's comment, and will not feature in our revised manuscript.*

**To demonstrate this, we consider the NGRIP $\delta^{18}$O data, which ranges in temporal resolution from 4 to 7 years over the different DO events going back to 60ka. When estimated using the original Erhardt et al. implementation, we find a significant positive correlation (p<0.05 assessed using a standard linear regression) between the autocorrelation time $\tau$ and the temporal resolution. However, we find no such correlation between the temporal resolution and $\alpha = e^{\frac{-dt}{\tau}}$, where $dt$ is the temporal**

**resolution. This is shown in Additional Figure 7. This pattern whereby there is a significant correlation between resolution and autocorrelation time, but not between resolution and alpha, is repeated for all four considered proxies and using both the original and extended methods. This is therefore a robust finding that is not due to the quirks of any particular proxy or of any particular implementation of the ramp fitting method.**

**One would certainly not expect to find any meaningful autocorrelation at a lag of e.g. 10 years in annually resolved climate data. However, the situation here is subtly different: We are in fact saying that there is autocorrelation between the means in successive decades, e.g. of years 1-10, 11-20, 21-30, etc. Taking decadal means in this way largely eliminates the annual variability, and so the AR(1) noise mostly represents multi-annual or decadal modes of variability, which we fully expect to have autocorrelation times of a decade or more. Whilst these long autocorrelation times are certainly initially surprising, we therefore do not feel they are indicative of an issue with our approach.**

**The referee is correct to state that our choice of prior puts larger probabilities on longer autocorrelation times, in comparison to E19, and that we explicitly exclude the possibility of zero autocorrelation. However, in comparison to E19 we also put a greater prior probability on short autocorrelation times (between 0 and 10 years), and there is no difference between the cases of an infinitesimally small autocorrelation time and zero autocorrelation. Ultimately, Figure 3e shows that there is no significant link between the autocorrelation time and the bias for the case of decadal resolution, so even if our implementation does overestimate the autocorrelation time this cannot be the source of the bias.**

To overcome the bias that the authors discovered, they use "analogous" data to estimate the bias and later correct for that. Generally the authors do not provide much detail about the exact parameters that they use to generate the "analogous" datasets that they use to make the bias correction for the different parameters.

It would be great if the authors could provide more detailed information about the parameters that they chose such that the results are later reproducible. Their statements about the fact that they cannot reliably estimate the onset or the length of the transition given their methods makes me also wonder how they estimated all the parameters that they need for the "analogue" data.

Did they use a different method or can they use the ramp-fitting method to estimate all other parameters needed for the surrogate data reliably? Especially in light of the statement by the authors that the relationship of the bias to the ramp parameters is complicated it would seem to be necessary to estimate this parameters 2very precisely or carry out the bias calculations for a range of possible parameters. I strongly urge the authors to investigate this in more detail before making definitive claims about a bias.

**We apologise for failing to provide this information. We will include a table in the appendices of our revised manuscript which provides all of the parameters for the "analogous" synthetic transitions. We will also present the range of parameters seen**

**for each variable and proxy, to justify our choice of ranges in Figure 3. Other than the duration, which is simply fixed at a plausible value, all of the other parameters are estimated by taking the mean over all available events of the posterior means given by our extended ramp fitting method. We will make this more explicit in our revised manuscript.**

**We considered estimating the bias for a range of possible parameters, as the referee suggests, but in the end we feel that it is sufficient to demonstrate that there is a significant bias given a plausible set of parameters for each variable or proxy. The bias will in fact be different for every individual DO event, but to estimate this bias for every individual event, for each variable and proxy, would add massively to the complexity of our analysis whilst, in our view, not meaningfully changing our findings.**

The proposed bias correction also carries additional uncertainty with it, how do the authors propagate that to the estimates later?

**We do not propagate the uncertainty in the parameters for each set of "analogous" transitions to the estimates of the bias. We apologise for not making this clear and will do so in our revised manuscript. We will add a line to the caption of Table 3 stating:**

**"The uncertainties provided here reflect the uncertainty in the bias given a particular set of transition parameters. There is additional, unquantified uncertainty due to our imperfect knowledge of the true transition parameters."**

In the analysis of the modelled precipitation data, the authors state, that the noise characteristics of the data change over the transition. The authors somewhat disregard this observation with the notion that they failed to include this in the model. However, the results of the analysis are dependent on the noise model adequately describing the data. This is especially worrisome as the precipitation time series exhibits the biggest lead over the temperature in the model data. The fact, that the authors state that inclusion of changing noise characteristic over transition negatively impacts the reliability of the method might hint at a deeper problem. This makes me question the results that they present and the applicability of the estimates for the generation of the "analogous" data and thus the bias correction that the authors apply.

**We agree that differing noise characteristics between the stadial and interstadial present a problem. We also agree that our estimate of the bias in precipitation is definitely flawed, but we make this clear and it is the best estimate we are able to make. In our view, it is sufficient to demonstrate that the lead of precipitation over temperature could plausibly be the result of bias. Although we certainly cannot prove that this is the case, it also cannot be ruled out, and so the apparent lead is therefore not reliable. One does not need to know the exact size of the bias to demonstrate that such a bias exists.**

Specific remarks

58–63
Order of publications is wrong.

**Apologies, we will correct this.**

Table 1
This suggests a connection between Ca and the NAO, please justify or remove

**We will remove any suggestion of connections between model output variables and ice core proxies, as these are not important for our study.**

125ff Focus of the work: If that is the case, please rephrase the introduction and the abstract accordingly. In this case you can also remove all the model data. Or spend some time discussing the data and its interpretation. There is no need for statistics without a clear question to answer.

**As also mentioned in our response to the other referees, we will no longer make any identification between proxies and climate variables. The focus of our work is indeed on the robustness of the method, but this can only be judged with respect to the kind of data that the method is designed to be applied to. We feel that the model data is a useful example of a potential application that allows us to demonstrate how the bias could lead to a false conclusion.**

138ff too. This is a wrong statement. If that were true then OLS-regression would be probabilistic,

**Our apologies for this misleading statement. We will instead write the following:**

> **"The addition of an AR(1) noise process makes the model stochastic and, upon the introduction of a convenient prior distribution directly yields a Bayesian posterior distribution for the model parameters $\{t_0, t_1, x_0, x_1, \alpha, \sigma\}$, where $\alpha$ and $\sigma$ define the autocorrelation and the amplitude of the noise respectively."**

143ff Please show proof of this.

**For an arbitrarily chosen abrupt warming event from the climate model simulations, we applied both the original and extended methods to the AMOC time series using 25 different search windows. The standard deviation (over the different search windows) of the posterior mean onset time was 16.0 years using the original method, but this was reduced to just 4.3 years when using the extended method. We will ensure that this is clearly stated in our revised manuscript.**

149 Remove reference to Erhardt et al., as they do not use any frequentist methods.

**We will remove this reference.**

169ff Nomenclature: There are some already established terms that the authors should use as much as possible instead of inventing new ones: the average of the posterior distribution is the posterior average or posterior mean.

**We thank the referee for pointing this out and will adopt their suggested terms.**

Fig 3 The presentation of the results is unclear. The authors do not state what they do with the other parameters that were varied. I assume that they just averaged over all the other parameters that are not shown in the figure.

However this approach will hide any co- dependencies of the bias between different parameters.

**We will make clear in our revised manuscript that parameters were fixed at standard values unless explicitly varied.**

213ff : Please specify how you set these two sets of data up. I would imaging that the exact difference between the two resolutions depends a lot on the parameters as well as the timing of the onset, depending if it happens at a boundary of an averaging period. The results here are also contradicted a little by Fig C1, which shows clear differences between annual and decadal resolution data

**We agree that this was not made sufficiently clear. We take the mean over each decade of the annual data (e.g. years 1-10, 11-20, etc) and these form the data points of our decadal data. Having performed further tests, we have verified that the bias does not depend on whether the onset happens at the boundary or in the middle of an averaging period.**

Table 2 The bias of what? Specify in the table header.

**We apologise for not making this clear. This is the bias in the posterior mean onset time.**

Sect 3.3 Please specify the exact parameters used to generate the analogous transitions. You say that you cannot reliably estimate the transition length, but you do not show that you can actually estimated the other parameters without issues. I think this is minor but a few sentences in this regard would be good.

**The exact parameters of the analogous transitions will be given in a table in the appendices of our revised manuscript. Following the suggestion of referee #2, we also will include tests of the performance regarding the estimation of all other parameters.**

230ff It is unclear how the posterior mean and event-means are set up here. Are you generating 19 events or are you subsampling 19 events out of your 1000 realisations to calculate the event- averaged timing?

**We apologise that this is unclear. We have 19 abrupt warming events from the studied set of climate model simulations. We take the means over these 19 events (event-means) of the posterior means of each parameter. We will ensure that this is made clear in our revised manuscript.**

239ff Please elaborate on your the heteroscedacity issue: If the method is unreliable with the inclusion of different noise levels over the transition, than how is it reliable enough to estimate analogous parameters. It is also unclear if you generated the analogues with or without heteroscedacity which makes the last sentence of the paragraph quite confusing. Please rework. It also needs to be clearly shown and discussed whether the bias correction for the modelled precipitation is valid and if the model can be applied at all to the heteroscedastic data. The issue with these models is that their results are conditional on the model being a good description of the data, if that is not the case the results are to be taken only with a grain of salt and proper justification.

**For precipitation, we generate two sets of analogous transitions: one with heteroscedacity and one without. We will rewrite this paragraph to make this clear.**

**Although we agree that the way we have estimated the noise parameters on either side of the transition is imperfect, we feel it was the best approach to take in the circumstances.**

**We agree that the model is only valid if the model is a good representation of the data, and that this is not the case when dealing with heteroscedacity. In our view this only reinforces our point that leads and lags estimated using such a transition model are unreliable. After all, heteroscedacity is also present to some extent in most proxies across DO events, e.g. Capron et al. mention that the calcium record shows greater noise during stadials than interstadials, and so this issue would equally affect all the previous studies based on this transition model. We will make clear in our revised manuscript that heteroscedacity presents a particular problem for this transition model, but it is certainly not the sole reason for bias.**

Tab 3 : Please do not format the table in such a way that Ca seems to be associated with the NAO. This is potentially misleading especially as you shy away from discussing the proxy interpretation at all. Also, Even though temperature and precipitation over the North Atlantic that you use is probably not unrelated to the values at NGRIP, they are not the same. Please redo the analysis with the relevant variables or rework the manuscript in such a way that it is clear that you are not looking at model output for Greenland at all. This included the Table where these are stated next to each other as if they are the same.

**As above, we will remove any suggestion of connections between model variables and proxies.**

255ff Please discuss why the lags are so different between the different variables. Is that because of the shape of the transitions or the noise levels or both?

**The differences in the lags are due to both different noise characteristics and differently shaped transitions - it is difficult to separate the two.**

Sect. 3.5 Again there is no clear description on how the analogous transitions were generated. Without that information the results cannot be reproduced or verified by anyone.

**We apologise for this. We will provide the relevant parameters in an appendix and also make available the code that was used to generate the synthetic transitions.**

281ff I think the leading paragraph is a bit misleading without the actual information under which circumstances the bias arises.

**We politely disagree. The cause of the bias is unknown, and so it is extremely difficult to prove that any apparent time lags are not caused by bias. This certainly reduces the trustworthiness of any leads or lags found using this kind of method.**

285ff This paragraph probably should state somewhere that you look at this from a frequentist point of view. Erhardt et al. never made any claims about statistical significance, they provide credible intervals which are the parameter ranges that are consistent with the data under the assumptions of the model. This is not the same as the assumptions that go into the significance test, namely that under repeated observations and calculation of the significance interval of 95% the true value of the parameter will fall into the significance interval 95% of the time. These two views are fundamentally different and are not necessarily compatible with each other.

**We will make clear that we adopt a frequentist perspective, and that other approaches are available.**

318ff The paper mostly focuses on the adjusted method that you are proposing, and not the original method of E19. Please rephrase as all results with regard to the original method are only in the appendix of the paper.

**We will include in our revised manuscript additional results which clearly demonstrate that the original method of E19 is also biased (see Additional Figure 2 and Additional Table 1 on page 10 of this response)**

327f The conclusions of Capron et al. (2021) are based on fundamentally different reasons than the once here. Your's are purely related to the method and the interpretation of the results whereas those of Capron et al. (2021) are based on considerations with respect to climate variability and the tight coupling between different parts of the climate system. Please remove or rephrase.

**There is certainly a connection between the conclusions drawn by Capron and ours: Capron et al. state that either the elements of the climate system were too tightly coupled to detect significant leads and lags, or that individual DOs followed different mechanistic patterns. In the first case, one could still imagine that there existed a method that was able to detect phase relations of the transitions. So they implicitly say that the elements of the climate system are 'too tightly coupled to disentangle a characteristic sequence with the available methods.' We show that the Bayesian ramp-fit indeed cannot do this job, underpinning their assessment.**

**We will however add a comment on the possibility that different DO events followed different causal patterns.**

329ff The conclusion is inconsistent. The method is either good or not for the investigation of DO events.

**We apologise for being unclear. We mean to say that, even though we have demonstrated a flaw in this method, there are none of which we are aware that can do a better job. We will rewrite the conclusion to be clearer.**

Please decide and rewrite. Please also state why you think that the method would be suited if the lags are larger than 20yrs. This is not entirely clear given the discussion beforehand.

**We accept that the value of 20 years is not sufficiently supported, and we cannot give an exact value for the minimum time lag that we can be sure is not due to bias. We will rephrase this sentence as following to be more accurate:**

**"However, it appears impossible to reliably determine the temporal phasing when dealing with decadal-scale time lags such as those that have been suggested for DO warming events."**

Supplement Choice of prior distributions

Either Equation 10 or your code is not correct, please check.

**We have not been able to find the mistake that the referee has apparently identified. On lines 102-103, we return 0 likelihood unless $0 < \sigma \leq |dy|$. Otherwise, we do not explicitly define a prior to sigma, which we believe is equivalent to a uniform prior. If we are mistaken then we would be very grateful if the referee could explain more precisely what our error is and correct us.**

The prior for tau in your analysis puts a lot more emphasis on much longer and much shorter autocorrelation times than the one in the original method. This is a bit problematic as we do not expect to see decades long autocorrelation in climate data as far as I know. Please elaborate. Sampler setup Please clearly describe the setup of the ensemble sampler. I have noted that you chose to keep the setup of E19 despite increasing the number of parameters. This should also go hand in hand with a change in the sampling strategy i.e. burnin and thinning and maybe even the number of ensemble walkers. These need to be carefully chosen to not get nonsensical results in the end.

**Our choice of prior for the autocorrelation time was designed to be less prescriptive whilst retaining the same peak of the probability distribution as in E19. As discussed on pages 30-32 of this response, we do not feel that the long autocorrelation times for decadal data are necessarily indicative of an issue.**

**Regarding the sampling strategy, we found that the setup of E19 was still valid and so did not see a need to change it. We manually inspected a subset of our synthetic**

**transitions for convergence, and found only very rare occasions where the sampler failed to converge.**

Figures C1-C3 Judging from a closer comparison of the results from your method to the results of the original E19 method in Figures C1 and C2 it would seem that the addition of the slopes generally decreases the performance of the method. This seems to be the case for the sensitivity of the bias towards noise, the transition length and very, very worryingly towards the inclusion of an interstadial slope. Please elaborate and discuss this especially in light of the analysis that you base upon the result from the fit including the slopes. Also it would seem, that at zero slopes prior and after the transition the original method overall outperformed yours, which would mean that part of the bias is because of the inclusion of the slopes into the model not despite.

**We thank the referee for raising this important point. We agree that this needs to be addressed in our revised manuscript. We go into detail regarding this matter in response to Referee #2 on pages 9-11. Broadly speaking, we find that the addition of slopes results in better performance when slopes are in fact present, but worse performance when they are not - as might be expected. As shown in Additional Table 1 on page 10, for some variables / proxies the original method has lower bias, and for some the extended method has lower bias. Overall, the bias is present in both cases and so our conclusions stand.**

---

## Author Response (AR1)

**The Temporal Phasing of Rapid Dansgaard–Oeschger Warming Events Cannot Be Reliably Determined**

**Authors' changes in response to the referees' comments.**

**Referee #1**

**The authors would like to thank the referee for their insightful comments and suggestions. The referee's comments are normal text and our responses follow in bold text.**
* * *
**General comments:**

The approaches described here, i.e. studying the impact of the Bayesian ramp fitting on the uncertainty of the estimation of lags between climatic parameters simulated by GCM or reconstructed from ice core records, is usually a discussion topic for paleoclimate studies. I think it is a perfectly valid study, and do not question the relevance of the publication, but highlight this point because I would say that the results of your study correspond to what would be the discussion of a paleo reconstruction of a DO event lag study: indeed, classical paleo papers would estimate the lags and in the discussion attempt to provide uncertainties or biases of their estimates. Here, I found the structure of the result section a bit confusing and would suggest the authors to reorganise it a bit keeping in mind what the results are. Specifically, to me, the actual results come in section 3.3., while sections 3.1 and 3.2 detail sensitivity tests on the parameters that were used for the synthetic transitions and the resolution.

**We recognise that the structure of our study is somewhat unusual for the field. However, the main purpose of this work is to test and critique the method being employed. From this perspective, section 3.3 is simply a further example that demonstrates the importance of the bias we find. We considered employing a more "standard" paleo reconstruction structure in which we would present a results section focused on the ice core data and then demonstrate the bias in the discussion section. However, if our study were structured in this way then the key finding of bias would not be introduced until late on in the manuscript, at which point it would largely invalidate the "results". For these reasons, we have retained the original structure.**

Overall, there are implicit equivalences that are made between the study of the outputs of CCSM4 and the parameters reconstructed from the NGRIP ice core which might be not accurate.

In Section 2.2, a list of what climatic parameters are usually associated with the proxies retrieved from the ice core record is provided, but it is mentioned that the one to one equivalence is not totally correct, for instance, δ 18 O is influenced by both temperature and sea ice extent changes (Sime et al., 2019). Then, the way that Table 3 and Figures 4 and 5 are represented suggests that some parameters are equivalent with the way that they are represented either on the same line or with the same colour.

**As multiple referees have made similar comments, it is clear that the equivalences we suggest between ice core proxies and model output variables are a source of confusion and that this distracts from the key point of our study. We have therefore removed any such suggestions of equivalences from the revised manuscript. In particular, we have separated out Table 2 (Table 3 of the initial manuscript) in such a way that it is clear that we are dealing with two different sets of data between which we do not make a connection. Furthermore, we have ensured that Figures 4 and 5 use distinct colours to avoid any suggestion of equivalence.**

Finally, I'm a bit confused with the choice of the reference parameter for the lag estimate for CCSM4 and the NGRIP ice core. Indeed, I don't understand why in CCSM4 the lag is calculated against temperature, while NGRIP ice core, it is calculated against the calcium, which is referenced to be a proxy of the NAO. This might be something classically done here, but since amongst the four parameters used from NGRIP ice core, δ 18 O would be the one the most closely related to temperature, it appears that the lag as presented here would have different meanings for models and paleoclimate reconstructions.

**We agree that this is potentially confusing, and thank the referee for pointing this out. The choice of the reference for the lag estimates is largely arbitrary, in the sense that we do not consider any single proxy / variable to be more important than any others. As we have removed any equivalences between proxies and model variables (see comments above), we hope that this confusion has resolved itself. The analyses of the model simulations and the ice core proxies should be considered as separate demonstrations of the impact of the bias we find.**

**Specific comments:**

Introduction: the introduction overall feels too detailed, to the point that I struggled to see what was the topic of the manuscript when I first read it.

**We have reduced the length of the introduction and made sure that the topic is made more obvious. In particular, we have removed any reference to the multi-archive studies (Buizert et al., 2018; Adolphi et al., 2018; Corrick et al., 2020; Svensson et al., 2020) as these are only tangentially relevant to this work.**

Line 58 says "Subsequent to this work…", suggesting that the overall timeline of the study of lags of climatic parameters for DO events are detailed, while an introduction probably just needs the most up-to-date information that is needed to understand the interest of the manuscript.

**We agree that the phrasing here suggests a more detailed review of the literature than is necessary or is indeed the case. There are three key papers regarding this topic that have been published in the last five years (Erhardt et al., 2019; Capron et al., 2021; Riechers and Boers, 2021) and we think that it is important to discuss the key findings of each. However, we have rewritten this section to make it clear that these three papers come to somewhat different conclusions, and that there is not yet full consensus, rather than suggesting (as our initial manuscript perhaps does) that there is a timeline whereby later research has superseded earlier work.**

Line 44: "On the first line of evidence, Adolphi et al…" It is not clear what line of evidence this is. The similar formulation line 53 also puzzled me.

**In our original manuscript, by "the first line of evidence" we referred to multi-archive studies, and by "the second line of evidence" we referred to analysis of multi-tracer records from Greenland ice cores. We have removed any reference to multi-archive studies, which we hope solves the confusion here.**

Line 89: "which Vettoretti et al. (2022) have kindly provided." "Kindly" is not an appropriate word to use, especially considering that most journals now require open access to the data of a published paper.

**We have removed the word "kindly" from this line.**

Lines 124 – 125: "For in-depth discussion of the physical interpretation of these proxies, see Section 2 of Erhardt et al. (2019) and references therein." This could be personal style, but I believe that a manuscript should be stand-alone. The detailed discussion of the physical interpretation of the proxies is beyond the scope of this manuscript, but overall idea should be provided in the introduction or the methods.

**We would reiterate that this study is primarily concerned with testing the Bayesian ramp-fitting method, and so we feel that the current level of discussion of the proxy interpretations is sufficient. As discussed above, we have also removed any suggestion of equivalence between proxies and model variables, which we feel further reduces the need for detailed discussion of proxy interpretations in this work.**

Lines 128 – 129: "The aerosol concentrations were measured using Continuous Flow Analysis (CFA) at temporal resolution ranging from 2 years for the most recent period to 3 years for the oldest period" Was anything measured for this manuscript? Is the fact that the data are measured via CFA relevant here? This sentence was very confusing, making me think that you measured aerosol concentrations within this manuscript. Overall, only the effective resolution of the data is important here. Considering that all this information is already provided in the table, I would remove this paragraph.

**We agree with this suggestion and have removed this paragraph.**

Line 138: "The addition of an AR(1) noise process makes the model probabilistic" Where is the AR(1) noise added? Why is it not explicitly given in an equation which shows exactly what is added where?

**Thank you for raising this. We have explicitly written out the AR(1) noise in equation (1), as well as more clearly setting out the difference between the deterministic and noise components of the transition model.**

Line 143: "This leads to improved agreement of the transition model with the analysed data (Figure 1)." This is not clear. Is the improved agreement shown in the histogram below each figure? What do they mean? they have no label, no caption, and no explanation.

**Thank you for this remark. Indeed, the histograms are not conveniently described in the figure caption, nor are the y-axes of the histograms labelled. We apologise for the oversight and have corrected this issue in our revised manuscript. The blue and orange histograms below each panel respectively show the posterior distributions for the onset and end times of the transition. The y-axes of the histograms are now labelled as Probability Density / Years$^{-1}$, as is the case in Figure 2.**

**We have not quantified the overall improvement of the fits. However, we believe that the improvement is clear from a visual comparison of the top row with the bottom row, particularly panels c and f. We have added the sentence: 'The additions of slopes before and after the transition yields a clear improvement of the fit of the deterministic model core to the data.' at the end**

of the model caption. A more quantitative assessment of the two different implementations is also now included in Appendix A.

Lines 143 – 144: "Additionally, our extension of the method reduces the sensitivity of the transition timing to the search window, which is otherwise one of the drawbacks of this method (Capron et al., 2021)." Also unclear what is meant here.

**We agree that this sentence was not clear for people that are not already familiar with this method. We therefore have added the following explanation in our revised manuscript: 'In order to apply the ramp-fit to individual transitions, one has to select a data window around that transition. There are no objective criteria for the starting point and the endpoint of these windows, other than that no other transition should be included. However, changing the boundaries of the data window influences the results of the estimation. This effect is reduced if slopes before and after the transition are allowed.' Also, as requested by another referee, we have added here a quantification of this reduction in sensitivity to the search window for a particular case.**

Line 157: The detailed description of each hypothesis is one line of text each. They should be included here, and not in the appendix.

**Thank you for this remark. We have moved the formulation of the hypothesis to this point:**

**"– H0: The population mean time lag of this variable relative to temperature is equal to zero.**

**–H1: The population mean time lag of this variable relative to temperature is not equal to zero."**

Lines 160 – 161: "We conduct our analysis of the NGRIP ice core in the same manner. In this instance, we calculate time lags for the other three proxies relative to Ca"

Why calcium which you say represent the NAO (line 120), and not d18O which is closer to temperature? This would be more coherent compared to what is done with models?

**As discussed above, we have removed any equivalence between proxies and model variables. The choice of reference from which to calculate leads and lags is then arbitrary.**

Lines 227 – 229: "However, we have established that there is no unbiased means by which to estimate these transition durations, and so we could not guarantee that this would be any more accurate."

Where was this established? And shouldn't be this line in discussion rather than in Results?

**Indeed, this statement was a bit misleading. We were trying to say the following: For constructing the analogous synthetic transitions, we could in principle estimate typical transition durations from the data using the ramp-fit. However, we have established that the transition duration is itself prone to bias, as bias in the onset time directly propagates to bias in the transition duration. This was mentioned in lines 202-205 of the initial manuscript.**

**We cannot definitively say that there does not exist any method to accurately estimate the transition duration, as this would require us to test every conceivable approach. What we were instead intending to communicate is that we have not been able to identify any such unbiased method for estimating the duration. Because of this, we simply use a plausible fixed value to estimate the bias in the different model variables and ice core proxies.**

**Whilst we had previously indicated that we would move this line to the discussion section, upon further reflection we feel that it is important to explain more carefully here. Our revised manuscript therefore reads:**

**"We make an exception for transition duration, choosing instead to use a fixed duration of 50 years for all of our "analogous" transitions. This is because we suspect that much of the apparent difference in duration between different variables is in fact due to the bias we have identified. As we have been unable to identify any unbiased method by which to estimate the true transition durations, we simply fix them at a plausible value based on the durations of the transitions in two ice-cores as estimated in a previous study (Capron et al., 2021)."**

Figure 5: The matching colours between Figures 4 and 5 suggest equivalency between precipitation and Sodium, sea ice and d18O, and AMOC and thickness. The colours should be different in both figures if they are not supposed to represent equivalent parameters.

**We thank the referee for noting this. We have changed the colours to avoid any suggestion of equivalence.**

**Referee #2**

**The authors would like to thank Referee #2 for their extremely thorough and useful comments. The referee's comments are normal text, and the authors' responses are in bold.**
* * *
The manuscript presents an analysis of some limitations of ramp fitting as a tool to determine the characteristics of abrupt events seen in palaeoclimate data and model runs. The study is relevant, generally well presented although (some more details could be wished for in places), and suitable for publication in Climate of the Past. First, the ramp fitting method is extended to take pre- and post-transition trends in the data into account. This is a good move when the method is applied to model data which exhibit more pronounced trends in especially the interstadials, but see further comments below on the need for modelling a trend in the stadials. The main finding is that the ramp fitting method detects ramp start points that are generally too early when the method is tested on synthetic data, especially with high noise levels. This effect is denoted "bias", and the bias is then quantified for a range of scenarios.

The effect can be anything from one year upwards, depending mainly on the noise-to-signal levels. I have some methodological concerns and some regarding the presentation of the results, that I think should be addressed before publication:

1. The manuscript does not really discuss the magnitude of the suggested bias compared to the credible intervals produced by the ramp fitting method as published in earlier studies. If, for example, the bias is a few years in cases where the credible intervals span decades, the bias will unlikely lead to wrong conclusions. In this case, the existence of a systematic bias would still be a valid result (and to people interested in the methodology also an important one), but the analysis should consistently address whether the bias is sufficiently contained within the existing uncertainty estimates.

**This is an important point, and we thank the referee for drawing attention to it. As can be seen from the blue histograms in Figure 1, showing the posterior distribution for the transition onset time, the width of the credible interval is generally on the order of a few decades. It is therefore absolutely correct to state that, for individual events, the bias is contained within the existing uncertainty estimates. So, if this method is used only to compare individual events, as for example is done by Capron et al. (2021), then the bias we have uncovered is relatively unimportant.**

**However, this changes if one attempts to calculate characteristic leads and lags from a set of DO events. Although employing different methods, both Erhardt et al. (2019) and Riechers & Boers (2021) essentially combine many**

events in order to reduce the uncertainty on the leads / lags between proxies. Because of this, the range of the credible intervals shrinks to around a decade - for example, Erhardt et al. quote a lag of 7 ± 6 years between calcium and sodium in the NGRIP core. The bias we find is therefore no longer contained within the existing uncertainty estimates, and so when trying to assess leads or lags this bias could easily lead to wrong conclusions.

This is the point we are trying to make with our hypothesis test in Section 3.4: Without any consideration of bias, we have an apparently statistically significant lag between precipitation and temperature in the CCSM4 simulations. However, if we correct for a plausible estimate of the bias, this is no longer the case. This is a clear example of a situation where the bias leads one to the wrong conclusion.

We agree that our original manuscript does not make this distinction explicitly enough. We have made clear in the discussion and conclusion of our revised manuscript that the bias only becomes particularly important when considering leads and lags that have been calculated by combining many events, reducing the uncertainty in the process.

2. There is often a downward slope during interstadials, especially in the model time series. However, it is not obvious whether the model should be changed to allow/detect a slope during the stadial pre-ramp period. Do the authors find significant (negative or positive) slopes of pre- ramp levels in real data (both proxy and model data)? It is simply not clear in the manuscript whether the addition of a slope in the stadial fixes a real problem.

Perhaps the simplest way to persuade the reviewer that there are meaningful slopes during the pre-ramp stadial is to show some examples where this is obviously the case - for example AMOC in the CCSM4 simulation with 225 ppm of $CO_2$.

[Figure]

*Additional Figure 1: The extended ramp fitting method as applied to AMOC for three abrupt warming events in the CCSM4 simulation 225 ppm of $CO_2$. There is an obvious positive pre-ramp (stadial) slope in all three cases. This is included in our revised manuscript, but it instead simply included here to address the referee's comment.*

**Furthermore, for the NGRIP ice core proxies, our extended method finds slopes of comparable magnitude in the pre-ramp stadial and post-ramp interstadial. This is made clear in our revised manuscript as all of the relevant parameters used to create the "analogous" synthetic parameters for the purpose of bias estimation are listed in Tables D1 and D2 in the appendix. We most often find that the slope during the pre-ramp stadial is positive - that is to say in the same direction as the ramp itself. See also the mean parameters for each proxy (including slopes) shown on Additional Figure 2 (page 10). The proxies all show significant slopes, in the sense that these slopes lead to a significant bias, however we have not tested the statistical significance of these slopes in isolation.**

It is not surprising that allowing a negative slope in the stadial before a steep positive ramp will tend to lead to earlier ramp onsets, especially with noisy data. As I understand it, this would likely lead the authors to conclude that the model without stadial slopes has a bias. But if there is no significant slope in the stadial (and preferably a physical reason to why such a slope would be expected), it is not clear that allowing the model to have a sloping stadial is an improvement in terms of representing the underlying physics. If there is no significant stadial slope, an observed difference in the timing of the ramp onset between the two models (with/without stadial slope) is simply a result of different assumptions about the signal structure and not necessarily a bias as such. The authors should analyze this question carefully to justify whether/when/why addition of a pre-ramp slope is an improvement. And they should test (for synthetic data created without any pre-ramp slope in the noise-free data) how adding this extra degree of freedom changes the detected ramp characteristics and report the results explicitly in the manuscript.

**We would like to make it very clear that we always calculate the bias against the true, known value of the onset time for synthetic transitions. We do not ever conclude that bias is present based on the difference between the original and extended implementations of the ramp-fitting method. Furthermore, panel f in Figure 3 shows that we actually see later ramp onsets (compared to the case of zero slope) when we have a negative slope in the stadial before a steep positive ramp - contrary to what the referee suggests they would expect.**

**However, we agree that we have not sufficiently compared the efficacy of the original (without slopes) and extended (with slopes) implementations. We have therefore expanded Figure 3 to show the bias of the original Erhardt et al. implementation (Additional Figure 2, page 10). From visual comparison, we**

can see that the original implementation (with no slopes) shows lower bias for the case of no or negligible slopes in either stadial or interstadial (the middle band of panels b and f, or the bottom region of panels c and g). However, this original implementation suffers strongly from bias when there is even a slight slope in the pre-ramp stadial.

[Figure]

*Additional Figure 2: The bias in the transition onset time, based on sensitivity testing of the original Erhardt et al. implementation of the ramp fitting method, for annual (a-d) and decadal (e-h) resolution. Overlaid are points showing the mean transition parameters for the ice-core proxies and model output variables that are considered in this study. When not explicitly varied, parameters are fixed at standard values. These panels are incorporated into Figure 3 in our revised manuscript.*

When faced with transitions that include significant pre- and post-ramp slopes, we see better results (less bias) from representing these slopes in the ramp-fitting method. However, when dealing with transitions that contain no such slopes, we see worse results (more bias) as we have introduced two unnecessary new degrees of freedom. For more realistic intermediate scenarios it is less immediately obvious which of the two implementations would lead to more bias. To address this point, we have added an extra column to Table 2 (Table 3 of the initial manuscript) that includes the bias based on the original Erhardt implementation that does not include slopes. We include these results here to demonstrate that neither of the two implementations is obviously superior overall.

| Variable / Proxy | Onset Time Bias Under Original Implementation / Years | Onset Time Bias Under Extended Implementation / Years |
|---|---|---|
| Temperature | -0.7 ± 0.2 | -1.0 ± 0.2 |

| Precipitation | -8.8 ± 0.7 (-7.1 ± 0.6) | -8.9 ± 0.6 (-6.0 ± 0.5) |
|---|---|---|
| Sea Ice | -0.9 ± 0.2 | -0.8 ± 0.2 |
| NAO | -1.5 ± 0.2 | -1.2 ± 0.2 |
| AMOC | -1.8 ± 0.4 | -2.1 ± 0.3 |
| $\delta^{18}O$ | -7.6 ± 0.8 | -10.2 ± 0.9 |
| Annual Layer Thickness | -9.2 ± 0.5 | -6.0 ± 0.5 |
| Na | -6.8 ± 0.6 | -9.4 ± 0.6 |
| Ca | -9.7 ± 0.5 | -4.8 ± 0.4 |

*Additional Table 1: Onset Time Bias for the different model variables and ice-core proxies in both the original and extended implementations of the ramp-fitting method. The uncertainties here are the standard errors on the mean across 1000 synthetic transitions. Unlike in Table 3 of our initial manuscript, the uncertainty in each individual event is now fully propagated, resulting in slightly larger uncertainties. These uncertainties reflect only the uncertainty in the bias given a particular set of transition parameters. They do not account for uncertainty in our estimates of those transition parameters. This forms Table 2 in our revised manuscript. Note that we no longer equate certain variables and proxies.*

3. In Capron, the results of a single test of possible bias are given in the Methods section (and Fig. S2). Here, no significant bias was detected (the 50-year duration of the synthetic data was reconstructed as approximately 51 years by the method (and as an add-on to the previous point: the results in the S2 figure illustrate that the uncertainties are considerable and easily would include a bias of a few years).

Because the results differ, it would be interesting if the authors could discuss their analysis in the context of the published test. Maybe the way the test data were created could make a difference (also see next point)?

**This is an important point and we thank the referee for raising it. We have made sure to address this in the discussion section of our revised manuscript:**

**"Capron et al. (2021) also test for possible bias using synthetic transitions with autocorrelated noise and find no significant bias in either the transition midpoint or duration, implying that the transition onset time must also be unbiased. This stands in contrast to our findings in this study, and so merits further consideration. One possible reason for the discrepancy is that Capron et al. (2021) only use 20 synthetic transitions. This may not be sufficient for the bias to become apparent due to the large variability between individual events.**

A further possible reason could be that Capron et al. (2021) construct their synthetic transitions with no slopes before or after the abrupt ramp. We observe in Figure 3 that, when using the original Erhardt et al. (2019) implementation, synthetic transitions without any slopes before or after the ramp show very little bias, but also that the bias grows rapidly when even slight slopes are included. We suggest that the root of the discrepancy is therefore the assumption made by Capron et al. (2021) that there are no slopes in either the stadial that precedes the transition or the interstadial that follows."

4. The authors do not describe in detail how they generate their synthetic test data, and when they estimate the bias, how many independent realizations are run (my experience is that especially for high noise levels, averaging over at least several tens of realizations is necessary). This information should be provided somewhere.

Also, and especially because the conclusions differ from those of earlier tests (e.g. those mentioned in Capron), the synthetic data series used here should be made available to allow others to make future tests against the exact same test data, thus reducing the risk that differences could be caused by implementation issues.

This study makes use of around 200,000 synthetic transitions, and so we do not feel that it is practical to provide these. We do however provide the code which was used to create these synthetic transitions so that interested readers can gain a full understanding of how this was done. We have also added the following in Section 3.1:

"We would ideally like to vary all five transition parameters simultaneously in order to assess possible inter-dependencies of the bias on different transition parameters. However, it would be challenging to visualise and interpret the resulting five-dimensional parameter-space. We therefore vary only two transition parameters at any one time - Noise / Signal Ratio and one other - whilst keeping the rest fixed at standard values as follows:

- **Autocorrelation Time = 10 years for the case of decadal resolution and 1 year for the case of annual resolution.**
- **Greenland Stadial Slope = 0 Kiloyears$^{-1}$**
- **Greenland Interstadial Slope = -1 Kiloyears$^{-1}$. Note that the Greenland Interstadial Slope is always <= 0 in our sensitivity tests due to the sawtooth shape of DO events. In Figure 3 we plot the absolute value or magnitude of this slope, which is positive. Note that the slopes are normalised relative to the amplitude of the transition. This standard value for the Greenland Interstadial Slope therefore means that the time**

**series will return to its pre-transition level 1000 years after the transition ends.**
- **Transition Duration = 50 years.”**

**We have also included the following in the caption of Figure 3:**

**“Each panel contains 100 unique sets of parameter values, and for each set of parameters we take an event-average over 100 synthetic transitions. When not explicitly varied, parameters are fixed at standard values as given in Section 3.1.”**

**To more clearly explain the bias corrections in Section 3.3, we have included in Tables D1 and D2 the parameter values used for the “analogous” synthetic transitions to each model variable and ice core proxy.**

5. Related to the former point, one could suspect that especially when adding relatively high levels of autocorrelated noise, the transitions (i.e., ramps) of the synthetic data actually become more gradual (with longer durations and/or earlier onsets as a consequence). This is what was tested in Capron Fig. S2 for one scenario, but the results could be different for other parameters and other noise-to-signal levels (see PS below). If this indeed is the case, I would say that it changes the way we should look at this: it's not “just” the ramp fitting procedure that has an issue, but an inherent limitation related to the nature of climate data with autocorrelated noise (or data affected by autocorrelated short-term variability, if you like). I think this issue is related to what the authors are hinting at around line 226-228, but it could be expanded and made clearer.

**This is an interesting interpretation of our findings, but we do not fully agree. The synthetic transitions that we use for testing have a true duration that is known and unrelated to the degree of noise. Clearly, higher noise-to-signal ratios will lead to a higher degree of uncertainty in both the onset time and the duration. However, it is generally possible in statistics to have an estimator that remains unbiased even as the level of uncertainty increases. We therefore feel that one cannot conclude from our work that bias of this sort is an inherent limitation of climate data with autocorrelated noise - it remains possible that there exists an unbiased method to address this problem, and that we simply haven't been able to identify it.**

**In light of this, we have rephrased lines 226-228 of our initial manuscript to state that “we have been unable to identify any unbiased method”, instead of the current statement that “there is no unbiased [method]”.**

**It also might actually be that there are no conflicting views here, and that the referee simply chose different words than we would choose to express very**

**similar thoughts. We agree that given the current methods, the autocorrelation of the noise in climate data ultimately limits this sort of analysis. However, we would still ascribe the bias to the method and not to the data.**

**We have added the following lines to Section 3.1 of our revised manuscript to reflect the possibility that the referee highlights:**

**"This might suggest that such bias is a fundamental limitation of any attempt to identify the timing of abrupt transitions in noisy climate data. However, we equally cannot prove that there is no possible unbiased method - certainly in many other statistical settings there are estimators that remain unbiased even as their uncertainty increases."**

6. Somewhat along the same lines: It is not surprising in itself that high noise levels lead to longer transitions: As the noise level increases, (probably all) ramp detection methods will have a harder time reconstructing the (no-noise) ramp duration, and there will therefore be a larger spread in the detected ramp durations. If the noise is large enough, the range of ramp durations becomes asymmetric because the ramp durations by design are positive. An example with a synthetic ramp duration of 50 years: For high noise levels, the distribution of accepted solutions produced by the method will include values that are larger than 100 years, but by design none which are smaller than zero. The mean of the distribution will be biased towards larger durations and lead to earlier onset points (and/or later ramp end points).

In the test described in Capron, this effect of asymmetric distributions is observed although it does not lead to a bias in transition duration similar to the one reported in the current manuscript (see PS below). However, by using the median rather than the mean, the effect on the result can be reduced. This highlights two points. Firstly, using the median may be more appropriate than the mean. This should be discussed and tested explicitly. Secondly, this is not really an issue with the ramp fitting method, but a consequence of noisy ramped data. The observation is consistent with Fig. 3h which shows that there is largest bias towards early onsets for the most noisy and fastest transitions. I think it would strengthen the manuscript a lot if the authors could go a bit deeper into discussing the issues raised in point 5 and 6.

If they agree with these points, I encourage them to think about how to phrase their results in a way that better acknowledges that it's not "just" a methodological problem with ramp fitting, but an inherent limitation that both is related to the method and to the nature of autocorrelated ramped data and/or that (some of) their bias comes from the way the synthetic data are created.

The conclusion of Capron already discusses some of these issues (and tries to separate those that originate from the complexities of the climate system and those that are more related to the noise and method). It would be good if the authors spend

a bit more space on discussing how their results confirm, refine (or even contradict) the ideas presented there, and separate between the different types of difficulties.

**The referee is correct to state that for high enough levels of noise, the distributions will become asymmetric because the duration must be greater than 0. This may partially explain the bias towards early transition onsets and longer durations. However, this does not explain why the other transition parameters beside the noise / signal ratio also impact the bias, or why we sometimes see a bias towards later transitions (and so shorter durations), e.g. Figure 3h for true transition duration > 80 years. We therefore feel that this is certainly not the whole picture.**

**Our key aim is to assess the bias affecting calculations of leads and lags. It is not obvious to us whether the posterior mean or posterior median is more appropriate for this purpose, but we agree that using the median may reduce the apparent bias. We have tested this, and we show below (Additional Figure 3) a figure that is equivalent to Figure 3 in our original manuscript but using the median as suggested. We do, as the referee suggests, observe a slight reduction in the magnitude of the bias, but this is a relatively small change that we feel does not impact any of our conclusions. Nonetheless, we have included Figure C3 showing this in our revised and we also discuss this point in Section 3.1 as follows:**

**"Defining the bias as the expectation of the error in the posterior-median onset time, rather than the posterior-mean, results in slightly more positive values for the bias (see Figure C3), meaning that the generally negative biases we observe are somewhat decreased in magnitude. However, the differences are small and the overall pattern unchanged. This demonstrates that the choice between posterior-mean and posterior-median when estimating bias is unimportant, and so we only consider the posterior-mean going forwards."**

[Figure]

**Additional Figure 3: The bias in the median onset time, using the extended ramp fitting method. We see very similar results to the bias in the mean onset time (Figure 3 of original manuscript), with only a slight reduction in the size of the bias. This is included as Figure C3 in the appendices of our revised manuscript.**

Finally, we feel that the distinction between an issue with the ramp fitting method and an issue with noisy data is somewhat unclear, being that the ramp fitting method is inherently designed to work with noisy data. Furthermore, the synthetic data that we create assumes the same underlying transition model as the ramp fitting method itself, and so there shouldn't be any issues arising because of the synthetic data generation. In our view, it remains unclear whether this is purely a methodological problem or in fact an inherent limitation of dealing with noisy climate data, as the referee suggests. This is ultimately a question of whether or not there exists a method that can produce an unbiased estimate of the timing and duration of a transition in noisy, autocorrelated data. As stated previously, we do not know of any such method, but equally we do not feel confident in stating that such a method does not exist.

These suggestions and interpretations put forward by the referee are intriguing, and we hope that we have sufficiently addressed them in our revised manuscript.

[PS on the test from Capron Fig. S2: While writing the review, and with inspiration from the manuscript and some tests performed by a student in Copenhagen last year (not published), I first ran some more realizations of the same test as in Capron: The mean reconstructed ramp duration was 51.9 years, and the median 51.5 years.This would probably correspond to an early ramp onset bias of ~1 year, considerably less

than reported in the manuscript at hand, and well within any reasonable uncertainty estimate based on the width of the distributions. Reducing the noise level to half led to no significant bias (median 49.6 years, mean 50.2 years). Then I tried with double noise level (stadial noise level is the default in this test, which is already a relatively high noise level) and observed a more pronounced bias in transition length, which was also larger when using the mean rather than median. I did not check this through well enough to share the results, so I provide only this indication. I think this supports the idea mentioned in point 5-6. It also indicates that in Capron (at least for Calcium and in the absence of significant slopes of the pre- and post-transition values), bias does not play any important role for the results.]

**We thank the referee for making the effort to perform tests of their own in order to help refine our study. As mentioned in our response to point 3 above, when using both synthetic transitions and a ramp fitting method that do not include slopes, we find a weak bias, even at relatively high levels of noise. However, when we account for such slopes, we find that both implementations (with and without slopes) of the ramp fitting method that we considered do in fact have meaningful bias for all of the ice core proxies considered.**

On a more technical note, I assume that the authors have tested their implementation (including generation of synthetic data). It would be nice with a summary of the tests: Does the method faithfully identify the properties of the synthetic data (including the autocorrelation time) for low noise levels? Figure 3 goes some way in demonstrating this, but I think a bit more details and a dedicated section would be appropriate (possibly in a supplement).

**We present here the results of some low noise tests. The noise / signal ratio is fixed at 0.01, and unless explicitly varied the other parameters are fixed at their standard values as given on page 12 of this response. We see that autocorrelation time is slightly overestimated, and that large negative slopes in the pre-ramp stadial are underestimated, although we rarely see such large negative pre-ramp slopes in either the model or ice-core data. These are limitations that need to be discussed, and we thank the referee for raising this issue. We have included Additional Figure 4 (below) as Figure A3 in our revised manuscript.**

[Figure]

*Additional Figure 4: Tests of the extended ramp fitting method at low noise / signal. This is included in the appendices of our revised manuscript.*

Finally, I think the title is not fully aligned with the content and conclusions. The following line from the conclusion "Despite the bias that we have uncovered here, we nevertheless suggest that this Bayesian ramp fitting method remains the best approach to understanding the temporal phasing of DO events" does not really work well together with the title "The Temporal Phasing of Rapid Dansgaard–Oeschger Warming Events Cannot Be Reliably Determined".

**We agree that the title and conclusions do not fully align as currently stated. In line with our response to the referee's point 1, we have changed the highlighted line of the conclusion to the following:**

**"The Bayesian ramp fitting method considered in this study remains a powerful tool for investigating individual abrupt transitions, as the bias that we find is small relative to the uncertainty of individual events. When calculating leads and lags from a sample of events, however, the bias becomes large relative to the uncertainty and so is likely to lead to false conclusions."**

In summary, I think the manuscript is appropriate for publication in Climate of the Past, but that additions and revisions are needed to fully document the analysis and make it clearer to which degree and under which circumstances the extended model with stadial slopes represents an improvement in the context of the data at hand, discuss in more depth whether the bias is a methodological problem or arise from the inherent difficulties of identifying ramps in noisy data, and assess whether the proposed bias will influence earlier conclusions based on the ramp fitting methodology.

Review by Sune O. Rasmussen, Jan 10 th , 2024. Line-by-line comments

29: "However, as of yet it has not been possible to conclusively identify a cause for these rapid Dansgaard–Oeschger warming events" could be rephrased. The exact mechanisms leading to the described changes are not understood but the AMOC state change is generally accepted as the main cause of most of the observed changes in the climate system.

**We have changed the word "cause" to "trigger" and cite Capron et al., as they state in their opening paragraph that "no consensus exists yet to explain what triggers the abrupt warmings".**

34-39: The papers of Adolphi and Corrick align records from ice cores and speleothems and have to deal with non-climatic dating offsets between different dating approaches. The WAIS and Svensson papers use synchronized ice-core

records. Because the uncertainties (in terms of both magnitude and nature) are quite different, it would be better to describe them independently. Bias in event detection of ~20 years are only marginally relevant in this context, which is not the case for the methods described in line 39-41.

**We agree with the referee that our work is largely irrelevant to the papers cited in these lines, and vice versa. We have removed all discussion of inter-archive studies and focus solely on the studies that compare different proxies within ice cores.**

39: (Buizert et al., a) should be cited as "WAIS Divide Project Members".

**We no longer cite this study.**

47-48 and 51-52: Elaborate or remove. The current sentence is vague. How do they contradict each other (given the error bars)? Consider if it would make the discussion clearer to include a discussion of the difference of absolute vs. relative dating errors. An ice-core timescale can build up significant bias over long stretches, and the published maximum error of GICC05 is therefore large in MIS 3.

**As stated above, we have removed all discussion of these inter-archive studies.**

58-63: I believe the papers came out in the reverse order: Capron et al. was accepted while the manuscript of Riechers and Boers was in the discussion phase. Please adjust wording accordingly, and in particular, be careful to cite the conclusions of Capron et al. correctly. I do not find a statement in Capron et al. to support the claim "they also concluded that applying a method of the type developed by Erhardt et al. (2019) to ice core data may not allow the identification of a unique order". In contrast, Capron et al. acknowledges that the Erhardt et al. approach of stacking may be appropriate under certain assumptions. The lack of a well-established order of events which is similar between events could both be due to noise / internal variability making detection difficult, but also that such a "standard sequence of events" may not exist.

**We thank the referee for pointing out that we had the papers in the wrong order. We also agree that our original formulation in this section was imprecise, and we have changed it in the revised manuscript. Our statement unintentionally suggests that it is a failure of the method that leads to no systematic pattern of leads and lags being found by Capron et al. However, as the referee helpfully notes, Capron et al. actually attribute this finding to either i) a large degree of climate variability obscuring a common mechanism or ii) different mechanisms for different DO events. We have revised this paragraph so that the order of the papers is correct and so that the conclusions of**

**Capron et al. are more accurately summarised. This section of the introduction reads as follows in our revised manuscript:**

**"A Bayesian ramp-fitting method, designed to address this challenge, has been developed and presented by Erhardt et al. (2019). Stacking multiple DO events in a multi-proxy analysis of Greenland ice-core data, Erhardt et al. (2019) identified time 45 lags between the different proxies. Considering DO events back to 60,000 years before present, the authors concluded that atmospheric changes preceded the reduction in sea-ice extent by around a decade. Using the same method, Capron et al. (2021) followed up on this research by extending a similar multi-proxy analysis of Greenland ice-core data back to 120,000 years ago. Unlike the previous study, Capron et al. (2021) suggested that the presence or absence of time lags might not be possible to determine due to both the tight coupling of different climate components and the substantial variability between different DO 50 events.**

**Subsequently, Riechers and Boers (2021) re-examined the same ice-core data considered by Erhardt et al. (2019), using the same ramp-fitting method but adopting a different statistical framework that more rigorously propagates the uncertainties in the onset time of each DO event. Under this new framework, the authors found that the time lags between proxies are not statistically significant, in disagreement with Erhardt et al. (2019). Therefore, although the method developed by Erhardt et al. (2019) 55 holds much promise, it has not yet led to a conclusive understanding of the temporal phasing of DO events. However, this may perhaps be possible in the future through application to either improved ice-core records or data from model simulations."**

72: Maybe reconsider use of "cause" (as above).

**We have changed "cause" to "trigger" as above.**

90: Proofreading needed.

**We thank the referee for bringing to our attention the poor grammar in this line, and we have corrected it. The highlighted sentence now reads:**

**"Each of these six simulations is 8000 years long, and exhibits millennial-scale variability which strongly resembles DO events as observed in paleoclimate archives"**

93-94: Vague. Refer to Vettoretti et al. 2022 Fig. 2a or describe how your analysis is different.

**We have removed these lines as they are not necessary.**

110: It is not clear what "parameter ranges" refer to.

**Thank you for raising this lack of clarity. We meant the ranges of the ramp parameters that one obtains when applying the Bayesian ramp-fit to the annual simulation data. We have clarified this statement by writing**

> **"Alongside the full data-set that is available at 10-year resolution, there are a small number of annually-resolved simulations. Although we do not have a sufficient number of annually-resolved abrupt warming events to assess potential systematic time lags, an application of the Bayesian ramp-fit allows us to gain a sense of the ranges of the ramp and noise parameters. This in turn allows us to gain insight into how increased resolution impacts the bias in the ramp fitting method. "**

**instead of**

> **"Although not sufficient to assess potential systematic time lags, these roughly indicate the possible parameter ranges at this higher resolution, and so allow us to gain an insight into how increased resolution impacts the bias in the ramp fitting method."**

117: Describe somewhere (here, in Methods, or supplementary material) what "for which the method was successful" means.

**We meant by this that Erhardt et al. were successful in applying their method. We have changed the sentence that started on line 116 of the initial manuscript to the following:**

**"Whilst Erhardt et al. considered a larger set of transitions, these 16 are the only ones for which they were successfully able to apply their method to all four proxies."**

**As Erhardt et al. do not state the criteria they used to judge the success or otherwise of their method in each case, we refrain from commenting on this.**

119-120: Add charge to ion or spell out full names.

**We have added charges to the ions.**

120: Note that Capron et al. did not make a one-to-one "identification" of proxies and model data extracts: "Over each D-O-like transition, we extract the time series of four

climatic measures from the model on the assumption that they reflect some of the same elements of the climate system as our ice-core proxy data". Revise accordingly.

**Upon reflection, we feel that the identification of proxies with model variables is unimportant to the results of our study, and so for the sake of simplicity we have removed any mention of such a connection, including by reformatting Table 2 (Table 3 of the initial manuscript).**

122: Revise syntax.

**As we no longer equate certain proxies and model variables, we have removed this discussion.**

137: It would be prettier if x(t) was defined for $t\_0$ and $t\_1$ as well (as in appendix).

**Agreed. We have made Eq.(1) and Eq.(A1) consistent in the revised manuscript.**

154: This is a critical assumption which is also employed by Erhardt and discussed in Capron. I encourage the authors to consider and discuss the implications of the assumption (e.g. here or in section 2.5).

**Indeed, this assumption has far-reaching consequences for the physical interpretation of DO events. We agree with the referee that these should be made explicit. We have therefore added the following sentences:**

> **"This view is only meaningful if one accepts the 'one-mechanism hypothesis', which states that all DO events follow the same mechanism and that differences in their expression are exclusively due to internal climate variability. While this hypothesis may not be generally accepted for the real-world's DO events, we believe that the regularity of stadial-interstadial cycle seen in the DO simulations provides a strong argument that at least the simulated DO events were all governed by the same physics. We further emphasise that a statistical assessment in terms of hypothesis tests relies on the 'one-mechanism' hypothesis and would be meaningless if individual DO events were caused by different physical drivers."**

176: It is more common to use the SNR and not the "Noise / Signal ratio". I recommend using the more standard SNR unless there are good reasons for why not to do this.

**We recognise it is more standard to use SNR. In our case, the signal is fixed whilst we are varying the level of noise linearly. If we were to plot the bias in terms of SNR, we would therefore have non-equidistant data points. Overall, we feel that the Noise / Signal ratio both more directly reflects our approach (with signal fixed and noise varied) and provides greater ease of understanding plots such as Figure 3. On that basis we have stuck with our initial approach.**

178: Give more details on how the synthetic data were generated in order to realistically mimic true data.

**We have altered line 178 to the following:**

**"To quantify the strength of the bias, we construct synthetic transitions by the addition of randomly generated AR(1) noise to a piecewise linear ramp, exactly as in the extended transition model (Appendix A). Our intention is not to mimic true data as realistically as possible but instead to create data to which the ramp fitting method should be perfectly suited."**

192: (also see main point raised above). It is hardly surprising that adding a lot of noise to a steep transition tends to smoothen out the transition found by the ramp fitting method. In order to judge how serious this problem is, I suggest adding the following pieces of information: - How does the calculated UMOTE compare to the distributions of onset times? I.e., are the confidence intervals wide enough to include the bias?

**As in our response to the main point, we feel that the important comparison is not to the credible intervals of individual transition onsets, but instead to the uncertainty in calculated leads and lags derived from stacking many events.**

It would be informative to add some estimates of noise/signal for the actual data used by Erhardt at the start of this section, so the reader gets a feeling for how the observed biases compare to what would be expected in real data.

Same for the stadial and interstadial slopes: What are the realistic values for the Erhardt data sets?

**We have added shapes representing the mean values of the parameters in the Erhardt et al. NGRIP data to Figure 3, as were already present in our initial manuscript for the CCSM4 model variables.**

200-201: (Optional). Maybe comment on how an autocorrelation time on decadal scale could occur in the climate system?

202: Figure 3: Mention that the scales are not the same for a-d end e-h, respectively.

**Thank you. We have added this remark to the figure caption.**

I guess that noise/signal does not mean the same for annual and decadal data. Is there a way to compare the two more directly? For example for a certain annually resolved data set with a given noise/signal value, what is the corresponding noise/signal value for the same data set averaged to decadal resolution? That would elucidate the dependence of resolution more clearly.

**Unfortunately, it is not generally possible to directly compare noise / signal at different resolutions. Two annually resolved time series with the same noise / signal but different power spectra would have very different noise / signal ratios when down-sampled to decadal resolution.**

**As an example of this, we include below Additional Figure 5 showing the mean estimated noise parameters at both annual and decadal resolution for the "redder noise" and "whiter noise" synthetic transitions discussed in section 3.2. At annual resolution, the "redder noise" case has a lower noise / signal than the "whiter noise" case. However, due to the differing autocorrelation times, when we down-sample to decadal resolution we find the opposite. We have addressed this point in Section 3.2 of the revised manuscript when demonstrating that changing resolution does not change the bias:**

**"We are interested as to how the bias depends on the temporal resolution of the data. Although in Section 3.1 we have considered both annual and decadal resolution, it is far from trivial to make direct comparisons between the two. For climate time series, the changes in the noise parameters $\xi$ and autocorrelation time when changing resolution depend on the full power spectrum of the noise. This is different for every variable or proxy and so there is no generally applicable relationship with which to predict the impact of changing resolution. Furthermore, as previously stated we do not have access to sufficient annually resolved data from the CCSM4 model to allow for a meaningful comparison."**

[Figure]

***Additional Figure 5: The noise parameters of "whiter" and "redder" noise cases at (a) annual resolution and (b) after being down-sampled to decadal resolution. Also included are the dependencies of the bias on the two noise parameters, with other transition parameters fixed, showing that the bias is unchanged by down-sampling to a lower resolution. This is shown here solely to address the referee's comment, and does not feature in our revised manuscript.***

206: For the tests in C1-C4: Are these tests run on the same synthetic data? How exactly were the synthetic data generated, and what value of the autocorrelation length was applied? Synthetic data with characteristics spanning a range of parameter values are tested, but without describing how the ranges compare to those of the real data, the results are hard to interpret in a useful way. For the test on Figure C2: Is this relevant given that Capron did not use data with decadal resolution? Or am I misunderstanding the caption: "transitions with decadal resolution"?

**We apologise for not sufficiently explaining in our initial manuscript how this data was generated. As with all the synthetic data in this study, we add AR(1) noise to a deterministic linear ramp. When not explicitly being varied, all parameters are fixed at standard values as given on page 12 of this response. For the autocorrelation time, the standard values are 1 year for annual resolution and 10 years for decadal resolution. The tests in figures C1-C4 are not run on the exact same synthetic data, but each is run on a large set of synthetic data that is generated in exactly the same manner, and so any slight variations due to the random noise generation should be averaged out. Specifically, each data point represents the average over 100 synthetic transitions, as in Figure 3.**

**In Figure C2 we briefly assess the impact of using an alternative set of priors as suggested by Capron et al. Although annual resolution would be a closer reflection of the data used by Capron et al., for our purposes of simply checking that the bias is still present we do not feel that this is overly important. This is especially so as we demonstrate elsewhere that changing resolution does not change the bias.**

214-216: Please provide the details of the autocorrelation range observed in the model data. What noise-to-signal characteristics were chosen? It's fine to have the results illustrated in Fig. 3, but they should also be tabulated.

**We have included this information in Table D1 in the appendix.**

225: It would be clearer to integrate the section starting at 230 with this section.

**We agree. The first two paragraphs of section 3.3 are now:**

**"Following on from the above systematic testing, we investigate whether the ramp fitting method introduces any bias to the timing estimation of DO events in the CCSM4 model. To do this, we construct 1000 synthetic transitions that are "analogous" to each of the model variables of interest and assess the corresponding expected PMOTE as described above. The expected PMOTE serves as a first order approximation of the bias for each investigated climate variable and may thus be subtracted from the transition onset estimate obtained with the Bayesian ramp fitting method. To calculate representative parameter values we 290 take both the posterior-mean and the event-mean over the corresponding marginal posterior distributions obtained by applying the extended ramp fitting method to the 19 transitions in the CCSM4 model simulations. For each variable, this gives a single value for each transition parameter (shown in Fig. 3), which we use to create the "analogous" transitions.**

**We make an exception for transition duration, choosing instead to use a fixed duration of 50 years for all of our "analogous" transitions. This is because we suspect that much of the apparent difference in duration between different variables is in fact due 295 to the bias we have identified. As we have been unable to identify any unbiased method by which to estimate the true transition durations, we simply fix them at a plausible value based on the durations of the transitions in two ice-cores as estimated in a previous study (Capron et al., 2021)."**

241: Please explain what you mean by "reliability".

**We agree that this indeed was not clear. We have reformulated this statement as follows:**

> **"We find that this has a negative impact on the success rate of the Markov Chain Monte Carlo (MCMC) sampler that forms part of the method, in that there are many more cases where the sampler fails to converge, and so we do not attempt to use this implementation to assess the timing. Instead, it simply provides a means of calculating appropriate parameter values to use for the synthetic transitions that are "analogous" to precipitation."**

245: Please provide table with the characteristics of the synthetic data that resemble the ice-core data sets.

**These are in Table D2 of our revised manuscript.**

265: (Figure 5). It is not clear whether the Calcium data have also been corrected for bias and how this factors into Fig. 5: Is the zero defined from corrected or uncorrected Calcium data ramps? Please provide the details.

**Yes, indeed. That is an important bit of information that was missing here, and we have included it in the caption of Figure 5 and likewise in Figure 4. The transparent violin plots display lag distributions after bias-correcting both the reference variable and the target variable.**

303: I don't see why this is "another issue". It is quite clear that observing a too long duration is equivalent to the combination of a too early onset and a late end point, and as stated in the introduction, I would claim that both effects are likely to occur when you turn up the noise.

**We believe that there is a misunderstanding here. The term 'another issue' expresses that the bias correction suffers from different issues. We are at this point not talking about 'issues' in the estimation procedure. We have shown that the bias depends on the transition duration. So if we wanted to correct the bias in the application of the method, we would need an accurate estimate of the duration of the transition under study. However, the biases in the method themselves make such an estimate impossible, thus limiting the ability to use bias correction.**

**We hope that slightly rephrasing the sentence as follows has made things clearer:**

**"Another issue limiting the possibility for bias correction is our inability to estimate the transition duration in an accurate manner."**

312-15: Section 3.2 is based on CCSM4-like data, and it is not clear if the same result holds for paleoclimate data.

**The statement is indeed misleading. However, Section 3.2 is based on synthetic data and the conclusion that increasing resolution does not remove the bias is thus also equally valid for paleoclimate data. We have slightly modified the statement:**

**"But, as Section 3.2 shows, the temporal resolution of DO event paleo-data has no impact on the bias in the ramp fitting method."**

**and written in our revised version**

**"But, as Section 3.2 shows, the temporal resolution of the data comprising a transition has no impact on the bias in the ramp fitting method."**

Please clarify/rephrase. Throughout: Please add hyphens consistently to "sea-ice extent", "ice-core records" and other similar compound adjectives. It's somewhat a matter of style, but generally increases readability.

**Thank you for this suggestion to help increase readability. We have adopted this style throughout.**

**Referee #3**

**The authors would like to thank the referee for their extremely thorough comments and suggestions. The referee's comments are normal text, and our responses follow in bold.**
* * *
Review of Slattery et al.
In their manuscript Slattery et al. present an analysis of the performance of a modified ramp-fitting method for the determination of the onset of stadial to interstadial transitions and apply this modified method to decadal resolution model data.
Overall, the topic of the manuscript is of interest to the wider paleo-climate community and especially the detailed analysis of interstadial onsets in model output is interesting and a good fit for CP However, there are some larger issues with the manuscript concerning the

conceptual background as well as to the presentation of the results that need to be worked out before the manuscript should be considered for publication.

General remarks
The authors extent the ramp-fitting method of Erhardt et al. (2019) (E19) by including two extra parameters for the pre- and post-transition slope of the data. This is necessary to better describe the model-output that the authors want to apply the method to, especially the AMOC strength. The authors do not show, if the inclusion of the slopes into the model is necessary for all their data and also do not provide any evidence that the inclusion of the slopes decreases the uncertainty of the estimates for the relevant parameters.

**We agree that we did not make it sufficiently clear in our initial manuscript that the inclusion of slopes is either necessary or beneficial. We have included in Appendix A of our revised manuscript detailed comparison of the original and extended implementations of the ramp fitting method, including Figure A1 which demonstrates that the inclusion of slopes results in a reduction in the uncertainty for cases where slopes are in fact present, but an increase in uncertainty when no slopes are present.. We also give the averages and ranges of the different parameters (including slopes) for all of the model variables and ice core proxies in Tables D1 and D2. We hope that this demonstrates why including slopes is necessary and beneficial in many cases. We thank the referee for prompting us to consider this in more detail.**

One would also hope, that the addition of two extra parameters to the model would make the model consistently outperform the original E19 version. Judging from the figures in the appendix this does not necessarily be the case. However, due to presentation of the results that is difficult to judge.

**We agree that this is an important point to discuss, however we disagree that the addition of two parameters should consistently improve performance. For cases where slopes are present then clearly this is the case. However, when no slopes are present, the addition of two unnecessary parameters may plausibly lead to worse performance. As seen in Figure A1 of our revised manuscript / Additional Figure 6 below, this is exactly what we observe.**

**For regions of parameter space with zero or only slight slopes during the preceding stadial and following interstadial (e.g. the middle band of panels a & b, and the bottom region of panels c & d), the original method outperforms the extended method, in the sense of having lower uncertainty in the onset time. However, for regions of parameter space with more substantial slopes the extended method performs better than the original.**

[Figure]

*Additional Figure 6: The uncertainty in the onset time, depending on the noise noise / signal and slopes, for both the original (a,b) and extended (c,d) ramp fitting methods. The uncertainty is defined as the width of the 90% credible interval for the onset time. This is included as Figure A1 in the appendices of our revised manuscript.*

Before continuing, a note here on terminology: The authors chose to coin a new term for the average value of the posterior samples. This is unfortunate, as there already is a term that is used for this: posterior mean or more specifically the marginal posterior mean. This is the term that I will use and that I strongly urge the authors to use, too.

**We have adopted this suggestion.**

In an extensive sensitivity study, the authors test if the posterior mean of the transition onset is on average an unbiased estimator of the onset of the transition. Most, if not all, of the content of the paper hinges on the results of this sensitivity study. Despite that, the authors

never explicitly state the exact focus of the sensitivity study and the implications that come with this choice. However, that exact focus of the tests is a big potential issue: In the sensitivity tests, the marginal posterior mean does not need to equal to the known parameter value and same is true for any other point estimate from the full marginal posterior. If the posterior is skewed or multi-modal then any point estimate will grossly misrepresent its true location and disregard its shape. Especially if skewness of the marginal posterior is systematically induced due to the noise or due to trends around the transition, it can introduce a systematic bias into the marginal posterior mean.

This can for example be seen in Figure 2 of E19, where the two marginal posterior distributions for the start and the end of the transition show a clear indication of skewness and in the case of the onset maybe even bi-modality. The same argument is of course true for the length of the transition and also extends to all other estimated parameters.

Sune Olander Rasmussen (SOR) raised a not dissimilar issue in his review: By using the marginal posterior mean the authors disregard the sometimes large uncertainty in the estimated parameters that is carried in the complete marginal posterior distribution. As pointed out by SOR, the width of the marginal posterior distributions can easily be larger than the reported bias of the posterior mean. From my point of view, this sole, narrow focus on the marginal posterior means is in stark contrast to the strong general statements on applicability of the method throughout the paper, especially because this tight focus is not explicitly communicated to the reader.

**The referee is correct in that we put strong emphasis on the posterior mean. The point they make, that this is not necessarily a good point estimate of the true value, is also valid and important. However, we remain convinced that the bias of posterior mean with respect to the true value is a good indication of how lead and lag estimates derived from a set of events are corrupted by the inaccuracies of the method.**

**The key quantity we would like to probe is the degree of bias that will be propagated to calculations of leads and lags. As there are different proposed methods of calculating such leads and lags (e.g. Erhardt et al. 2019, Riechers & Boers 2021), it is not immediately obvious whether the posterior mean or posterior median is a better indicator of this bias. We include Additional Figure 3 (page 15 of this response) in the appendices of our revised manuscript (Figure C3) to demonstrate that we find very similar results when considering the median as we do when considering the mean. Given this, we feel comfortable in stating that the centre of the probability distribution is shifted away from the true value, which is what we term a bias, and that this will necessarily be propagated to the calculation of leads and lags.**

**We apologise for failing to address the relative size of the bias and the width of the marginal posterior distribution. As also mentioned in our response to Sune Olander Rasmussen (Referee #2), we agree that our bias is relatively unimportant when considering individual events due to the large uncertainty in the timing. However, when calculating leads and lags this uncertainty is reduced by stacking many events, and so our bias then becomes extremely important. This is explained in our revised conclusion as follows:**

**" The Bayesian ramp fitting method considered in this study remains a powerful tool for investigating individual abrupt transitions, as the bias that we find is small relative to the uncertainty of individual events. When calculating leads and lags from a sample of events, however, the bias becomes large relative to the uncertainty and so is likely to lead to false conclusions."**

In addition to the conceptual shortcomings of the sensitivity study, the presentation of its results in Figure 3 and in the appendices is also not ideal. If I guess correctly (there is nothing written in the manuscript), the authors average over the variables that they do now show to present the data in a reduced number of dimensions. The main issue with this approach is, that it hides any possible inter-dependencies between variables and does not allow for clear separation of the effect of the single variables. I think this is a missed opportunity and might be potentially misleading, depending on the parameter interdependencies.

**We thank the referee for pointing out that we have failed to sufficiently describe how the synthetic data in Figure 3 is generated. We have added the following in Section 3.1 our revised manuscript:**

**"We would ideally like to vary all five transition parameters simultaneously in order to assess possible inter-dependencies of the bias on different transition parameters. However, it would be challenging to visualise and interpret the resulting five-dimensional parameter-space. We therefore vary only two transition parameters at any one time - Noise / Signal Ratio and one other - whilst keeping the rest fixed at standard values as follows:**

- **Autocorrelation Time = 10 years for the case of decadal resolution and 1 year for the case of annual resolution.**
- **Greenland Stadial Slope = 0 Kiloyears$^{-1}$**
- **Greenland Interstadial Slope = -1 Kiloyears$^{-1}$. Note that the Greenland Interstadial Slope is always <= 0 in our sensitivity tests due to the sawtooth shape of DO events. In Figure 3 we plot the absolute value or magnitude of this slope, which is positive. Note that the slopes are normalised relative to the amplitude of the transition. This standard value for the Greenland Interstadial Slope therefore means that the time series will return to its pre-transition level 1000 years after the transition ends.**
- **Transition Duration = 50 years."**

**We have also included the following in the caption of Figure 3:**

**"Each panel contains 100 unique sets of parameter values, and for each set of parameters we take an event-average over 100 synthetic transitions. When not explicitly varied, parameters are fixed at standard values as given in Section 3.1."**

The authors apply their modified ramp model to climate model output of DO-type transitions that is available in decadal averaged output. For this the authors chose to stay with an AR(1)

noise term for the application of the ramp model. I am, like SOR, a bit puzzled by the long autocorrelation times in Fig 3e. Of all the parameters only the ocean circulation would likely have long-term memory, of all the other parameters, I would assume that decadal averages would mostly eliminate the autocorrelation.

I would ask the authors to provide a more detailed explanation of this phenomenon and a discussion of the influence of the long autocorrelation times on the timing estimates. Generally the autocorrelation of the residuals in the model is in competition with the transition ramp itself, for lack of a better term. If the model fails to describe the noise of the data accurately then the model will give unreliable results. Long autocorrelation times can smear out the transition for sure leading to larger uncertainties and maybe even "biases" towards earlier or later transitions. This can make the choice of the noise model and the choice of priors for its parameters very important. In the case of the presented study, the authors explicitly excludes the case for non-autocorrelated data and put, in comparison to E19, larger prior probability on long autocorrelation times.

Especially the former might be a serious issue in case of the decadal averaged data from the model and merits a thorough discussion.

**This is an interesting point, and we thank the referee for prompting us to consider it. We agree that the long autocorrelation times of the decadal resolution model output data are rather confounding. After some investigation, we believe that this is an inherent result of approximating climate variability as an AR(1) noise process, rather than being because of our choices of priors.**

[Figure]

***Additional Figure 7: The posterior mean values of the autocorrelation time tau and the equivalent autocorrelation coefficient alpha plotted against temporal resolution for the 16***

*studied DO events in the NGRIP $\delta^{18}O$ record. This is included to address the referee's comment, and does not feature in our revised manuscript.*

**To demonstrate this, we consider the NGRIP $\delta^{18}O$ data, which ranges in temporal resolution from 4 to 7 years over the different DO events going back to 60ka. When estimated using the original Erhardt et al. implementation, we find a significant positive correlation (p<0.05 assessed using a standard linear regression) between the autocorrelation time $\tau$ and the temporal resolution. However, we find no such**

**correlation between the temporal resolution and $\alpha = e^{\frac{-dt}{\tau}}$ , where $dt$ is the temporal resolution. This is shown in Additional Figure 7. This pattern whereby there is a significant correlation between resolution and autocorrelation time, but not between resolution and alpha, is repeated for all four considered proxies and using both the original and extended methods. This is therefore a robust finding that is not due to the quirks of any particular proxy or of any particular implementation of the ramp fitting method.**

**One would certainly not expect to find any meaningful autocorrelation at a lag of e.g. 10 years in annually resolved climate data. However, the situation here is subtly different: We are in fact saying that there is autocorrelation between the means in successive decades, e.g. of years 1-10, 11-20, 21-30, etc. Taking decadal means in this way largely eliminates the annual variability, and so the AR(1) noise mostly represents multi-annual or decadal modes of variability, which we fully expect to have autocorrelation times of a decade or more. Whilst these long autocorrelation times are certainly initially surprising, we therefore do not feel they are indicative of an issue with our approach.**

**The referee is correct to state that our choice of prior puts larger probabilities on longer autocorrelation times, in comparison to E19, and that we explicitly exclude the possibility of zero autocorrelation. However, in comparison to E19 we also put a greater prior probability on short autocorrelation times (between 0 and 10 years), and there is no difference between the cases of an infinitesimally small autocorrelation time and zero autocorrelation. Ultimately, Figure 3e shows that there is no significant link between the autocorrelation time and the bias for the case of decadal resolution, so even if our implementation does overestimate the autocorrelation time this cannot be the source of the bias.**

To overcome the bias that the authors discovered, they use "analogous" data to estimate the bias and later correct for that. Generally the authors do not provide much detail about the exact parameters that they use to generate the "analogous" datasets that they use to make the bias correction for the different parameters.

It would be great if the authors could provide more detailed information about the parameters that they chose such that the results are later reproducible. Their statements about the fact that they cannot reliably estimate the onset or the length of the transition given their methods makes me also wonder how they estimated all the parameters that they need for the "analogue" data.

Did they use a different method or can they use the ramp-fitting method to estimate all other parameters needed for the surrogate data reliably? Especially in light of the statement by the authors that the relationship of the bias to the ramp parameters is complicated it would seem to be necessary to estimate this parameters 2very precisely or carry out the bias calculations for a range of possible parameters. I strongly urge the authors to investigate this in more detail before making definitive claims about a bias.

**We apologise for failing to provide this information. We include Tables D1 and D2 in the appendices of our revised manuscript which provides all of the parameters for the "analogous" synthetic transitions, as well the range of parameters seen for each variable and proxy, to justify our choice of ranges in Figure 3. Other than the duration, which is simply fixed at a plausible value, all of the other parameters are estimated by taking the mean over all available events of the posterior means given by our extended ramp fitting method. We make this more explicit in our revised manuscript.**

**We considered estimating the bias for a range of possible parameters, as the referee suggests, but in the end we feel that it is sufficient to demonstrate that there is a significant bias given a plausible set of parameters for each variable or proxy. The bias will in fact be different for every individual DO event, but to estimate this bias for every individual event, for each variable and proxy, would add massively to the complexity of our analysis whilst, in our view, not meaningfully changing our findings.**

The proposed bias correction also carries additional uncertainty with it, how do the authors propagate that to the estimates later?

**We do not propagate the uncertainty in the parameters for each set of "analogous" transitions to the estimates of the bias. We apologise for not making this clear and are careful to do so in our revised manuscript. We have added a line to the caption of Table 2 of our revised manuscript stating:**

**"The uncertainties provided here reflect the uncertainty in the bias given a particular set of transition parameters. There is additional, unquantified uncertainty due to our imperfect knowledge of the true transition parameters."**

In the analysis of the modelled precipitation data, the authors state, that the noise characteristics of the data change over the transition. The authors somewhat disregard this observation with the notion that they failed to include this in the model. However, the results of the analysis are dependent on the noise model adequately describing the data. This is especially worrisome as the precipitation time series exhibits the biggest lead over the temperature in the model data. The fact, that the authors state that inclusion of changing noise characteristic over transition negatively impacts the reliability of the method might hint at a deeper problem. This makes me question the results that they present and the applicability of the estimates for the generation of the "analogous" data and thus the bias correction that the authors apply.

**We agree that differing noise characteristics between the stadial and interstadial present a problem. We also agree that our estimate of the bias in precipitation is**

definitely flawed, but we make this clear and it is the best estimate we are able to make. In our view, it is sufficient to demonstrate that the lead of precipitation over temperature could plausibly be the result of bias. Although we certainly cannot prove that this is the case, it also cannot be ruled out, and so the apparent lead is therefore not reliable. One does not need to know the exact size of the bias to demonstrate that such a bias exists.

Specific remarks

58–63
Order of publications is wrong.

**Apologies, we have corrected this.**

Table 1
This suggests a connection between Ca and the NAO, please justify or remove

**We have removed any suggestion of connections between model output variables and ice core proxies, as these are not important for our study.**

125ff Focus of the work: If that is the case, please rephrase the introduction and the abstract accordingly. In this case you can also remove all the model data. Or spend some time discussing the data and its interpretation. There is no need for statistics without a clear question to answer.

**As also mentioned in our response to the other referees, we no longer make any identification between proxies and climate variables. The focus of our work is indeed on the robustness of the method, but this can only be judged with respect to the kind of data that the method is designed to be applied to. We feel that the model data is a useful example of a potential application that allows us to demonstrate how the bias could lead to a false conclusion.**

138ff too. This is a wrong statement. If that were true then OLS-regression would be probabilistic,

**Our apologies for this misleading statement. We instead write the following:**

> **"The addition of the AR(1) noise process $\epsilon_i$ makes the model stochastic. For a given time series that comprises a DO event, the introduction of appropriate prior distributions directly yields Bayesian posterior distributions for the model parameters $\{t_0, t_1, x_0, x_1, \alpha, \sigma\}$, where $\alpha$ and $\sigma$ define the autocorrelation and the amplitude of the noise."**

143ff Please show proof of this.

**We explain this as follows in our revised manuscript:**

**"Additionally, for cases where slopes are present our extension of the method reduces the sensitivity of the transition timing to the search window, which is otherwise one of the drawbacks of this method (Capron et al., 2021). In order to apply the ramp-fit to individual transitions, one has to select a data window around that transition. There are no objective criteria for the starting point and the endpoint of these windows, other than that no other transition should be included. However, changing the boundaries of the data window influences the results of the estimation. This effect is reduced if slopes before and after the transition are allowed. For example, for an arbitrarily chosen transition of AMOC in the CCSM4 model, we applied both the original and extended ramp-fitting method to the same transition using 25 different search windows and found that the standard deviation in the posterior mean onset time decreased from 16.0 years when using the original method to just 4.3 years when using the extended method (Figure A2)."**

149 Remove reference to Erhardt et al., as they do not use any frequentist methods.

**We have removed this reference.**

169ff Nomenclature: There are some already established terms that the authors should use as much as possible instead of inventing new ones: the average of the posterior distribution is the posterior average or posterior mean.

**We thank the referee for pointing this out and have adopted their suggested terms.**

Fig 3 The presentation of the results is unclear. The authors do not state what they do with the other parameters that were varied. I assume that they just averaged over all the other parameters that are not shown in the figure.

However this approach will hide any co-dependencies of the bias between different parameters.

**We have made it clear in our revised manuscript that parameters were fixed at standard values unless explicitly varied.**

213ff : Please specify how you set these two sets of data up. I would imaging that the exact difference between the two resolutions depends a lot on the parameters as well as the timing of the onset, depending if it happens at a boundary of an averaging period. The results here are also contradicted a little by Fig C1, which shows clear differences between annual and decadal resolution data

**We agree that this was not made sufficiently clear. We take the mean over each decade of the annual data (e.g. years 1-10, 11-20, etc) and these form the data points of our decadal data. Having performed further tests, we have verified that the bias does not depend on whether the onset happens at the boundary or in the middle of an averaging period. This is explained as follows in Section 3.2 of our revised manuscript:**

**"Instead, we proceed by creating two further sets of synthetic transitions at annual resolution. These two sets of transitions represent two contrasting cases of low-autocorrelation ("whiter") noise or high-autocorrelation ("redder") noise, with parameters chosen based on the extremes of the range observed in the CCSM4 model. For each of these sets of synthetic transitions, we then down-sample to decadal resolution by dividing the annual time series into sections of 10 years and taking the meanover each section as a single data point for our decadal time series. (Fig C4). Finally, we separately apply our extended ramp fitting method to the annually and decadally resolved transitions. For both types of noise we find that the bias is unchanged, within uncertainty, when switching between the two resolutions. These results are summarised in Table 1. We also verify that when down-sampling to decadal resolution it does not matter whether the transition onset occurs at the edge of two averaging-sections or in the middle of one. We can therefore state that the temporal resolution has no impact on the bias, although higher resolution does at least reduce the uncertainty in the onset time estimates for individual transitions."**

Table 2 The bias of what? Specify in the table header.

**We apologise for not making this clear. This is the bias in the posterior mean onset time.**

Sect 3.3 Please specify the exact parameters used to generate the analogous transitions. You say that you cannot reliably estimate the transition length, but you do not show that you can actually estimated the other parameters without issues. I think this is minor but a few sentences in this regard would be good.

**The exact parameters of the analogous transitions are given in Tables D1 and D2 in the appendices of our revised manuscript. Following the suggestion of referee #2, we also include tests of the performance regarding the estimation of all other parameters as Figure A3.**

230ff It is unclear how the posterior mean and event-means are set up here. Are you generating 19 events or are you subsampling 19 events out of your 1000 realisations to calculate the event- averaged timing?

**We apologise that this was unclear. We have 19 abrupt warming events from the studied set of climate model simulations. We take the means over these 19 events (event-means) of the posterior means of each parameter. We have ensured that this is made clear in our revised manuscript by rewriting the opening of Section 3.3 as follows:**

**"Following on from the above systematic testing, we investigate whether the ramp fitting method introduces any bias to the timing estimation of DO events in the CCSM4 model. To do this, we construct 1000 synthetic transitions that are "analogous" to each of the model variables of interest and assess the corresponding expected PMOTE as described above. The expected PMOTE serves as a first order approximation of the bias for each investigated climate variable and may thus be subtracted from the transition onset estimate obtained with the Bayesian ramp fitting**

**method. To calculate representative parameter values we take both the posterior-mean and the event-mean over the corresponding marginal posterior distributions obtained by applying the extended ramp fitting method to the 19 transitions in the CCSM4 model simulations. For each variable, this gives a single value for each transition parameter (shown in Fig. 3), which we use to create the "analogous" transitions. We make an exception for transition duration, choosing instead to use a fixed duration of 50 years for all of our "analogous" transitions. This is because we suspect that much of the apparent difference in duration between different variables is in fact due to the bias we have identified. As we have been unable to identify any unbiased method by which to estimate the true transition durations, we simply fix them at a plausible value based on the durations of the transitions in two ice-cores as estimated in a previous study (Capron et al., 2021)."**

239ff Please elaborate on your the heteroscedacity issue: If the method is unreliable with the inclusion of different noise levels over the transition, than how is it reliable enough to estimate analogous parameters. It is also unclear if you generated the analogues with or without heteroscedacity which makes the last sentence of the paragraph quite confusing. Please rework. It also needs to be clearly shown and discussed whether the bias correction for the modelled precipitation is valid and if the model can be applied at all to the heteroscedastic data. The issue with these models is that their results are conditional on the model being a good description of the data, if that is not the case the results are to be taken only with a grain of salt and proper justification.

**For precipitation, we generate two sets of analogous transitions: one with heteroscedacity and one without. We have rewritten this paragraph as follows to make this clear:**

**"For completeness, we construct two sets of synthetic transitions that are "analogous" to precipitation: One where we do not account for the differing noise between stadials and interstadials, and one where we do. Our estimates of the magnitude of the 310 bias are increased by around 2 years for the original method and around 3 years for the extended method when we include the two distinct noise regimes in our synthetic transitions (see Table 2). This confirms our expectation that failing to do so would lead to an under-estimate of the degree of bias"**

**Although we agree that the way we have estimated the noise parameters on either side of the transition is imperfect, we feel it was the best approach to take in the circumstances.**

**We also agree that the model is only valid if the model is a good representation of the data, and that this is not the case when dealing with heteroscedacity. In our view this only reinforces our point that leads and lags estimated using such a transition model are unreliable. After all, heteroscedacity is also present to some extent in most proxies across DO events, e.g. Capron et al. mention that the calcium record shows greater noise during stadials than interstadials, and so this issue would equally affect all the previous studies based on this transition model.**

Tab 3 : Please do not format the table in such a way that Ca seems to be associated with the NAO. This is potentially misleading especially as you shy away from discussing the proxy interpretation at all. Also, Even though temperature and precipitation over the North Atlantic that you use is probably not unrelated to the values at NGRIP, they are not the same. Please redo the analysis with the relevant variables or rework the manuscript in such a way that it is clear that you are not looking at model output for Greenland at all. This included the Table where these are stated next to each other as if they are the same.

**As above, we have removed any suggestion of connections between model variables and proxies.**

255ff Please discuss why the lags are so different between the different variables. Is that because of the shape of the transitions or the noise levels or both?

**The differences in the lags are due to both different noise characteristics and differently shaped transitions - it is difficult to separate the two.**

Sect. 3.5 Again there is no clear description on how the analogous transitions were generated. Without that information the results cannot be reproduced or verified by anyone.

**We apologise for this. We provide the relevant parameters in Tables D1 and D2 and also make available the code that was used to generate the synthetic transitions.**

281ff I think the leading paragraph is a bit misleading without the actual information under which circumstances the bias arises.

**We politely disagree. The cause of the bias is unknown, and so it is extremely difficult to prove that any apparent time lags are not caused by bias. This certainly reduces the trustworthiness of any leads or lags found using this kind of method.**

285ff This paragraph probably should state somewhere that you look at this from a frequentist point of view. Erhardt et al. never made any claims about statistical significance, they provide credible intervals which are the parameter ranges that are consistent with the data under the assumptions of the model. This is not the same as the assumptions that go into the significance test, namely that under repeated observations and calculation of the significance interval of 95% the true value of the parameter will fall into the significance interval 95% of the time. These two views are fundamentally different and are not necessarily compatible with each other.

**We make clear that we adopt a frequentist perspective.**

318ff The paper mostly focuses on the adjusted method that you are proposing, and not the original method of E19. Please rephrase as all results with regard to the original method are only in the appendix of the paper.

**We include in our revised manuscript additional results which clearly demonstrate that the original method of E19 is also biased (see Additional Figure 2 and Additional**

**Table 1 on page 10 of this response). In particular, this is made clear in Figure 3 and Table 2 of our revised manuscript.**

327f The conclusions of Capron et al. (2021) are based on fundamentally different reasons than the once here. Your's are purely related to the method and the interpretation of the results whereas those of Capron et al. (2021) are based on considerations with respect to climate variability and the tight coupling between different parts of the climate system. Please remove or rephrase.

**There is certainly a connection between the conclusions drawn by Capron and ours: Capron et al. state that either the elements of the climate system were too tightly coupled to detect significant leads and lags, or that individual DOs followed different mechanistic patterns. In the first case, one could still imagine that there existed a method that was able to detect phase relations of the transitions. So they implicitly say that the elements of the climate system are 'too tightly coupled to disentangle a characteristic sequence with the available methods.' We show that the Bayesian ramp-fit indeed cannot do this job, underpinning their assessment.**

329ff The conclusion is inconsistent. The method is either good or not for the investigation of DO events.

**We apologise for being unclear. We mean to say that, even though we have demonstrated a flaw in this method, there are none of which we are aware that can do a better job. We have rewrite the conclusion to be clearer:**

**"The Bayesian ramp fitting method considered in this study remains a powerful tool for investigating individual abrupt transitions, as the bias that we find is small relative to the uncertainty of individual events. When calculating leads and lags from a sample of events, however, the bias becomes large relative to the uncertainty and so is likely to lead to false conclusions."**

Please decide and rewrite. Please also state why you think that the method would be suited if the lags are larger than 20yrs. This is not entirely clear given the discussion beforehand.

**We accept that the value of 20 years is not sufficiently supported, and we cannot give an exact value for the minimum time lag that we can be sure is not due to bias. We have rephrased this sentence as following to be more accurate:**

**"It therefore appears impossible to reliably determine the temporal phasing when dealing with decadal-scale time lags such as those that have been suggested for DO warming events."**

Supplement Choice of prior distributions

Either Equation 10 or your code is not correct, please check.

**We have not been able to find the mistake that the referee has apparently identified. On lines 102-103, we return 0 likelihood unless $0 < \sigma \le |dy|$. Otherwise, we do not**

**explicitly define a prior to sigma, which we believe is equivalent to a uniform prior. If we are mistaken then we would be very grateful if the referee could explain more precisely what our error is and correct us.**

The prior for tau in your analysis puts a lot more emphasis on much longer and much shorter autocorrelation times than the one in the original method. This is a bit problematic as we do not expect to see decades long autocorrelation in climate data as far as I know. Please elaborate. Sampler setup Please clearly describe the setup of the ensemble sampler. I have noted that you chose to keep the setup of E19 despite increasing the number of parameters. This should also go hand in hand with a change in the sampling strategy i.e. burnin and thinning and maybe even the number of ensemble walkers. These need to be carefully chosen to not get nonsensical results in the end.

**Our choice of prior for the autocorrelation time was designed to be less prescriptive whilst retaining the same peak of the probability distribution as in E19. As discussed on pages 30-32 of this response, we do not feel that the long autocorrelation times for decadal data are necessarily indicative of an issue.**

**Regarding the sampling strategy, we found that the setup of E19 was still valid and so did not see a need to change it. We manually inspected a subset of our synthetic transitions for convergence, and found only very rare occasions where the sampler failed to converge.**

Figures C1-C3 Judging from a closer comparison of the results from your method to the results of the original E19 method in Figures C1 and C2 it would seem that the addition of the slopes generally decreases the performance of the method. This seems to be the case for the sensitivity of the bias towards noise, the transition length and very, very worryingly towards the inclusion of an interstadial slope. Please elaborate and discuss this especially in light of the analysis that you base upon the result from the fit including the slopes. Also it would seem, that at zero slopes prior and after the transition the original method overall outperformed yours, which would mean that part of the bias is because of the inclusion of the slopes into the model not despite.

**We thank the referee for raising this important point. We agree that this needed to be addressed in our revised manuscript. We go into detail regarding this matter in response to Referee #2 on pages 9-11. Broadly speaking, we find that the addition of slopes results in better performance when slopes are in fact present, but worse performance when they are not - as might be expected. As shown in Additional Table 1 on page 10, for some variables / proxies the original method has lower bias, and for some the extended method has lower bias. Overall, the bias is present in both cases and so our conclusions stand. This is discussed thoroughly in Section 3.1 and Appendix A of our revised manuscript.**

---

## Referee Report (RR1)

**Additional comments to the revised manuscript and the authors' response to the referees' comments to the original manuscript "The Temporal Phasing of Rapid Dansgaard–Oeschger Warming Events Cannot Be Reliably Determined" by John Slattery et al.,**
**https://doi.org/10.5194/egusphere-2023-2496**

The suggested revisions represent improvements that cover most of my concerns. I provide a few additional comments below (in normal typeface following the authors' responses in bold), and in particular, I think the title should be revised to something closer connected to the conclusions. It is, for example, quite unclear what "reliably" means without the context provided in the text (which will hardly fit into the title): as discussed in the reviews and in the updated manuscript, the biases will often be too small to be of any practical importance unless data from many events are stacked or otherwise combined. Also, it's not clear what the mentioned phasing refers to (between events, between proxies …), and "Rapid" is not needed as Dansgaard–Oeschger events always are rapid.

Some suggestions to illustrate my point:
"Methodological biases hamper the detection of climatic leads and lags across Dansgaard–Oeschger events"
"Estimating biases during detection of leads and lags between climate elements/mechanisms across Dansgaard–Oeschger events"

**Furthermore, for the NGRIP ice core proxies, our extended method finds slopes of comparable magnitude in the pre-ramp stadial and post-ramp interstadial. This will be made clear in our revised manuscript as all of the relevant parameters used to create the "analogous" synthetic parameters for the purpose of bias estimation will be listed in a table in the appendix. We most often find that the slope during the pre-ramp stadial is positive - that is to say in the same direction as the ramp itself. See also the mean parameters for each proxy (including slopes) shown on Additional Figure 2 (page 10). The proxies all show significant slopes, in the sense that these slopes lead to a significant bias, however we have not tested the statistical significance of these slopes in isolation.**

Why not?
The observation that allowing a pre-ramp slope changes the ramp location is not surprising in itself. If the observed pre-ramp slopes are not significant, it's not clear that (or how often) the updated model is an improvement. In particular, for non-model data, it remains to be demonstrated that there are significant slopes in the stadials and that the updated model is indeed an improvement and not 'just' another model.

**Capron et al. test 20 realisations of a synthetic ramp, and find no significant evidence of a bias in either the transition midpoint or duration. Neither of these are directly comparable to the transition onset time, which we focus on as we feel it is more physically meaningful when trying to understand the progression of DO events. Nonetheless, if there were a bias in the onset time then Capron et al. would surely have seen this reflected in either the midpoint or duration, and they do not in fact see this. As the referee notes, the uncertainties shown in Supplementary Figure 2 of Capron et al. are large, and so could include a bias of a few years. Even so, the test conducted by Capron et al. would seem to rule out decadal-scale bias of the kind that we find in our study, at least for this particular combination of transition shape and noise. We would therefore suggest that the transitions tested by Capron et al. happen to lie in a region of parameter space for which**

**the bias is small, even at high levels of noise. When using the original Erhardt et al. implementation we find very little bias for synthetic transitions with no pre- or post-ramp slopes (see Additional Figure 2 on page 10 of this response). …**

I think this is a likely explanation. I guess it also means that the bias problem is expected to be relatively small for most of the data sets of transitions derived from ice cores.

**… Although the shape of the deterministic ramps used by Capron et al. is not made clear, it is likely that they are flat before and after the ramp. …**

Indeed, this is the case, and this should be clear from Capron et al.: "i.e., a linear change in the raw or logarithmically-transformed data between two stable states"

**…Instead, we demonstrate that their finding depends strongly on the unstated assumption that there are no pre- or post-ramp slopes. …**

Citing Capron et al.: "The assumption that the transitions are adequately described by a linear change from one stable state to another is not trivial and has been challenged previously, but neither our observations nor the current understanding of the nature of the transitions justifies employing a model with more degrees of freedom."

The authors are very welcome to challenge this assumption, but it is hardly unstated.

Line-by-line comments:

- Line 1 and throughout: Hyphens should be used in "sea-ice extent", "ice-core records" and other similar compound adjectives, but not when "ice core", "sea ice" etc. appear as nouns.
- Line 68. Revise grammar.
- Line 74-76 seems inconsistent with line 12.
- Line 84: Models provide a lot more direct insights into the dynamics than ice-core proxies, but "they provide complete information" seems like an overstatement.
- Line 109: "appears to depends" .. no s or just "depends" … the paper details the event duration's dependence on CO2 quite explicitly.
- Line 109: The "chosen range" range does not make sense. The model oscillates with a range of CO2 values (which is not chosen), but the range of CO2 investigated is broader than this,
- Line 174: Missing "the"
- Caption fig. 1: The last sentence does not apply to a) and d)
- Line 191: in terms in terms
- Line 243: Rather "synthetic data series"?
- Line 279-280: yes, but especially for the Greenland stadial slope, the range of values used in the tests is much larger than the values observed in data, so the effect may look more dramatic than what is realistic.
- Line 307: Challenging
- Line 424: Suggestion: One important caveat is that the bias we have identified is generally fairly small relative to the timing uncertainty of individual proxies across single DO events, …
- Line 427: Suggestion: may involve
- Line 433: It seems unlikely that 20 synthetic data series with different noise realizations would not show any bias if it indeed was a problem. A more likely explanation is given in the authors' comments: "We would therefore suggest that the transitions tested by Capron et al. happen to lie in a region of parameter space for which the bias is small, even at high levels of noise."

- Line 473-477: It would be fair to mention around here – or elsewhere in the conclusion - that (significant) slopes are not always present in the data and that the original model outperforms the extended model in the absence of slopes in the data.

Sune Olander Rasmussen, June 19th, 2024.

---

## Author Response (AR2)

**Authors' response accompanying minor revisions to Slattery et al. (2024) - CP/TC.**

**Referee #1 - Mathieu Casado**

**The authors would like to thank the referee for their insightful and helpful feedback throughout the review process. The referee's comments are in bold, and our responses follow in normal text.**

**This revision of the manuscript about simulating DO variability to evaluate leads/lags between climate parameters has been considerably improved. I suggest to accept the manuscript after taking into the following specific comments.**

**Specific comments:**

**Lines 61 to 65: "Whilst these model simulations are imperfect representations of real DO events, they nonetheless provide an invaluable means to help us investigate the question of whether it is possible to conclusively identify a trigger for rapid Dansgaard–Oeschger warming events. This is because, unlike ice-core proxies, they provide complete information about different components of the climate."**

**The second sentence seems unnecessary, if not inaccurate. First, there are other records to study past climate, including DO events than ice cores, which are ignored, which is surprising. Second, "complete information about difference components" is extremely vague. Third, GCM are not complete, otherwise there would be no need for observations. The first sentence was fine on its own, consider deleting the second one.**

We have removed the second sentence quoted here.

**Lines 66 to 68: "We build upon these recent advances, examining in detail the causes of uncertainty and the question of what can therefore be learned from paleo-archives about the onset time of DO events in different climate elements - or proxies for these - such as temperature, precipitation, atmospheric circulation, sea ice, and AMOC."**

**This sentence should be streamlined and simplified, or maybe even omitted.**

We have streamlined this sentence to the following: "We build upon these recent advances, examining in detail the causes of uncertainty in the onset time of DO events in different paleoclimate proxies and model variables."

**Referee #2 - Sune O. Rasmussen**

**The authors would like to thank the referee for their insightful and helpful feedback throughout the review process. The referee's comments are in bold, and our responses follow in normal text. Where the referee has quoted our previous responses / revisions, these are in bold italics.**

**The suggested revisions represent improvements that cover most of my concerns. I provide a few additional comments below (in normal typeface following the authors' responses in bold), and in particular, I think the title should be revised to something closer connected to the conclusions. It is, for example, quite unclear what "reliably" means without the context provided in the text (which will hardly fit into the title): as discussed in the reviews and in the updated manuscript, the biases will often be too small to be of any practical importance unless data from many events are stacked or otherwise combined. Also, it's not clear what the mentioned phasing refers to (between events, between proxies …), and "Rapid" is not needed as Dansgaard–Oeschger events always are rapid. Some suggestions to illustrate my point:**
**"Methodological biases hamper the detection of climatic leads and lags across Dansgaard–Oeschger events"**
**"Estimating biases during detection of leads and lags between climate elements/mechanisms across Dansgaard–Oeschger events"**

We agree that the previous title does not reflect the conclusions well. We adopt a form of the referee's second suggestion:

**"**Estimating Biases During Detection of Leads and Lags Between Climate Elements Across Dansgaard–Oeschger Events**"**

We also slightly adjust the abstract to reflect this change: "Dansgaard--Oeschger (DO) events occurred throughout the last glacial period. Greenland ice-cores show a rapid warming during each stadial to interstadial transition, alongside abrupt loss of sea ice and major reorganisation of the atmospheric circulation. Other records also indicate simultaneous abrupt changes to the oceanic circulation. Recently, an

advanced Bayesian ramp fitting method has been developed and used to investigate time lags between transitions in these different climate elements, with a view to determining the relative order of these changes. Here, we critically review this method in both its original implementation and a new, extended implementation. Using ice-core data, climate model output, and carefully synthesised data representing DO events, we demonstrate that both implementations of the method suffer from biases of up to 15 years. These biases mean that the method will tend to yield transition onsets that are too early. Further investigation of DO warming event records in climate models and ice-core data reveals that the biases are on the same order of magnitude as potential timing differences between the abrupt transitions of different climate elements. Additionally, we find that higher-resolution records would not reduce these biases. We conclude that decadal-scale leads and lags between climate elements across DO events cannot be reliably detected, as we cannot exclude the possibility that they result solely from the biases we present here."

*Furthermore, for the NGRIP ice core proxies, our extended method finds slopes of comparable magnitude in the pre-ramp stadial and post-ramp interstadial. This will be made clear in our revised manuscript as all of the relevant parameters used to create the "analogous" synthetic parameters for the purpose of bias estimation will be listed in a table in the appendix. We most often find that the slope during the pre-ramp stadial is positive - that is to say in the same direction as the ramp itself. See also the mean parameters for each proxy (including slopes) shown on Additional Figure 2 (page 10). The proxies all show significant slopes, in the sense that these slopes lead to a significant bias, however we have not tested the statistical significance of these slopes in isolation.*

**Why not? The observation that allowing a pre-ramp slope changes the ramp location is not surprising in itself. If the observed pre-ramp slopes are not significant, it's not clear that (or how often) the updated model is an improvement. In particular, for non-model data, it remains to be demonstrated that there are significant slopes in the stadials and that the updated model is indeed an improvement and not 'just' another model.**

We agree that this was not sufficiently demonstrated before. To put this beyond all doubt, we have explicitly tested the significance of the stadial slopes for the NGRIP data. We include the following as a supplementary figure in Appendix A, along with the paragraph quoted below:

[Figure]

Figure A1: Slopes in the pre-transition stadial for NGRIP Ca2+. Seven of the sixteen events considered show significant downward slopes (p<0.05 assessed using phase-randomised Fourier surrogates).

"To test whether ice-core data show significant slopes in the stadial periods preceding DO events, we consider the data segments used by Erhardt et al. (2019). We only consider data up to the fifth percentile of the posterior distribution for the transition onset time, as given by Erhardt et al. (2019), to ensure that the abrupt transition itself does not influence our results. We also only consider the sixteen DO events where Erhardt et al. (2019) provide onset times for all four proxies. We calculate linear slopes and test the significance of these using phase-randomised Fourier surrogates, which preserve the autocorrelation structure of the data. We find that seven of the sixteen events show a significant slope (p<0.05) in Ca2+ that is in the same direction as the following abrupt transition (Figure A1), as well as four for

the Annual Layer Thickness and one for Na+. This demonstrates the need to consider the impact of pre-transition slopes on the accuracy of the ramp fitting method, and to test whether this can be improved by directly incorporating said slopes into the transition model. This is in addition to the more obvious need to consider post-transition slopes due to the classic "saw-tooth" shape of DO events."

*Capron et al. test 20 realisations of a synthetic ramp, and find no significant evidence of a bias in either the transition midpoint or duration. Neither of these are directly comparable to the transition onset time, which we focus on as we feel it is more physically meaningful when trying to understand the progression of DO events. Nonetheless, if there were a bias in the onset time then Capron et al. would surely have seen this reflected in either the midpoint or duration, and they do not in fact see this. As the referee notes, the uncertainties shown in Supplementary Figure 2 of Capron et al. are large, and so could include a bias of a few years. Even so, the test conducted by Capron et al. would seem to rule out decadal-scale bias of the kind that we find in our study, at least for this particular combination of transition shape and noise. We would therefore suggest that the transitions tested by Capron et al. happen to lie in a region of parameter space for which the bias is small, even at high levels of noise. When using the original Erhardt et al. implementation we find very little bias for synthetic transitions with no pre- or post-ramp slopes (see Additional Figure 2 on page 10 of this response). …*

**I think this is a likely explanation. I guess it also means that the bias problem is expected to be relatively small for most of the data sets of transitions derived from ice cores. …**

We are glad that the referee is satisfied by this explanation. As shown in Table 3, for the four different NGRIP proxies we estimate early biases between five and ten years when using the original Erhardt et al. Method. Whether or not these are "relatively small" is subjective, as it entirely depends on what they are being compared to. Nonetheless, we feel that these biases are sufficient that they could potentially lead to false conclusions regarding leads and lags between different proxies, and so we feel that it is important to document the presence of these biases and warn the community of this possibility.

*Although the shape of the deterministic ramps used by Capron et al. is not made clear, it is likely that they are flat before and after the ramp. …*

**Indeed, this is the case, and this should be clear from Capron et al.: "i.e., a linear change in the raw or logarithmically-transformed data between two stable states"**

*…Instead, we demonstrate that their finding depends strongly on the unstated assumption that there are no pre- or post-ramp slopes. …*

**Citing Capron et al.: "The assumption that the transitions are adequately described by a linear change from one stable state to another is not trivial and has been challenged previously, but neither our observations nor the current understanding of the nature of the transitions justifies employing a model with more degrees of freedom." The authors are very welcome to challenge this assumption, but it is hardly unstated.**

We apologise for failing to notice that Capron et al. do in fact spell this out clearly, and we thank the referee for drawing it to our attention.

**Line-by-line comments:**
**- Line 1 and throughout: Hyphens should be used in "sea-ice extent", "ice-core records" and other similar compound adjectives, but not when "ice core", "sea ice" etc. appear as nouns.**

We thank the referee for this grammatical correction, which we have implemented throughout.

**- Line 68. Revise grammar.**

We have rewritten this sentence as follows:
"Unlike the previous study, Capron et al. (2021) suggested that any leads and lags between climate elements might be impossible to detect due to both the tight coupling of the different climate elements and the substantial variability between different DO events."

**- Line 74-76 seems inconsistent with line 12.**

We apologise for this inconsistency. In line 12 we are describing the implications of our results, whereas in lines 74-76 we are discussing the state of affairs which motivated this research. Prior to this study, it was in our view possible that the application of the Erhardt et al. method to new data could still resolve the question of leads and lags between climate elements. As a result of our finding of bias, we feel that this now seems highly unlikely. We have written the last sentence of the paragraph including lines 74-76 in an attempt to reflect this, which we hope satisfies the referee:
"Nonetheless, the previous research in this area has left open the possibility that such an understanding could be achieved in future through the application of this method to either improved ice-core records or data from model simulations."

**- Line 84: Models provide a lot more direct insights into the dynamics than ice-core proxies, but "they provide complete information" seems like an overstatement.**

We agree with the referee and have removed this statement.

**- Line 109: "appears to depends" .. no s or just "depends" … the paper details the event duration's dependence on CO2 quite explicitly.**

We thank the referee for this suggestion and have changed "appears to depends" to "depends"

**- Line 109: The "chosen range" range does not make sense. The model oscillates with a range of CO2 values (which is not chosen), but the range of CO2 investigated is broader than this,**
The referee is entirely correct. We have removed this sentence as we do not feel that it is necessary.

**- Line 174: Missing "the" - Caption fig. 1: The last sentence does not apply to a) and d)**

We agree with the referee and have clarified in fig. 1 that the improvement is only visually clear for the CCSM4 model data in (e) and (f).

**- Line 191: in terms in terms**

We thank the referee for pointing out this mistake and have corrected it.

**- Line 243: Rather "synthetic data series"?**

We agree with the referee and have implemented this suggestion.

**- Line 279-280: yes, but especially for the Greenland stadial slope, the range of values used in the tests is much larger than the values observed in data, so the effect may look more dramatic than what is realistic.**

Whilst we agree that the majority of events in both the ice core and model data show Greenland stadial slopes that lie within a smaller range than that which we use, there are several individual events in NGRIP with slopes that are outside of our considered range. Table D2 shows that all four proxies in NGRIP have a posterior-mean Greenland stadial slope that is above the maximum of our considered range (1.8 kiloyears$^{-1}$) for at least one DO event. However, we agree that it is necessary to note that effect may be overdramatised here, and have extended the sentence on lines 281-282 so that it now reads as follows:

"We find that this leads to a too-early bias that can exceed 10 years, although the extremities of the range considered here (and so the largest biases) occur only rarely in the NGRIP data."

We hope that this caveat satisfies the referee.

**- Line 307: Challenging**

We thank the referee for this correction.

**- Line 424: Suggestion: One important caveat is that the bias we have identified is generally fairly small relative to the timing uncertainty of individual proxies across single DO events, …**

We have adopted this suggestion.

**- Line 427: Suggestion: may involve**

We have adopted this suggestion.

**- Line 433: It seems unlikely that 20 synthetic data series with different noise realizations would not show any bias if it indeed was a problem. A more likely explanation is given in the authors' comments: "We would therefore suggest that the transitions tested by Capron et al. happen to lie in a region of parameter space for which the bias is small, even at high levels of noise."**

We agree with the referee, and we thank them for this comment. We have condensed the paragraphs beginning on lines 430 and 435 into a single paragraph that includes this explanation, as follows:

"Capron et al. (2021) also test for possible bias using synthetic transitions with autocorrelated noise and find no significant bias in either the transition midpoint or duration, implying that the transition onset time must also be unbiased. This stands in contrast to our findings in this study, and so merits further consideration. We suggest that the transitions tested by Capron et al. (2021) lie in a region of the parameter space for which the bias is small, even at high levels of noise. The most obvious reason for this is that the absence of any slopes before or after the ramp favours small biases. We observe in Figure 3 that, when using the original Erhardt et al. (2019) implementation, synthetic transitions without any slopes before or after the ramp show very little bias, even for relatively high levels of noise. However, the bias grows rapidly when even slight slopes are present. We therefore suggest that a key cause of the discrepancy is the assumption made by Capron et al. (2021) that there are no slopes in either the stadial that precedes the transition or the interstadial that

follows. Similarly, Figure 3 shows that for certain "sweet spots" of the transition duration, the bias remains small even as the level of noise increases. It could also be the case that the transition duration chosen by Capron et al. happens to lie in such a sweet spot, and that this is partly why they do not find significant bias.

**- Line 473-477: It would be fair to mention around here – or elsewhere in the conclusion - that (significant) slopes are not always present in the data and that the original model outperforms the extended model in the absence of slopes in the data.**

We agree that this would be fair. We have therefore changed the sentence starting on line 476 from:
"Furthermore, this bias cannot be alleviated by incorporating said slopes directly into the ramp fitting method."

to:
"Directly incorporating said slopes into the ramp fitting method reduces the extent of the bias when significant slopes are in fact present in the data, but doing so also worsens the problem when they are not, as is often the case."